# On Gossip Algorithms for Machine Learning with Pairwise Objectives

**Igor Colin**                                                          *igor.colin@telecom-paris.fr*
*LTCI, Télécom Paris, Institut Polytechnique de Paris*

**Aurélien Bellet**                                                     *aurelien.bellet@inria.fr*
*PreMeDICaL team, Inria, Idesp, Inserm, Université de Montpellier*

**Stephan Clémençon**                                        *stephan.clemencon@telecom-paris.fr*
*LTCI, Télécom Paris, Institut Polytechnique de Paris*

**Joseph Salmon**                                                      *joseph.salmon@inria.fr*
*IMAG, IROKO, Univ Montpellier, Inria, CNRS*

**Reviewed on OpenReview:** *https://openreview.net/forum?id=VxxpURovJF*

## Abstract

In the IoT era, information is more and more frequently picked up by connected smart sensors with increasing, though limited, storage, communication and computation abilities. Whether due to privacy constraints or to the structure of the distributed system, the development of statistical learning methods dedicated to data that are shared over a network is now a major issue. Gossip-based algorithms have been developed for the purpose of solving a wide variety of statistical learning tasks, ranging from data aggregation over sensor networks to decentralized multi-agent optimization. Whereas the vast majority of contributions consider situations where the function to be estimated or optimized is a basic average of individual observations, it is the goal of this article to investigate the case where the latter is of pairwise nature, taking the form of a *U*-statistic of degree two. Motivated by various problems such as similarity learning, ranking or clustering for instance, we revisit gossip algorithms specifically designed for pairwise objective functions and provide a comprehensive theoretical framework for their convergence. This analysis fills a gap in the literature by establishing conditions under which these methods succeed, and by identifying the graph properties that critically affect their efficiency. In particular, a refined analysis of the convergence upper and lower bounds is performed.

## 1 Introduction

The era of Big Data and widespread artificial intelligence has begun. It uses technological building blocks to automatically collect, store and process massive data of different types and formats in a short space of time. The machine learning craze is spreading to almost every field, as the Internet of Things (IoT) and the widespread use of technology for analysis make ever more data available, at ever finer granularity. Expectations are immense. However, whether for predictive or interpretative purposes, the statistical analysis of data collected using modern technologies still raises a wide variety of methodological questions in order to design smarter devices. The abundance of new applications, such as monitoring the state of health of complex infrastructures, the availability of massive data samples and the technological constraints inherent in acquiring and accessing information (e.g. sensor networks) and computing, have pushed the scientific community to develop new algorithms, beyond the example of successful applications such as computer vision, machine-listening or automatic language translation. In the IoT era, information can be collected by connected smart sensors, shared across the network they form, and sometimes needs to be analyzed

in a *distributed* way, due to systemic or privacy constraints for example. Gossip algorithms have been designed to solve various statistical tasks in this context, such as data aggregation or decentralized multi-agent optimization, see *e.g.,* Loizou and Richtárik (2016), Zantedeschi et al. (2020), Xin et al. (2020a) or Hendrikx et al. (2020). While the situation where the functional of interest is a basic average of individual observations has received much attention in the literature, see *e.g.,* Agarwal et al. (2010), Boyd et al. (2011), Bianchi and Jakubowicz (2013), Xin et al. (2020a) or Scaman et al. (2018), the present article focuses on the case where the function to be estimated or to be optimized in a distributive manner is of *pairwise* nature, taking the form of a $U$-statistic of degree two (*i.e.,* an average over all pairs of individual observations), see Clémençon et al. (2008), Clémençon et al. (2016) or Laforgue et al. (2019). Indeed, many criteria that are relevant for evaluating the performance of decision rules in complex problems such as clustering, ranking, metric and similarity learning or graph reconstruction involve statistical quantities that are precisely of this form, see *e.g.,* Clémençon (2014), Clémençon and Vayatis (2010), Vogel et al. (2018) and Papa et al. (2016). This structural difference prevents the direct application of standard decentralized averaging techniques, and calls for dedicated algorithmic and theoretical tools.

Unlike local learning tasks such as classification or regression, these global problems require the computation of pairwise interactions. The main motivation of this paper is to propose and analyze gossip algorithms adapted to pairwise objectives.

Classically, it is assumed that agents can exchange a limited amount of information per unit of time, via a communication infrastructure modeled by a connected graph whose nodes they constitute. Merging all the data at a given node is not always possible, due to memory capacity constraints for example, nor necessarily desirable, for security and confidentiality reasons. It is therefore necessary to implement a distributed estimation or optimization strategy, based solely on the local calculations performed by the agents and on the communication enabled by the network structure.

Beyond communication constraints, gossip-based strategies are particularly well suited to realistic distributed settings such as IoT or edge networks, where each device has limited memory and computational resources. In such scenarios, storing or transmitting the entire dataset to a central node is often infeasible, and sometimes undesirable for privacy or security reasons. Gossip protocols also naturally tolerate temporary node failures, communication delays, and asynchronous operation, since information continues to propagate through alternative paths without requiring global synchronization. These properties make gossip-based algorithms a natural candidate for decentralized learning with pairwise objectives.

The assumption that all local data processing and communication can be synchronized by means of a global clock, to which all agents have access, can greatly simplify the analysis of distributed algorithms. However, the asynchronous framework, in which each node processes local data and manages its information transfer to its neighbors in the communication network according to its own clock (modeled as a homogeneous Poisson process) only, is also considered in the Appendix (the extension requires only minor adjustments).

In this article, we first investigate the issue of computing a $U$-statistic in a decentralized setting, *i.e.,* from a sample of observations that is partitioned and distributed over a network. Whereas much attention has been paid to the standard distributed averaging problem, in both synchronous and asynchronous cases, refer to Boyd et al. (2006) (see also Karp et al. (2000), Kempe et al. (2003), Silvestre et al. (2018) and Wang et al. (2017)), few methods have been proposed in the case of pairwise averages. A fundamental difficulty in the pairwise setting is that, unlike classical averaging, the observations required to form pairwise estimates are not initially available at all nodes. Instead, auxiliary data must be progressively disseminated across the communication graph by gossip interactions, inducing a transient and non-uniform sampling of pairs. Understanding how this non-uniformity impacts the bias and variance of local estimates over time is a central challenge addressed in this paper.

A dedicated algorithm for both synchronous and asynchronous settings is introduced in Colin et al. (2015), referred to as GoSta. This estimation procedure can be viewed as a mixture of local estimation of partial estimates and standard averaging method, *cf* Boyd et al. (2006), and attains a convergence rate of order $\mathcal{O}(1/t)$ after $t \geq 1$ iterations. Although a theoretical analysis was provided in Colin et al. (2015), it focused solely on the bias of the estimator.

In this article, we examine the expected error, thereby proposing the first theoretical guarantee of convergence for the procedure, as well as a variance result that can be used for concentration analyses or robust estimation applications.

We also address the problem of decentralized optimization in the case where the objective function is of the form of a $U$-statistic of degree two. Following in the footsteps of Duchi et al. (2012a), Colin et al. (2016) introduced and analyzed distributed synchronous and asynchronous optimization algorithms relying on the dual averaging of subgradients for pairwise functions. Beyond the rate bounds established, highlighting in particular the impact of the network's communication structure, the experimental results provided empirical evidence of the performance of the approach. However, the upper bound presented depends on a bias term related to the propagation of observations on the graph. Although numerical experiments suggested that this bias disappears quickly, its dependence on the optimization process did not allow one to draw firm conclusions about the convergence of the procedure.

We propose to extend gossip algorithms from single-observation objectives to pairwise objectives, and analyze their statistical and optimization properties. Our main contributions are threefold: (i) for estimation, we provide the first complete non-asymptotic analysis of the GoSta algorithm, including expectation and variance bounds; (ii) for optimization, we establish convergence guarantees and show that the bias present in previous works vanishes; (iii) we derive a novel lower bound tailored to the pairwise setting. In addition, we include numerical experiments illustrating the predicted convergence behavior, the decay of the bias induced by pairwise gossip sampling, and the role of the communication graph topology.

The article is organized as follows. In Section 2, the essential concepts of decentralized data processing are briefly reviewed, along with standard approaches to distributed estimation and optimization. The state of the art in the case of $U$-statistics is also presented for comparison with the results obtained in the following sections. Decentralized estimation of a $U$-statistic is addressed in Section 3. In Section 4, distributed optimization of a pairwise objective is considered, rate bounds are stated for both the error upperbound and lowerbound. Section 5 reports numerical experiments conducted on a real-world dataset, illustrating the practical behavior of the proposed algorithms. Some concluding remarks are collected and lines of further research are sketched in Section 6. Technical details and extension to the asynchronous setting are deferred to the Appendix section.

## 2 Background and Preliminaries

First, we describe the main features of decentralized systems in this section and then summarize the essential ideas underlying the gossip approach to distributed estimation and optimization. For the sake of completeness, we also briefly recall the basic notions of $U$-statistics theory and review the state of the art in distributed learning with pairwise objectives.

Here and throughout, the Euclidean norm of any vector $z \in \mathbb{R}^n$ with $n \geq 1$, viewed as a $n \times 1$ column vector, is denoted by $||z|| = (\sum_{k=1}^{n} z_k^2)^{1/2}$. By $\mathbf{1}_n = (1, \ldots, 1)$ is meant the vector in $\mathbb{R}^n$ whose coordinates are all equal to one, and by $\mathcal{B}(0, D) = \{z \in \mathbb{R}^n : ||z|| < D\}$ the ball centered at $\mathbf{0}_n = (0, \ldots, 0) \in \mathbb{R}^n$ with radius $D > 0$. The cardinality of any finite set $\mathcal{A}$ is denoted by $|\mathcal{A}|$.

### 2.1 Decentralized Setup - Motivation and Framework

A variety of modern applications, among which statistical estimation and signal processing in sensor networks (Zhao and Guibas, 2004), coordination in multi-agent systems, distributed localization and federated learning (Kairouz et al., 2021), require solving tasks across datasets that are naturally distributed across a large number of nodes. The design of efficient algorithmic solutions must take into account the many constraints that arise from this context. Depending on the use case, these constraints may include the following:

$C_1$ *There is no central server to centralize data or orchestrate calculations (or such centralization is deemed too costly);*

$C_2$ *Node-to-node communication over the network is costly;*

$C_3$ *The computing and storage capacities of each node are severely limited;*

$C_4$ *Network-wide synchronization of nodes significantly degrades performance.*

In this paper, we will consider decentralized problems in which at least constraint $C_1$ holds. Federated learning is therefore outside the scope of our analysis, although we do analyze synchronous contexts where a central *controller* orchestrates the learning process. Both decentralized and federated learning offer ways of mitigating the privacy risks and costs of centralized learning. Decentralized computing has tradition-ally been studied in the context of sensor networks (Zhao and Guibas, 2004). However, recent large-scale deployments of connected smart devices have also made the decentralized framework relevant to a variety of use cases in which nodes communicate with only a small subset of other peers, due to physical and/or efficiency constraints. It is worth pointing out that the relevance of decentralized computing has recently been demonstrated even in tightly-coupled systems (*e.g.,* data centers). A striking example is the distributed training of machine learning models across a large number of compute nodes (CPUs or GPUs) in a computing cluster, where using a 'master node' to aggregate intermediate computations quickly becomes a performance bottleneck, as it has to collect results from all the other nodes (see for instance Lian et al., 2017; Daily et al., 2018).

**Formal setup.** We now formally describe the framework considered here and introduce some key notations that will be used throughout the paper. We consider a set of $n \geq 1$ nodes (*e.g.,* agents, sensors, connected objects) indexed by $i \in \{1, \ldots, n\}$. Each node $i$ holds a local data sample $X_i$ from the data set $\mathcal{D}_n = \{X_1, \ldots, X_n\}$. The network topology is modeled as an undirected connected graph $\mathcal{G} = (\mathcal{V}, \mathcal{E})$, where the vertex set $\mathcal{V}$ is the set of nodes $[n] := \{1, \ldots, n\}$. The edge set $\mathcal{E}$ consists of unordered pairs of vertices from the set $\mathcal{V}$ and describes the communication constraints: for all $i \neq j$, $(i, j) \in \mathcal{E}$ if and only if nodes $i$ and $j$ can communicate directly with each other. Denoting $\mathbf{A}$ as the adjacency matrix (*i.e.,* $[\mathbf{A}]_{ij} = 1$ if and only if $(i, j) \in \mathcal{E}$, and $[\mathbf{A}]_{ij} = 0$ otherwise), the degree of node $i \in \mathcal{V}$ is given by $d_i = \sum_{j \neq i} [\mathbf{A}]_{ij}$ and represents the number of neighbors of $i$ in $\mathcal{G}$. In practice, keeping the maximum degree small, ideally independent of $n$, ensures that nodes do not have to communicate with a large number of peers, thereby taking into account constraints $C_2$ and $C_3$. The communication properties of the network, *i.e.,* the connectivity of the graph $G$, plays a crucial role in the performance of decentralized algorithms. They are generally described by the eigenvalues of the graph Laplacian, see *e.g.,* Chung (1997). This symmetric and positive semi-definite $n \times n$ matrix is defined by $\mathbf{L} = \mathbf{D} - \mathbf{A}$, where $\mathbf{D} = \mathrm{diag}(d_1, \ldots, d_n)$ is the degree matrix. Remarkably, up to a renormalization, the matrix $\mathbf{L}$ is the transition matrix of a random walk on the connected graph $\mathcal{G}$. Its smallest non-zero eigenvalue, the *spectral gap*, characterizes the stochastic stability of the random walk on $\mathcal{G}$: if the connected graph $\mathcal{G}$ is non-bipartite, this specific finite-state Markov chain is aperiodic, irreducible and geometrically ergodic, with the uniform distribution on $\mathcal{G}$ as limit distribution. The convergence occurs at a geometric rate that depends on the spectral gap in an explicit fashion, see Chung (1997).

**Gossip algorithms.** The general idea behind decentralized algorithms is that each node alternates between two stages: a local update stage and a communication stage with its neighbors. In this paper, we focus on a certain type of decentralized algorithms, namely *gossip* protocols (Shah, 2009; Dimakis et al., 2008). Rather than requiring communication with all neighbors, we consider a randomized gossip protocol in which each node exchanges information with only one *random neighbor* at each stage. Such randomized peer-to-peer communication makes gossip naturally resilient to temporary node failures or intermittent connectivity: if some devices disconnect or pause, information can still propagate along alternative paths without requiring global synchronization. Decentralized algorithms can be *synchronous* or *asynchronous*. In the synchronous setting, nodes have access to a global clock. At each time step $t$ (tick of the clock), all nodes perform a local update and a communication step, and the next step only begins when all nodes have finished. Although this assumption is not always realistic, and significant slowdowns can occur, we first focus our analysis on the synchronous framework, as it allows for more explicable rates. The asynchronous framework, where communication occurs randomly on the network and without a global clock, is next considered. One may refer to Boyd et al. (2006) for more details about synchronous and asynchronous time models. Two classes of decentralized problems have been widely studied: *estimation* and *optimization*. Let $f(\theta; X_1, \ldots, X_n)$ be some function of the data set $\mathcal{D}_n$ parameterized by $\theta \in \mathbb{R}^d$. Broadly speaking, estimation refers to the problem of computing (or approximating) the value of $f$ for a known value of $\theta$, while optimization aims at finding

the parameters $\theta$ that minimize $f$. Clearly, optimization appears more challenging than estimation, and the difficulty of solving these problems efficiently under the decentralized constraints described in Section 2.1 strongly depends on the structure of the function $f$. In fact, the vast majority of existing work has focused on decentralized estimation and optimization problems where $f$ is *separable across nodes*. Namely:

$$f(\theta; X_1, \ldots, X_n) = \frac{1}{n}\sum_{i=1}^{n} f_i(\theta; X_i) \ . \tag{1}$$

For estimation purposes, the case (1) encompasses many interesting statistics, particularly those in the form of sums and averages of quantities calculated from each data point. The thesis of Tsitsiklis (1984) is one of the earliest references on decentralized averaging, followed by a large number of more recent works (see *e.g.,* Gupta et al., 2001; Kempe and Kleinberg, 2002; Kempe et al., 2003; Xiao and Boyd, 2004; Boyd et al., 2006; Mosk-Aoyama and Shah, 2008, and references therein). More specifically, gossip algorithms have been studied for this problem in (Kempe et al., 2003; Boyd et al., 2006). Here we briefly describe the main ideas behind Boyd et al. (2006)'s classic gossip averaging algorithm. The algorithm operates in an asynchronous framework: nodes wake up asynchronously and average their local estimate with that of a randomly chosen neighbor. As long as the network graph is connected and non-bipartite, the local estimates converge to (1) at a rate $\mathcal{O}(e^{-ct})$ where the constant $c$ can be related to the spectral gap of the network graph, showing faster convergence for well-connected networks.

The optimization of separable functions has attracted a great deal of interest in recent decades, mainly, but not exclusively, in the context of federated learning (Kairouz et al., 2021). Several methods have been proposed to improve the convergence rate by extending Nesterov's acceleration (Even et al., 2021) or by using a more specific approach (Hendrikx et al., 2019), by optimizing the balance between communication and local computation (Scaman et al., 2018), or by means of variance reduction techniques (Xin et al., 2020b;a). Other approaches have tackled the non-i.i.d. framework, either for personalization or for multitask learning (Zantedeschi et al., 2020; Li et al., 2022b; Vanhaesebrouck et al., 2017; Dai et al., 2022). Another way of improving the convergence properties of decentralized and federated optimization is to tackle the communication bottleneck: quantization techniques developed for compressed sensing have been extended to the decentralized case to improve communication efficiency during the optimization process (Haddadpour et al., 2021; Albasyoni et al., 2020; Hamer et al., 2020; Li et al., 2022c).

Although privacy is one of the constraints motivating decentralized and federated learning, these designs remain vulnerable to various privacy attacks. However, local data storage allows communication to be modified through noise or encryption to provide satisfactory privacy guarantees, at the cost of slower learning (Kasyap and Tripathy, 2021; Ji et al., 2024; Jeon et al., 2021). Similar approaches have been adopted for robustness, where adapting communication rules can lead to robust optimization processes (Li et al., 2022a; Raynal et al., 2023; Zecchin et al., 2022). Finally, fairness in distributed learning is also an active research topic, as preserving fairness while limiting user information leakage – in the case of inter-device learning – leads to significantly lower learning rates (Biswas et al., 2024; Du et al., 2021; Jiang and Lu, 2019).

We point out that all the above approaches are restricted to functions that are separable across agents, as in (1). seen in the next section, many interesting estimation and optimization problems are only separable across *pairs* of agents, thus motivating the design of the algorithms described and analyzed in Section 4, tailored to pairwise objectives. A key aspect of pairwise objectives in decentralized settings is that, unlike separable averaging, the information required to form pairwise interactions is *not* initially available at every node: nodes start with one local observation and must progressively acquire additional (auxiliary) observations through gossip exchanges. As a consequence, pairwise estimates and gradients are formed from a transiently *non-uniform* sampling of pairs, which only approaches the uniform distribution after sufficient mixing of the auxiliary variables. Quantifying the effect of this progressive dissemination on bias and variance is one of the main technical challenges addressed in our analysis.

## 2.2 Pairwise Averages and $U$-statistics: Definition and Examples

In many situations, the quantity of interest is not a simple mean $\mathbb{E}[f(\theta; X)]$ taken w.r.t. the distribution of a r.v. $X$ but takes the form $\mu(h) = \mathbb{E}[h(\theta; X, X')]$, where $X$ and $X'$ are two independent and identically

distributed random vectors, taking their values in some measurable space $\mathcal{X}$ with probability distribution $F(\mathrm{d}x)$ and $h : \mathcal{X} \times \mathcal{X} \to \mathbb{R}$ is a symmetric measurable mapping, square integrable w.r.t. the product measure $F \otimes F$.

A natural estimator of $\mu(h)$ based on an i.i.d. sample $\mathcal{D}_n = \{X_1, \ldots, X_n\}$ drawn from $F$ is the average over all pairs

$$\widehat{U}_n(h) = \frac{2}{n(n-1)} \sum_{1 \leq i < j \leq n} h(X_i, X_j) \ . \tag{2}$$

The quantity (2) is known as the $U$-statistic of degree two with kernel $h$ based on the sample $\mathcal{D}_n$. For $U$-statistics of degree higher than two, we refer the reader to Appendix B. As may be shown by a classic Lehmann-Scheffé argument, it is the unbiased estimator of $\mu(h)$ with minimum variance. However, the reduced variance property has a price when it comes to analyzing the fluctuations of such a functional (uniformly over a class of kernels possibly), the terms averaged in (2) exhibiting a complex dependence structure. This technical difficulty can be overcome in order to prove limit theorems (*e.g.,* SLLN, CLT, LIL) for such statistics by means of *linearization techniques (i.e.,* Hoeffding decomposition, Hajek projection), see *e.g.,* Lee (1990) for an account of the asymptotic theory of $U$-statistics. Similar approaches can also be used to establish concentration bounds for $U$-statistics; see Clémençon et al. (2008) and the references therein for further details. See also Clémençon et al. (2016) for scalability challenges tackled by means of sampling schemes and Laforgue et al. (2019) for robustness issues. Many commonly used statistics measuring dispersion, such as empirical variance and the Gini mean difference, take the form of a $U$-statistic (2). Similarly, in various statistical learning problems – whether supervised or unsupervised – the empirical risk criterion can also be expressed as a $U$-statistic.

**Metric learning.** Numerous machine learning methods involve a metric between data points. The performance of such techniques is crucially determined by the choice of an adequate metric. It is precisely the goal of metric learning to adapt the metric to the data analyzed. Motivated by many applications (*e.g.,* computer vision, information retrieval), this problem has been recently the subject of much attention in the literature, see for instance Bellet et al. (2013) and the references therein. Inspired by biometric identification problems, it is formulated in Vogel et al. (2018) in the framework of supervised multi-class classification: a random label $Y$, taking values in a finite set $[K]$ with $K \geq 2$, is assigned to a random observation $X$ valued in a feature space $\mathcal{X}$. Given independent copies $(X_1, Y_1), \ldots, (X_n, Y_n)$ of the random couple $(X, Y)$, the learning task consists in finding a metric $D : \mathcal{X} \times \mathcal{X} \to \mathbb{R}_+$ such that pairs of points with the same label are close, while pairs with different labels are distant from each other. The risk of a metric candidate $D$ can be formulated as follows:

$$R(D) = \mathbb{E}\left[\phi\left((1 - D(X, X')) \cdot (2\mathbb{I}\{Y = Y'\} - 1)\right)\right] \ ,$$

where $(X', Y')$ is an independent copy of $(X, Y)$ and $\phi(u)$ is a convex loss function upper bounding the indicator function $\mathbb{I}\{u \geq 0\}$ (*e.g.,* the hinge loss $\phi(u) = \max(0, 1 - u)$). Its empirical version is the $U$-statistic of degree two:

$$R_n(D) = \frac{2}{n(n-1)} \sum_{1 \leq i < j \leq n} \phi\left((1 - D(X_i, X_j)) \cdot (2\mathbb{I}\{Y_i = Y_j\} - 1)\right) \ .$$

**Ranking.** Given objects described by a random vector of features $X \in \mathcal{X}$ and the (temporarily hidden) ordinal and real valued labels $Y$ assigned to them, the goal of *supervised ranking* is to rank them in the same order as that induced by the labels, on the basis of a training set of labeled examples. This statistical learning problem finds its motivation in a wide variety of applications, *e.g.* medical diagnosis support, search engines, credit-risk screening, e-commerce. Rankings are generally defined by means of a scoring function $s : \mathcal{X} \to \mathbb{R}$, transporting the natural order on the real line onto the feature space. Ideally, the larger $s(X)$, the larger the label $Y$ with overwhelming probability. Learning such a scoring function $s$ from independent

labeled data $(X_1, Y_1)$, ..., $(X_n, Y_n)$ drawn as the generic random pair $(X, Y)$ can be formulated as the problem of minimizing the $U$-statistic known as the *empirical ranking risk*, refer to Clémençon et al. (2005):

$$\mathcal{L}_n(s) = \frac{2}{n(n-1)} \sum_{1 \leq i < j \leq n} \ell\left(-(s(X_i) - s(X_j)) \cdot (Y_i - Y_j)\right) , \tag{3}$$

where $\ell : \mathbb{R} \to \mathbb{R}_+$ is a given loss function, *e.g.*, $\ell(u) = \mathbb{I}\{u > 0\}$. The quantity (3) is a $U$-statistic of degree two with kernel $h_s((x, y), (x', y')) = \ell(-(s(x) - s(x')) \cdot (y - y'))$ for $(x, y)$ and $(x', y')$ in $\mathcal{X} \times \mathbb{R}$. Observe incidentally that, in the case where the label $Y$ is binary (*i.e.,* bipartite ranking), the ranking risk is equal to the popular AUC criterion, up to an affine transform.

**Clustering.** The unsupervised learning task one completes when segmenting a dataset $X_1, \ldots, X_n$ in a feature space $\mathcal{X}$ into finite subgroups depending on their similarity is referred to as clustering: observations in the same segment should be more similar to each other than to those lying in other segments. See *e.g.* Chapter 14 in Friedman et al. (2009) for a review of popular clustering methods. More precisely, consider a symmetric function $D : \mathcal{X} \times \mathcal{X} \to \mathbb{R}_+$ such that $D(x, x) = 0$ for any $x \in \mathcal{X}$. $D$ measures the dissimilarity between pairs of observations $(x, x') \in \mathcal{X}^2$: the smaller the quantity $D(x, x')$, the more similar $x$ and $x'$. Fix also the number of desired clusters $M \geq 2$. It is the objective of clustering methods to find a partition $\mathcal{P}$ of the feature space $\mathcal{X}$ in a class $\Pi$ of partition candidates that minimizes the following *empirical clustering risk*:

$$R_n(\mathcal{P}) = \frac{2}{n(n-1)} \sum_{1 \leq i < j \leq n} D(X_i, X_j) \cdot \Phi_{\mathcal{P}}(X_i, X_j) , \tag{4}$$

where $\Phi_{\mathcal{P}}(x, x') = \sum_{\mathcal{C} \in \mathcal{P}} \mathbb{I}\{(x, x') \in \mathcal{C}^2\}$ indicates whether two points $x$ and $x'$ belongs to the same cell of the partition or not. When the $X_i$'s are i.i.d. realizations of a generic r.v. $X$ with distribution $F(\mathrm{d}x)$, the quantity (4), usually referred to as the *intra-cluster similarity* or *within cluster point scatter*, is a $U$-statistic of degree two with kernel $h_{\mathcal{P}}(x, x') = D(x, x') \cdot \Phi_{\mathcal{P}}(x, x')$ for all $(x, x') \in \mathcal{X}^2$, provided that $\iint_{(x,x') \in \mathcal{X}^2} D^2(x, x') \Phi_{\mathcal{P}}(x, x') F(\mathrm{d}x) F(\mathrm{d}x') < +\infty$. The statistical analysis of the clustering performance of minimizers of (4) over a class $\Pi$ of appropriate complexity can be found in Clémençon (2014).

**Remark.** (*U*-STATISTICS VS *V*-STATISTICS) *When the symmetric kernel function $h : \mathcal{X} \times \mathcal{X} \to \mathbb{R}$ is such that $h(x, x) = 0$ for all $x \in \mathcal{X}$ (which is the case for most kernels used in the problems mentioned above), we naturally have: $\sum_{i \neq j} h(X_i, X_j) = \sum_{i,j} h(X_i, X_j)$. However, when this is not the case, the quantity $\sum_{i,j} h(X_i, X_j)$ is classically referred to as a $V$-statistic—when divided by $n^2$—and is obviously obtained by adding to the pairwise summation $\sum_{i \neq j} h(X_i, X_j)$ the basic i.i.d. sum $\sum_i h(X_i, X_i)$. For simplicity, with a slight abuse of language, $(1/n^2) \sum_{i,j} h(X_i, X_j)$ will be called a $U$-statistic throughout the article, the possible presence of diagonal terms in the functional of interest having no impact on the algorithms proposed and their analysis.*

Although the field of decentralized learning has received a lot of attention, especially since the advent of federated learning, work on pairwise decentralized learning remains scarce. This problem was first investigated in Pelckmans and Suykens (2009), where a method for estimating *partial sums* of a $U$-statistic was proposed and then extended to full estimation by requiring two gossip protocols to run in parallel. This method was further refined in Colin et al. (2015), using only one gossip protocol in contrast, along with refined convergence rates for both synchronous and asynchronous settings. To the best of our knowledge, the pairwise optimization problem has only been tackled in Colin et al. (2016), using an algorithm that combines distributed dual averaging (Duchi et al., 2010b) with pairwise estimation (Colin et al., 2015). The rates derived in Colin et al. (2016) provide some intuition about the problem characteristics, but do not guarantee convergence of the method, as they only show convergence in the case where the gradient biased estimates actually converge to the true gradient. Our new analysis shows that this is true without any additional assumption, and provides a better understanding of the impact of bias on the optimization process. A key element in our analysis is to show that the gradient bias induced by the auxiliary-observation propagation decays at a rate controlled by the mixing properties of the communication graph (in particular, through its spectral gap), which leads to explicit non-asymptotic convergence guarantees.

# 3 Decentralized Estimation of a Pairwise Functional

To begin with, we consider the decentralized estimation problem for a pairwise functional. Formally, let $(\mathbf{x}_1, \ldots, \mathbf{x}_n) \in \mathcal{X}^n$ be a sample of $n \geq 2$ points in a feature space $\mathcal{X} \subseteq \mathbb{R}^d$ with $d \geq 1$ and let $h : \mathcal{X} \times \mathcal{X} \to \mathbb{R}$ be a measurable function, symmetric in its two arguments. The goal pursued here is to estimate, in the decentralized framework described in Section 2.1 and with non-asymptotic guarantees, the following quantity:

$$\widehat{U}_n(h) = \frac{1}{n^2} \sum_{i,j=1}^{n} h(\mathbf{x}_i, \mathbf{x}_j) \ . \tag{5}$$

Let $F$ be a probability distribution on $\mathcal{X}$. As pointed out in Remark 2.2, the statistic (5) slightly differs from the $U$-statistic (2), which forms an unbiased estimator with minimum variance of the parameter $\mathbb{E}[h(X, X')] = \int \int h(x, x') F(dx) F(dx')$ as soon as $h$ is square integrable w.r.t. $F \otimes F$ and the $\mathbf{x}_i$'s are independent realizations of the distribution $F$: their difference is of order $O_{\mathbb{P}}(n^{-3/2})$ and (2) differs from (5) by a factor of $n/(n-1)$ when $h(x,x) = 0$ for all $x \in \mathcal{X}$. When clear from context, we will omit to index the functional (5) by $h$ and will use the notation $\widehat{U}_n$. We define $\mathbf{H} \in \mathbb{R}^{n \times n}$ such that $[\mathbf{H}]_{kl} := h(\mathbf{x}_k, \mathbf{x}_l)$ for all $1 \leq k, l \leq n$ and $\overline{\mathbf{h}} = \mathbf{H} \mathbf{1}_n / n$ as the vector of partial sums. Observe that $\widehat{U}_n = \mathbf{1}_n^\top \overline{\mathbf{h}} / n$.

## 3.1 GoSta Algorithm

In Boyd et al. (2006) and other related papers, the gossip averaging methods rely on activated neighbors averaging their local estimates in order to converge to the network mean. Such algorithms cannot be extended to efficiently compute (5) as it depends on *pairs* of observations. Indeed, in the separable averaging case, the values to be averaged are already locally available from the start, whereas in the pairwise case each node must progressively acquire *auxiliary* observations from other nodes in order to form pairwise evaluations $h(\mathbf{x}_i, \mathbf{x}_j)$. As a consequence, the pairs effectively used in local updates are non-uniform until the auxiliary observations have sufficiently mixed over the graph. This problem is investigated in Pelckmans and Suykens (2009) with U1-GOSSIP and U2-GOSSIP algorithms. In U1-GOSSIP, each node $i \in [n]$ estimates its partial $U$-statistic:

$$\widehat{U}_n^{(i)} := \frac{1}{n} \sum_{j=1}^{n} h(\mathbf{x}_i, \mathbf{x}_j) \ . \tag{6}$$

The spirit of the algorithm is a bit different from the standard gossip case. For each node $k \in [n]$, an estimator $z_k(t)$ is initialized to zero and an auxiliary observation $\mathbf{y}_k(t)$ is initialized to $\mathbf{x}_k$. At each iteration $t \geq 1$, if a communication is initiated between nodes $i$ and $j$, they swap their auxiliary observations:

$$\mathbf{y}_i(t) = \mathbf{y}_j(t-1) \text{ and } \quad \mathbf{y}_j(t) = \mathbf{y}_i(t-1) \ .$$

Then, *every* node updates its estimator using its (local) pair of observations:

$$\forall k \in [n], \ \ z_k(t) = \frac{t-1}{t} z_k(t-1) + \frac{1}{t} h(\mathbf{x}_k, \mathbf{y}_k(t)) \ .$$

Both algorithms are detailed in Appendix C.3 and C.4. Introduced in Colin et al. (2015), GoSta algorithm is based on the observation that $\widehat{U}_n = n^{-1} \sum_{i=1}^{n} \widehat{U}_n^{(i)}$, where $\widehat{U}_n^{(i)}$ are the partial sums defined in (6). The goal is thus similar to the usual gossip averaging problem, with the key difference that each local value $\widehat{U}_n^{(i)}$ is itself an average depending on the entire data sample. Consequently, our algorithm combines two steps at each iteration: a data propagation step to allow each node $i$ to estimate $\widehat{U}_n^{(i)}$, and an averaging step to ensure convergence to the desired value $\widehat{U}_n$. Here, we only present the algorithm for the—simpler—synchronous setting in Algorithm 1, the asynchronous version being detailed in the optimization case.

## 3.2 Non-asymptotic Estimation Bounds

We now carry out a non-asymptotic analysis of the GoSta-sync algorithm presented in Section 3.1. The following result is originally stated in Colin et al. (2015) and provides a bound in expectation taken w.r.t.

---

**Algorithm 1** GoSta-sync

---

**Require:** Each node $k \in [n]$ holds observation $\mathbf{x}_k$
1: Each node $k \in [n]$ initializes its auxiliary observation $\mathbf{y}_k = \mathbf{x}_k$ and its estimate $z_k = 0$
2: **for** $t = 1, 2, \ldots$ **do**
3:      **for** $p = 1, \ldots, n$ **do**
4:          $z_p \leftarrow \frac{t-1}{t} z_p + \frac{1}{t} h(\mathbf{x}_p, \mathbf{y}_p)$
5:      **end for**
6:      Draw $(i, j)$ uniformly at random in $E$
7:      $z_i, z_j \leftarrow \frac{1}{2}(z_i + z_j)$
8:      Swap auxiliary observations of nodes $i$ and $j$: $\mathbf{y}_i \leftrightarrow \mathbf{y}_j$
9: **end for**
10: **return** Each node $k \in [n]$ has the estimate $z_k$ at disposal

---

the random edge activations for the deviation between the estimate produced by Algorithm 1 and the target $\widehat{U}_n$. More precisely, the expectation in Proposition 1 is taken with respect to the algorithmic randomness (random edge activations and auxiliary-observation swaps), for a fixed dataset $(\mathbf{x}_1, \ldots, \mathbf{x}_n)$.

**Proposition 1** (Convergence in expectation). *Let $\mathcal{G} = ([n], \mathcal{E})$ be a connected and non-bipartite graph, $(\mathbf{x}_1, \ldots, \mathbf{x}_n)$ a sample of $n \geq 2$ points in $\mathcal{X} \subset \mathbb{R}^d$ and $(\mathbf{z}(t))_{t \geq 1}$ the sequence of estimates generated by Algorithm 1. For all $k \in [n]$ and any $t \geq 1$, we have:*

$$\left\| \mathbb{E}[\mathbf{z}(t)] - \widehat{U}_n \mathbf{1}_n \right\| \leq \frac{|\mathcal{E}|}{t \lambda_{n-1}} D(h) \ ,$$

*where $\lambda_{n-1}$ is the second smallest eigenvalue of the graph Laplacian $\mathbf{L}$ and*

$$D(h) := \left\| \overline{\mathbf{h}} - \widehat{U}_n \mathbf{1}_n \right\| + 2 \left\| \mathbf{H} - \overline{\mathbf{h}} \mathbf{1}_n^\top \right\|_{\mathrm{F}} \ .$$

Proposition 1 shows that all the local estimates generated by Algorithm 1 converge in expectation to $\widehat{U}_n$ at a rate of order $O(1/t)$. In addition, the constants involves in the rate bound above reveal the impact of graph structure and of the distribution of the data points on it on convergence. Indeed, the quantity $D(h)$ is *data-dependent* and reflects the difficulty of the estimation problem itself through a measure of dispersion. In contrast, $|\mathcal{E}|/\lambda_2$ is a *network-dependent* term since $\lambda_{n-1}$ is the second smallest eigenvalue of the graph Laplacian $\mathbf{L}$. The value $\lambda_{n-1}$ is also known as the spectral gap of $\mathcal{G}$ and graphs with a larger spectral gap typically have better connectivity, see Chung (1997) for a detailed analysis of the graph Laplacian notion. It should be noticed that pairwise estimation rates are slower than those in mean estimation, which are geometric (Boyd et al., 2006). This is due to the fact that mean estimation relies on values already computed by the nodes, whereas pairwise estimation requires the nodes to estimate these quantities while averaging them.

In the separable averaging setting, each node starts with a locally available value and the only source of error is the mixing of the averaging dynamics, which yields geometric convergence. In contrast, in the pairwise setting each node must first estimate its partial sum by exchanging auxiliary observations. At early times, the auxiliary observations have not mixed yet, so the pairs $(\mathbf{x}_k, \mathbf{y}_k(t))$ used in the updates are drawn from a non-uniform distribution over the dataset, which induces bias. Although Proposition 1 guarantees that the bias of the estimate decreases in $O(1/t)$, it does not ensure that the (random) estimate converges to the target (5). In particular, controlling $\|\mathbb{E}[\mathbf{z}(t)] - \widehat{U}_n \mathbf{1}_n\|$ alone does not prevent fluctuations of $\mathbf{z}(t)$ around its mean. In the following theorem, we state an upper bound on the expected gap, therefore providing such a guarantee. This result is also helpful when considering robustness constraints or guarantees in high probability.

**Theorem 1** (Expected deviation). *Let $\mathcal{G} = ([n], \mathcal{E})$ be a connected and non-bipartite graph, $(\mathbf{x}_1, \ldots, \mathbf{x}_n)$ a sample of $n \geq 2$ points in $\mathcal{X} \subset \mathbb{R}^d$ and $(\mathbf{z}(t))_{t \geq 1}$ the sequence of estimates generated by Algorithm 1. For all $k \in [n]$ and any $t \geq 1$, we have:*

$$\mathbb{E} \left\| \mathbf{z}(t) - \widehat{U}_n \mathbf{1}_n \right\| \leq \frac{1}{\sqrt{t}} \cdot \sqrt{\left( 1 + \frac{2|\mathcal{E}|}{\lambda_{n-1}} \right) \left\| \mathbf{H} - \overline{\mathbf{h}} \mathbf{1}_n^\top \right\|_{\mathrm{F}}^2 + \frac{4|\mathcal{E}|}{\lambda_{n-1}} \left( 1 + \frac{1}{2t} \right) \|\mathbf{H}\|_{\mathrm{F}}^2} \ ,$$

where $\lambda_{n-1}$ is the second smallest eigenvalue of the graph Laplacian $\mathbf{L}$.

We almost recover a convergence rate of order $O(1/\sqrt{t})$, up to a factor of $\sqrt{2 + 1/t}$. As can be seen by examining the technical proof given in Appendix C.2, this term is a consequence of the bias of the early estimates propagating through the network, but it quickly vanishes.

## 4 Decentralized Optimization of a Pairwise Objective Function

Motivated by the statistical learning problems mentioned in Section 2.2, we now tackle the problem of *minimizing* an objective function of the form of a *pairwise* average of the network agents' data, such as (5), with respect to a parameter $\boldsymbol{\theta}$ on which it depends:

$$F(\boldsymbol{\theta}) := \frac{1}{n^2} \sum_{1 \leq i,j \leq n} f(\boldsymbol{\theta}; \mathbf{x}_i, \mathbf{x}_j) \ , \tag{7}$$

where $\Theta \subseteq \mathcal{B}(\mathbf{0}_d, D) \subseteq \mathbb{R}^d$ is the parameter space with $D > 0$ and, for $1 \leq i, j \leq n$, the function $f_{ij} := f(\cdot; \mathbf{x}_i, \mathbf{x}_j) : \Theta \subset \mathbb{R}^d \to \mathbb{R}$ is $L$-Lipschitz with $L > 0$, convex and possibly non-smooth. Finally, for $1 \leq i \leq n$, we denote $f_i := (1/n) \sum_{j=1}^n f_{ij}$ the partial objectives. In general, we will use $\widehat{\cdot}$ notation for network averages, and $\bar{\cdot}$ notation for temporal averages.

Again, the main difficulty in solving the optimization problem

$$\min_{\boldsymbol{\theta} \in \Theta} F(\boldsymbol{\theta}) \tag{8}$$

arises from the fact that each term of the sum depends on *two* agents $i$ and $j$, making standard local update schemes impossible unless data is exchanged between nodes. In contrast to separable objectives, where each node can compute (unbiased) gradients of its local loss immediately, the pairwise setting requires each node to progressively acquire auxiliary observations from other nodes. As a result, the pairs used in early iterations are non-uniform over the dataset, and the corresponding gradient estimates are biased until sufficient mixing has occurred on the communication network. To the best of our knowledge, efficient search of a solution $\boldsymbol{\theta}^\star$ of (8) in a decentralized setting has only been tackled in Colin et al. (2016).

The methods of Colin et al. (2016) rely on *dual averaging* (Nesterov, 2009; Agarwal et al., 2010)[1]. This choice is guided by the fact that dual averaging is often easier to analyze than subgradient descent in constrained or regularized settings, because it maintains a linear accumulator of (sub)gradients while the (non-linear) smoothing/projection operator is applied separately:

$$\pi_t : \begin{cases} \mathbb{R}^d & \to \quad \Theta \\ \mathbf{z} & \mapsto \quad \arg\max_{\boldsymbol{\theta} \in \Theta} \left\{ \gamma(t) \boldsymbol{\theta}^\top \mathbf{z} - \|\boldsymbol{\theta}\|^2 / 2 \right\} \end{cases} \tag{9}$$

This work builds upon the analysis carried out in Duchi et al. (2012a), where a distributed dual averaging algorithm is proposed to optimize an average of *separable* functions $f(\cdot; \mathbf{x}_i)$. In that setting, each node $i$ can compute *unbiased* estimates of its local (sub)gradient $\nabla f(\cdot; \mathbf{x}_i)$ that are iteratively averaged over the network—see Appendix D.1.3 for details. Unfortunately, there is a major difference with the optimization problem considered here. In our setting, node $i$ cannot compute unbiased estimates of $\nabla f_i(\cdot) = \nabla (1/n) \sum_{j=1}^n f(\cdot; \mathbf{x}_i, \mathbf{x}_j)$ because the latter depends on all data points, while each node $i \in [n]$ only holds $\mathbf{x}_i$. To circumvent this, the algorithm relies on a gossip data propagation step similar to that introduced in Section 3 so that nodes can compute *biased* estimates of $\nabla f_i(\cdot)$ while keeping communication and memory overheads low for each node.

We first recall the pairwise optimization algorithm in the synchronous context in Section 4.1. We then provide a refined analysis of the gradient bias via an ergodic argument in Section 4.2. Finally, we prove a lower-bound result for pairwise decentralized optimization procedures in Section 4.3.

---

[1]For completeness, background on (distributed) dual averaging is provided in Appendix D.1.1.

---

**Algorithm 2** Gossip dual averaging for pairwise function in the synchronous case

---

**Require:** Step size $(\gamma(t))_{t \geq 1} > 0$
1: Each node $i \in [n]$ initializes $\mathbf{y}_i = \mathbf{x}_i$, $\mathbf{z}_i = \boldsymbol{\theta}_i = \bar{\boldsymbol{\theta}}_i = 0$
2: **for** $t = 1, \ldots, T$ **do**
3:     Draw $(i, j)$ uniformly at random in $E$
4:     $\mathbf{z}_i, \mathbf{z}_j \leftarrow \frac{\mathbf{z}_i + \mathbf{z}_j}{2}$
5:     Swap auxiliary observations: $\mathbf{y}_i \leftrightarrow \mathbf{y}_j$
6:     **for** $k = 1, \ldots, n$ **do**
7:         $\mathbf{z}_k \leftarrow \mathbf{z}_k + \nabla_{\boldsymbol{\theta}} f(\boldsymbol{\theta}_k; \mathbf{x}_k, \mathbf{y}_k)$
8:         $\boldsymbol{\theta}_k \leftarrow \pi_t(\mathbf{z}_k)$
9:         $\bar{\boldsymbol{\theta}}_k \leftarrow \left(1 - \frac{1}{t}\right) \bar{\boldsymbol{\theta}}_k + \frac{1}{t} \boldsymbol{\theta}_k$
10:    **end for**
11: **end forreturn** Each node $k \in [n]$ has $\bar{\boldsymbol{\theta}}_k$

---

### 4.1 Decentralized Pairwise Function Optimization

In the synchronous framework, we assume that each node has access to a global clock, so that each node can be updated simultaneously with each clock tick. We assume that the scaling sequence $(\gamma(t))_{t \geq 1}$ is the same for every node. At any time $t \geq 1$, each node $i \in [n]$ stores: a gradient accumulator $\mathbf{z}_i$, its original observation $\mathbf{x}_i$, and an *auxiliary observation* $\mathbf{y}_i$, initialized at $\mathbf{x}_i$ and evolving through data propagation. The algorithm is summarized in Algorithm 2. At each iteration, an edge $(i, j) \in E$ is drawn uniformly at random. Nodes $i$ and $j$ average their accumulators and swap their auxiliary observations. Finally, *all* nodes perform a dual averaging step, using their local pair $(\mathbf{x}_k, \mathbf{y}_k)$ to form a (biased) partial gradient estimate.

**Average iterate and gradient bias.** For any $t > 0$, define the average accumulator $\hat{\mathbf{z}}(t) := \frac{1}{n} \sum_{i=1}^n \mathbf{z}_i(t)$ and the corresponding average iterate

$$\widehat{\boldsymbol{\omega}}(t) := \pi_t(\hat{\mathbf{z}}(t)) \ .$$

Then, we denote by $\widehat{\boldsymbol{\epsilon}}(t)$ the average gradient bias across the network:

$$\widehat{\boldsymbol{\epsilon}}(t) := \frac{1}{n} \sum_{i=1}^n \left( \nabla f(\boldsymbol{\theta}_i(t); \mathbf{x}_i, \mathbf{y}_i(t)) - \frac{1}{n} \sum_{j=1}^n \nabla f(\boldsymbol{\theta}_i(t); \mathbf{x}_i, \mathbf{x}_j) \right) \ .$$

Equipped with these notations, we can state the following result (initially stated in Colin et al. (2016)), which adapts the convergence rate of centralized dual averaging under a term capturing the contribution of the bias.

**Proposition 1.** *Let $\mathcal{G} = ([n], \mathcal{E})$ be a connected and non-bipartite graph, and let $\boldsymbol{\theta}^\star \in \arg\min_{\boldsymbol{\theta} \in \Theta} F(\boldsymbol{\theta})$. Let $(\gamma(t))_{t \geq 1}$ be a non-increasing and non-negative sequence. For any $i \in [n]$ and any $t \geq 0$, let $\mathbf{z}_i(t) \in \mathbb{R}^d$ and $\boldsymbol{\theta}_i(t) \in \mathbb{R}^d$ be generated according to Algorithm 2. Then, for any $i \in [n]$ and $T > 1$, we have:*

$$\mathbb{E}\left[ F(\bar{\boldsymbol{\theta}}_i(T)) - F(\boldsymbol{\theta}^\star) \right] \leq C_1(T) + C_2(T) + C_3(T) \ ,$$

*where*

$$\begin{cases} C_1(T) = \dfrac{1}{2T\gamma(T)} \|\boldsymbol{\theta}^\star\|^2 + \dfrac{L^2}{2T} \sum_{t=1}^{T-1} \gamma(t) \\[2ex] C_2(T) = \dfrac{3L^2}{T\left(1 - \sqrt{1 - \lambda_{n-1}/|\mathcal{E}|}\right)} \sum_{t=1}^{T-1} \gamma(t) \\[2ex] C_3(T) = \dfrac{1}{T} \sum_{t=1}^{T-1} \mathbb{E}_t[(\widehat{\boldsymbol{\omega}}(t) - \boldsymbol{\theta}^\star)^\top \widehat{\boldsymbol{\epsilon}}(t)] \end{cases} \ ,$$

*and $\lambda_{n-1}$ is the second smallest eigenvalue of the graph Laplacian $\mathbf{L}$.*

We refer the reader to Appendix D.2 for a detailed proof. As in the estimation case, the bound in Proposition 1 decomposes into: a *centralized* term $C_1(T)$, a *network disagreement* term $C_2(T)$ depending on the spectral gap, and a *pairwise-specific* term $C_3(T)$ coming from the gradient bias induced by auxiliary-data propagation.

Importantly, Proposition 1 by itself does not guarantee convergence unless one controls $C_3(T)$. While Colin et al. (2016) observed empirically that the bias term vanishes quickly, the upper bound does not provide an explicit non-asymptotic guarantee that $C_3(T)$ vanishes. In the next section, we refine the analysis by explicitly quantifying the bias decay in terms of the mixing properties of the auxiliary-observation process on the graph.

## 4.2 Pairwise Ergodic Dual Averaging - Gradient Bias Convergence Analysis

We now focus on a slightly different objective from that in (8). For $i \in [n]$, define $\tilde{F}_i : \Theta \times \Delta_n$ as follows:

$$\tilde{F}_i : \begin{cases} \Theta \times \Delta_n & \to & \mathbb{R} \\ (\theta, \boldsymbol{\xi}) & \mapsto & \sum_{j=1}^n \xi_j f_{ij}(\theta) \end{cases} ,$$

where $\Delta_n = \{\boldsymbol{\xi} \in \mathbb{R}_+^n : \|\boldsymbol{\xi}\|_1 = 1\}$ is the simplex in $\mathbb{R}^n$. Note that

$$F(\theta) = \frac{1}{n} \sum_{i=1}^n \tilde{F}_i\left(\theta, \frac{\mathbf{1}_n}{n}\right) = \frac{1}{n} \sum_{i=1}^n f_i(\theta) .$$

Let $\mathcal{S}_n = \{\mathbf{e}_1, \ldots, \mathbf{e}_n\}$ be the canonical basis of $\mathbb{R}^n$ and let $(\boldsymbol{\xi}_i(t))_{t \geq 0}$ be a sequence of (not necessarily independent) random variables over $\mathcal{S}_n$ identified with $[n]$. For $t \geq 0$, denote by $P_i(t)$ the distribution of $\boldsymbol{\xi}_i(t)$ and assume

$$\lim_{t \to +\infty} \|P_i(t) - \mathbf{1}_n/n\|_{\mathrm{TV}} = 0 ,$$

where $\|\cdot\|_{\mathrm{TV}}$ is the total variation norm and $\mathbf{1}_n/n$ is the uniform distribution on $[n]$.

This abstraction captures the auxiliary-observation mechanism in gossip pairwise optimization: at time $t$, node $i$ can only pair $\mathbf{x}_i$ with a single auxiliary sample (encoded by $\boldsymbol{\xi}_i(t)$), whose distribution becomes approximately uniform only after sufficient mixing on the graph.

We make the additional assumption that one cannot access $\nabla_\theta \tilde{F}_i(\cdot, \mathbf{1}_n/n)$ directly. Instead, at iteration $t \geq 1$, one can compute $\nabla_\theta \tilde{F}_i(\cdot, \boldsymbol{\xi}_i(t))$, yielding the noisy update

$$\begin{cases} \mathbf{z}_i(t+1) & = & \mathbf{z}_i(t) + \nabla_\theta \tilde{F}_i(\theta(t), \boldsymbol{\xi}_i(t)) \\ \boldsymbol{\theta}_i(t+1) & = & \pi_t(\mathbf{z}_i(t+1)) \end{cases} . \tag{10}$$

For any $\epsilon > 0$, define the mixing time $\tau_i(\epsilon)$:

$$\tau_i(\epsilon) := \sup_{t \geq 0} \inf \left\{ s \geq 0 : \|P_i(t+s|t) - \mathbf{1}_n/n\|_{\mathrm{TV}} \leq \epsilon \right\} ,$$

where $P_i(t+s|t)$ is the conditional distribution of $\boldsymbol{\xi}_i(t+s)$ given the filtration $\mathcal{F}_t^{(i)} := \sigma(\boldsymbol{\xi}_i(1), \ldots, \boldsymbol{\xi}_i(t))$.

We are now ready to quantify explicitly the impact of the gradient bias and state the following refined rate bound.

**Theorem 2** (Pairwise ergodic dual averaging)**.** *Let $\mathcal{G} = ([n], \mathcal{E})$ be a connected and non-bipartite graph, and let $\boldsymbol{\theta}^\star \in \arg\min_{\boldsymbol{\theta} \in \Theta} F(\boldsymbol{\theta})$. Let $(\gamma(t))_{t \geq 1}$ be a non-increasing, non-negative sequence such that $\gamma(t) \propto t^\alpha$ for some $\alpha \in (-1, 0)$. For any $i \in [n]$ and any $t \geq 0$, let $\mathbf{z}_i(t) \in \mathbb{R}^d$ and $\bar{\boldsymbol{\theta}}_i(t) \in \mathbb{R}^d$ be generated according to Algorithm 2. Then, for any $\varepsilon > 0$,*

$$\mathbb{E}\left[F(\bar{\boldsymbol{\theta}}_i(T)) - F(\boldsymbol{\theta}^\star)\right] \leq C_1(T, \varepsilon) + C_2(T) + C_3(T, \varepsilon) ,$$

*where*

$$
\begin{cases}
C_1(T, \varepsilon) = \dfrac{D^2}{2T\gamma(T)} + \dfrac{\left(1 + 12\tau(\varepsilon)\right)L^2}{2T} \sum_{t=1}^{T} \gamma(t) \\[3ex]
C_2(T) = \dfrac{3L^2}{T\left(1 - \sqrt{1 - \lambda_{n-1}/|\mathcal{E}|}\right)} \sum_{t=1}^{T-1} \gamma(t) \\[3ex]
C_3(T, \varepsilon) = 2LD\left(\varepsilon + \dfrac{\tau(\varepsilon)}{T}\right)
\end{cases} ,
$$

*and*

$$
\tau(\varepsilon) = \frac{\log(\sqrt{n}/\varepsilon)}{\log(c(\mathcal{G}))} \qquad with \qquad c(\mathcal{G}) := \frac{\lambda_{n-1}}{|\mathcal{E}|}\left(1 - \frac{\lambda_{n-1}}{2|\mathcal{E}|}\right) .
$$

*Sketch of proof.* In the centralized setting, the convergence analysis relies on the regret decomposition of Duchi et al. (2012b): for any $0 \le \tau < T$,

$$
\sum_{t=1}^{T}(f(\boldsymbol{\theta}(t)) - f(\boldsymbol{\theta}^\star)) = \sum_{t=1}^{T-\tau} f(\boldsymbol{\theta}(t)) - f(\boldsymbol{\theta}^\star) - F(\boldsymbol{\theta}(t), \boldsymbol{\xi}(t+\tau)) + F(\boldsymbol{\theta}^\star, \boldsymbol{\xi}(t+\tau)) \tag{11}
$$

$$
+ \sum_{t=1}^{T-\tau}(F(\boldsymbol{\theta}(t), \boldsymbol{\xi}(t+\tau)) - F(\boldsymbol{\theta}(t+\tau), \boldsymbol{\xi}(t+\tau))) \tag{12}
$$

$$
+ \sum_{t=\tau+1}^{T}(F(\boldsymbol{\theta}(t), \boldsymbol{\xi}(t)) - F(\boldsymbol{\theta}^\star, \boldsymbol{\xi}(t))) \tag{13}
$$

$$
+ \sum_{t=T-\tau+1}^{T}(f(\boldsymbol{\theta}(t)) - f(\boldsymbol{\theta}^\star)) . \tag{14}
$$

The key point is that each term isolates variations of either $\boldsymbol{\theta}$ or $\boldsymbol{\xi}$ but never both simultaneously, which allows one to handle the dependence induced by the evolving sampling distribution. Including the network disagreement contribution in the noiseless term (13) yields the stated bound; see Appendix D.2 for details. $\qquad\square$

Theorem 2 makes the bias mechanism explicit through $\tau(\varepsilon)$ and the residual term $C_3(T, \varepsilon)$. In gossip settings, the auxiliary process mixes geometrically, hence $\tau(\varepsilon)$ grows only logarithmically in $1/\varepsilon$.

**Corollary 1** (Corrected and refined rate for $\gamma(t) = a/\sqrt{t}$)**.** *Let $\mathcal{G} = ([n], \mathcal{E})$ be a connected and non-bipartite graph, and let $\boldsymbol{\theta}^\star \in \arg\min_{\boldsymbol{\theta} \in \Theta} F(\boldsymbol{\theta})$. Let $a > 0$ and $\gamma(t) = a/\sqrt{t}$. For any $i \in [n]$, let $\bar{\boldsymbol{\theta}}_i(T)$ be generated by Algorithm 2. Then, for any $T \ge 1$,*

$$
\mathbb{E}\left[F(\bar{\theta}_i(T)) - F(\theta^\star)\right] \le \frac{1}{\sqrt{T}}\left[\frac{\|\theta_0 - \theta^\star\|^2}{2a} + aL^2 + \frac{6aL^2}{1 - \sqrt{1 - \lambda_{n-1}/|\mathcal{E}|}} + \frac{12aL^2}{|\log(c(\mathcal{G}))|}\log(T)\right]
$$
$$
+ \frac{2L\|\theta_0 - \theta^\star\|}{T}\left(1 + |\log(c(\mathcal{G}))|\log(T)\right) ,
$$

*which corresponds to the choice $\varepsilon_T = 1/T$ in Theorem 2.*

In particular, the term appearing in the second line of the bound above is precisely the explicit remainder term that was previously hidden in $o(1/\sqrt{T})$ when $\varepsilon_T$ was not specified. Choosing $a = \|\theta_0 - \theta^\star\|/L$ minimizes the leading constant in the $1/\sqrt{T}$ term, yielding the equivalent (but more interpretable) form:

$$
\mathbb{E}\left[F(\bar{\theta}_i(T)) - F(\theta^\star)\right] \le \frac{L\|\theta_0 - \theta^\star\|}{\sqrt{T}}\left(\frac{3}{2} + \frac{6}{1 - \sqrt{1 - \lambda_{n-1}/|\mathcal{E}|}} + \frac{12\log(T)}{|\log(c(\mathcal{G}))|}\right)
$$
$$
+ \frac{2L\|\theta_0 - \theta^\star\|}{T}\left(1 + |\log(c(\mathcal{G}))|\log(T)\right) .
$$

**Remark.** *The asynchronous setting requires time estimators, inducing additional variance and a slower overall convergence rate. For completeness, the asynchronous implementation and associated rate are provided in Appendix D.2.3.*

### 4.3 A Lower Bound Result for Pairwise Decentralized Optimization

We derive a lower bound related to the synchronous decentralized optimization problem considered in Section 4.1 by extending the argument of Scaman et al. (2018) to the pairwise case. The lower bound is closely related to the distance induced by graph $\mathcal{G}$, which is why we introduce it here. Given a connected graph $\mathcal{G} = ([n], \mathcal{E})$, the distance $d_{\mathcal{G}}(i, j)$ between nodes $i$ and $j$ is the minimum length of a path between them.

**Theorem 3** (Lower bound). *Let $n > 2$ and let $\mathcal{G} = ([n], \mathcal{E})$ be a connected graph. Let $(i, j) \in [n]^2$ be two nodes of $\mathcal{G}$ that are at maximum distance, that is $d_{\mathcal{G}}(i, j) \in \arg\max_{(k,l)} d_{\mathcal{G}}(k, l)$ and let*

$$\tilde{\Delta} = \frac{1}{n-2} \sum_{k \notin \{i,j\}} d_{\mathcal{G}}(i, k) + d_{\mathcal{G}}(k, j) \ .$$

*There exist $n^2$ functions $f_{ij} : \mathbb{R}^d \to \mathbb{R}$ such that $F = n^{-2} \sum_{i,j} f_{ij}$ is $L$-Lipschitz and for any $k \in [n]$, any $R > 0$, any $t < (d-2)\min\{\bar{\Delta}, 1\}$ and any synchronous black-box procedure $(\boldsymbol{\theta}_k(t))$,*

$$F(\boldsymbol{\theta}_k(t)) - \min_{\|\boldsymbol{\theta}\| \leq R} F(\boldsymbol{\theta}) \geq \frac{RL}{36} \sqrt{\frac{1}{\left(1 + t/\tilde{\Delta}\right)^2} + \frac{1}{1+t}} \ .$$

Theorem 3 extends the construction of Scaman et al. (2018) from separable objectives to pairwise objectives. The proof is provided in Appendix D.3. The lower bound for decentralized pairwise optimization is similar to known lower bounds for separable objectives; the main difference lies in the appearance of $\tilde{\Delta}$. For separable objectives, the lower bound typically depends on the diameter (distance between two farthest nodes). In the pairwise case, information must effectively propagate from $i$ to $j$ through intermediate nodes $k$, which leads to the averaged two-hop distance quantity $\tilde{\Delta}$.

## 5 Numerical Experiments

In this section, we report numerical experiments illustrating the behavior of the proposed decentralized algorithms for the maximization of the Area Under the ROC Curve (AUC). This criterion is a standard performance measure in binary classification and ranking problems, see Section 2.2. Our experiments aim at assessing both the convergence properties of the decentralized pairwise optimization scheme and the practical impact of the bias induced by local information exchanges. In particular, we observe that the algorithms converge reliably and that the bias term rapidly becomes negligible as iterations progress.

Given a collection of feature vectors $\mathbf{x}_1, \ldots, \mathbf{x}_n \in \mathbb{R}^d$ with associated binary labels $\ell_1, \ldots, \ell_n \in \{-1, 1\}$, we consider linear scoring functions of the form $\mathbf{x} \mapsto \mathbf{x}^\top \boldsymbol{\theta}$, parameterized by $\boldsymbol{\theta} \in \mathbb{R}^d$. The AUC criterion measures the proportion of correctly ordered positive–negative pairs and can be written as

$$\mathrm{AUC}(\boldsymbol{\theta}) = \frac{\sum_{1 \leq i,j \leq n} \mathbb{1}_{\{\ell_i > \ell_j\}} \mathbb{1}_{\{\mathbf{x}_i^\top \boldsymbol{\theta} > \mathbf{x}_j^\top \boldsymbol{\theta}\}}}{\sum_{1 \leq i,j \leq n} \mathbb{1}_{\{\ell_i > \ell_j\}}} \ .$$

This quantity corresponds to the probability that the scoring function assigns a higher value to a positively labeled observation than to a negatively labeled one. Since the resulting optimization problem is non-smooth and non-convex, it is common to replace it with a convex surrogate. In our experiments, we rely on the logistic pairwise loss and consider the following empirical risk:

$$R_n(\boldsymbol{\theta}) = \frac{1}{n^2} \sum_{1 \leq i,j \leq n} \mathbb{1}_{\{\ell_i > \ell_j\}} \log\left(1 + \exp\left((\mathbf{x}_j - \mathbf{x}_i)^\top \boldsymbol{\theta}\right)\right) .$$

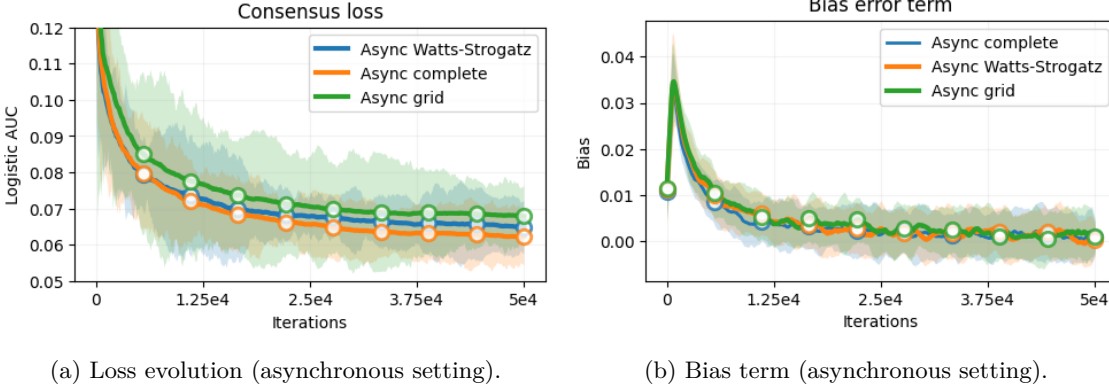

(a) Loss evolution (asynchronous setting).     (b) Bias term (asynchronous setting).

Figure 1: Logistic AUC. Solid lines are averages and filled area are standard deviations.

No explicit regularization is added in these experiments. All experiments are conducted on the Breast Cancer Wisconsin dataset,[2] which contains $n = 699$ samples in dimension $d = 11$. In the decentralized setting considered here, each node of the network stores exactly one observation.

**Network topologies.** To investigate the influence of communication constraints, we consider three network structures with markedly different connectivity properties:

- *Complete graph.* Every pair of nodes can communicate directly. This topology represents an idealized scenario, as information mixing is immediate and gradient estimates are expected to exhibit minimal bias. For a fixed number of nodes, the complete graph maximizes the spectral gap, see Chung (1997, Ch. 1). In our setting with $n = 699$, we obtain $\lambda_{n-1}/|\mathcal{E}| = 2.86 \cdot 10^{-3}$.

- *2D grid.* Each node has exactly four neighbors, leading to poor connectivity and a large graph diameter. This topology represents a challenging regime for decentralized optimization. In this case, the normalized spectral gap is $\lambda_{n-1}/|\mathcal{E}| = 4.30 \cdot 10^{-5}$.

- *Watts–Strogatz graph.* Following **?**, this random graph model interpolates between regular lattices and well-connected networks. It is controlled by the average degree $k$ and a rewiring probability $p \in (0,1)$. We choose $k = 5$ and $p = 0.3$, yielding intermediate connectivity properties with $\lambda_{n-1}/|\mathcal{E}| = 1.82 \cdot 10^{-4}$.

**Experimental setup.** For each network topology, we initialize all local parameters $\boldsymbol{\theta}_i$ to zero and run Algorithms 2 and 7 for 50 independent trials. The stepsize sequence is chosen as $\gamma(t) = 10^{-3}/\sqrt{t}$.

Figure 1a reports the evolution of the objective value in the asynchronous regime, together with the standard deviation of the local objectives across nodes. As expected, better-connected networks exhibit faster convergence. In particular, the complete and Watts–Strogatz graphs outperform the grid topology. In addition, the dispersion of the local estimates decreases as network connectivity improves.

Finally, Figure 1b illustrates the behavior of the bias term $\widehat{\boldsymbol{\epsilon}}(t)^\top \widehat{\boldsymbol{\omega}}(t)$. In all considered networks, this quantity rapidly converges to zero and remains several orders of magnitude smaller than the objective value throughout the optimization process. This empirical observation supports the theoretical analysis and explains the good practical performance of the proposed decentralized algorithm.

## 6 Concluding Remarks and Future Work

In this final section, we summarize the main contributions of our study and outline promising directions for extending our results. Beyond the theoretical guarantees established for decentralized estimation and

---

[2]https://archive.ics.uci.edu/ml/datasets/Breast+Cancer+Wisconsin+(Original)

optimization with pairwise objectives, we discuss potential adaptations of our methods to broader and more demanding learning scenarios.

## 6.1 Multiple Points per Node

For ease of presentation, we have assumed throughout the paper that each node $i$ holds a single data point $\mathbf{x}_i$. We now consider the case where each node holds the same number of points $k \geq 2$. First, it is easy to see that our results still hold if nodes swap their entire set of $k$ points (essentially viewing the set of $k$ points as a single one). However, depending on the network bandwidth, this solution may be undesirable. We thus propose another strategy where only two data points are exchanged at each iteration, as in the algorithms proposed in the main text. The idea is to view each "physical" node $i \in [n]$ as a set of $k$ "virtual" nodes, each holding a single observation. These $k$ nodes are all connected to each other as well as to the neighbors of $i$ in the initial graph $G$ and their virtual nodes. Formally, this new graph $G^{\otimes} = ([n]^{\otimes}, E^{\otimes})$ is given by $G \times \mathbb{K}_k$, the tensor product between $G$ and the $k$-node complete graph $\mathbb{K}_k$. It is easy to see that $|[n]^{\otimes}| = kn$ and $|E^{\otimes}| = k^2|E|$. We can then run our algorithms on $G^{\otimes}$ (each physical node $i \in [n]$ simulating the behavior of its corresponding $k$ virtual nodes) and the convergence results hold, replacing $1 - \lambda_{n-1}^G$ by $1 - \lambda_{n-1}^{G^{\otimes}}$ in the bounds. The following result gives the relationship between both gaps.

**Proposition 2.** *Let $G = ([n], E)$ be a connected, non-bipartite and non-complete graph. Let $k \geq 2$ and let $G^{\otimes}$ be the tensor product graph of $G$ and $\mathbb{K}_k$. We have that*

$$\lambda_{n-1}^{G^{\otimes}} = k\lambda_{n-1} \ ,$$

*where $\lambda_{n-1}$ and $\lambda_{kn-1}^{G^{\otimes}}$ are the second smallest eigenvalues of $\mathbf{L}^G$ and $\mathbf{L}^{G^{\otimes}}$ respectively.*

The proof is stated in the Appendix. Proposition 2 shows that the network-dependent term in our convergence bounds is only affected by a factor $k$. Furthermore, note that iterations involving two virtual nodes corresponding to the same physical node will not require actual network communication, which somewhat attenuates this effect in practice.

## 6.2 Differential Privacy

In many decentralized applications, privacy is a major concern: raw data is sensitive and agents may not be willing to share it with other peers in the network. To accommodate such privacy constraints into our algorithms, which are based on exchanging data points across the network, one can rely on the local model of differential privacy (Dwork and Roth, 2014; Duchi et al., 2013). In this model, agents randomize their inputs locally before sharing them. The simplest example of locally differentially private protocol in the case of data in a discrete domain is randomized response, in which each agent either shares its true value (with some probability $p$) or a randomly selected one (with probability $1-p$). This gives rise to a natural trade-off between privacy and utility. Bell et al. (2020) recently proposed protocols to compute pairwise $U$-statistics under local differential privacy. Their first protocol based on randomized response can be easily combined with our decentralized algorithms since it allows to compute *unbiased* estimates of the pairwise function of interest. They also provide error bounds with respect to the non-private setting. The protocol can be extended to continuous data through quantization, see (Bell et al., 2020) for details.

## 6.3 Conclusion

In this work, we provided a comprehensive theoretical analysis of decentralized estimation and optimization of pairwise objectives, filling key gaps left by existing methods. We established new non-asymptotic guarantees for both the estimation of $U$-statistics and the convergence of decentralized pairwise optimization under gossip protocols, highlighting the role of network topology and gradient bias. These results lay the groundwork for extending our framework to more challenging scenarios. In particular, future research could integrate robustness and fairness constraints, which are often naturally formulated as U-statistics, thereby enabling principled decentralized learning under realistic and socially responsible requirements.

**Acknowledgments**

This research was supported by the PEPR IA Foundry and Hi!Paris ANR Cluster IA France 2030 grants. The authors thank the program for its funding and support. 2030' program, as part of the Hi! PARIS Cluster and the PEPR IA Foundry.

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

## A   Additional notation

We here introduce notation that will be useful for the various proofs. Let $G = ([n], E)$ be a connected graph. We denote as $\mathbf{L} \in \mathbb{R}^{n \times n}$ its Laplacian, that is, $\mathbf{L} := \mathbf{D} - \mathbf{A}$, and we denote as $\lambda_1 \geq \ldots \geq \lambda_n$ its eigenvalues, sorted in decreasing order. For $\alpha \geq 1$, let $\mathbf{W}_\alpha$ be the symmetric matrix in $\mathbb{R}^{n \times n}$ defined as:

$$\mathbf{W}_\alpha := \frac{1}{|E|} \sum_{(i,j) \in E} \left( \mathbf{I}_n - \frac{1}{\alpha}(\mathbf{e}_i - \mathbf{e}_j)(\mathbf{e}_i - \mathbf{e}_j)^\top \right) = \mathbf{I}_n - \frac{\mathbf{L}}{\alpha|E|} \ .$$

In particular, $\mathbf{W}_2$ is the transition corresponding to gossip averaging and $\mathbf{W}_1$ is the transition of auxiliary observations. We also denote as $\lambda_1(\alpha) \geq \ldots \geq \lambda_n(\alpha)$ the corresponding eigenvalues, sorted in decreasing order. We denote as $\mathbf{J}_n := n^{-1}\mathbf{1}_n\mathbf{1}_n^\top \in \mathbb{R}^{n \times n}$ and for any $\alpha \geq 1$,

$$\tilde{\mathbf{W}}_\alpha := \mathbf{W}_\alpha - \mathbf{J}_n \ .$$

For $t > s \geq 0$ and a sequence of matrices $\mathbf{M}(0), \ldots, \mathbf{M}(t) \in \mathbb{R}^{n \times n}$, we denote as $\mathbf{M}(t:s)$ the product

$$\mathbf{M}(t:s) := \mathbf{M}(t)\mathbf{M}(t-1)\ldots\mathbf{M}(s+1) \ .$$

Additionally, $\mathbf{M}(t:) := \mathbf{M}(t)\ldots\mathbf{M}(0)$ and we use the convention $\mathbf{M}(t:t) = \mathbf{I}_n$.

## B   $U$-statistic of degree $r$

In this section, we present the more general definition of a $U$-statistic (Van der Vaart, 2000).

**Definition 1** ($U$-statistic). *Denote $r \geq 1$ as the degree of a $U$-statistic and let $(x_1, \ldots, x_n) \in \mathcal{X}^n$ be a sample of $n \geq r$ points in a feature space $\mathcal{X}$. Let $h : \mathcal{X}^r \to \mathbb{R}$ be a measurable function that is permutation symmetric in its $r$ arguments. The $U$-statistic of degree $r$ is defined as:*

$$\hat{U}_{r,n}(h) = \frac{1}{\binom{n}{r}} \sum_{(i_1, \ldots, i_r) \in I_{r,n}} h(x_{i_1}, \ldots, x_{i_r}),$$

*where $I_{r,n}$ is the set of all unordered tuples of $r$ different integers from $\{1, \ldots, n\}$.*

For $r = 2$, we obtain the classical $U$-statistic:

$$\hat{U}_n(h) = \frac{2}{n(n-1)} \sum_{1 \leq i < j \leq n} h(x_i, x_j).$$

## C   Pairwise estimation

### C.1   Preliminary Results

Here, we state preliminary results on the matrices $\mathbf{W}_\alpha$ that will be useful for deriving convergence proofs and compare the algorithms. First, we characterize the eigenvalues of $\mathbf{W}_\alpha$ in terms of those of the graph Laplacian.

**Lemma 1.** *Let $G = ([n], E)$ be an undirected graph and let $(\lambda_i)_{1 \leq i \leq n}$ be the eigenvalues of its Laplacian $\mathbf{L}$, sorted in decreasing order. For any $\alpha \geq 1$, we denote as $(\lambda_i(\alpha))_{1 \leq i \leq n}$ the eigenvalues of $\mathbf{W}_\alpha$, sorted in decreasing order. Then, for any $1 \leq i \leq n$,*

$$\lambda_i(\alpha) = 1 - \frac{\lambda_{n-i+1}}{\alpha|E|} \ .$$

*Proof.* Let $\alpha \geq 1$ and let $\phi_i \in \mathbb{R}^n$ be an eigenvector of $\mathbf{L}$ corresponding to the eigenvalue $\lambda_i$, then we have:

$$\mathbf{W}_\alpha \phi_i = \left( \mathbf{I}_n - \frac{1}{\alpha|E|}\mathbf{L} \right) \phi_i = \left( 1 - \frac{1}{\alpha|E|}\lambda_i \right) \phi_i \ .$$

Thus, $\phi_i$ is also an eigenvector of $\mathbf{W}_\alpha$ for the eigenvalue $1 - \frac{1}{\alpha|E|}\lambda_i$ and the result holds. $\quad\square$

The following lemmata provide essential properties on $\mathbf{W}_\alpha$ eigenvalues.

**Lemma 2.** *Let $n > 0$ and let $G = ([n], E)$ be an undirected graph. If $G$ is connected and non-bipartite, then for any $\alpha \geq 1$, $\mathbf{W}_\alpha$ is primitive, i.e., there exists $k > 0$ such that $\mathbf{W}_\alpha^k > 0$.*

*Proof.* Let $\alpha \geq 1$. For every $(i, j) \in E$, $\mathbf{I}_n - \frac{1}{\alpha}(\mathbf{e}_i - \mathbf{e}_j)(\mathbf{e}_i - \mathbf{e}_j)^\top$ is nonnegative. Therefore $\mathbf{W}_\alpha$ is also nonnegative. For any $1 \leq k < l \leq n$, by definition of $\mathbf{W}_\alpha$, one has the following equivalence:

$$([\mathbf{A}]_{kl} > 0) \Leftrightarrow ([\mathbf{W}_\alpha]_{kl} > 0) \ .$$

By hypothesis, $G$ is connected. Therefore, for any pair of nodes $(k, l) \in [n]^2$ there exists an integer $s_{kl} > 0$ such that $[(\mathbf{A})^{s_{kl}}]_{kl} > 0$ so $\mathbf{W}_\alpha$ is irreducible. Also, $G$ is non-bipartite so similar reasoning can be used to show that $\mathbf{W}_\alpha$ is aperiodic.

By the Lattice Theorem (see (Brémaud, 1999, Th. 4.3, p.75)), for any $1 \leq k, l \leq n$ there exists an integer $m_{kl}$ such that, for any $m \geq m_{kl}$:

$$[\mathbf{W}_\alpha^m]_{kl} > 0 \ .$$

Finally, we can define $\bar{m} = \sup_{k,l} m_{kl}$ and observe that $\mathbf{W}_\alpha^{\bar{m}} > 0$. $\qquad\square$

**Lemma 3.** *Let $G = ([n], E)$ be a connected and non bipartite graph. Then for any $\alpha \geq 1$,*

$$1 = \lambda_1(\alpha) > \lambda_2(\alpha),$$

*where $\lambda_1(\alpha), \lambda_2(\alpha)$ are respectively the largest and the second largest eigenvalue of $\mathbf{W}_\alpha$.*

*Proof.* Let $\alpha \geq 1$. The matrix $\mathbf{W}_\alpha$ is bistochastic, so $\lambda_1(\alpha) = 1$. By Lemma 2, $\mathbf{W}_\alpha$ is primitive. Therefore, by the Perron-Frobenius Theorem (see (Brémaud, 1999, Th. 1.1, p.197)), we can conclude that $\lambda_1(\alpha) > \lambda_2(\alpha)$. $\qquad\square$

Now that we have a relation between the graph structure and the eigenvalues of $\mathbf{W}_\alpha$, we are ready to prove our main results.

## C.2 Decentralized pairwise estimation

### C.2.1 Proof of Theorem 1 (Synchronous Setting)

Let $\mathbf{h} := \text{vec}(\mathbf{H}) \in \mathbb{R}^{n^2}$. For $t \geq 1$, one has

$$\mathbf{z}(t) = \frac{t-1}{t} \cdot \mathbf{W}_2(t)\mathbf{z}(t-1) + \frac{1}{t}\mathbf{B}\left(\mathbf{I}_n \otimes \mathbf{W}_1(t)\right)\mathbf{h}$$

$$= \frac{1}{t}\sum_{s=1}^{t} \mathbf{W}_2(t\!:\!s)\mathbf{B}\left(\mathbf{I}_n \otimes \mathbf{W}_1(s\!:\!)\right)\mathbf{h} \ .$$

Using spectral theorem, we decompose the mixing matrices as $\mathbf{W}_\alpha = \tilde{\mathbf{W}}_\alpha + \mathbf{J}_n$. The above expression can thus be expanded as follows:

$$\mathbf{z}(t) = \frac{1}{t}\sum_{s=1}^{t}\left(\tilde{\mathbf{W}}_2(t\!:\!s) + \mathbf{J}_n\right)\mathbf{B}\left(\mathbf{I}_n \otimes \left(\tilde{\mathbf{W}}_1(s\!:\!) + \mathbf{J}_n\right)\right)\mathbf{h}$$

$$= \hat{U}_n\mathbf{1}_n + \frac{1}{t}\sum_{s=1}^{t}\tilde{\mathbf{W}}_2(t\!:\!s)\mathbf{B}\left(\mathbf{I}_n \otimes \tilde{\mathbf{W}}_1(s\!:\!)\right)\tilde{\mathbf{h}} + \mathbf{J}_n\mathbf{B}\left(\mathbf{I}_n \otimes \tilde{\mathbf{W}}_1(s\!:\!)\right)\tilde{\mathbf{h}} + \tilde{\mathbf{W}}_2(t\!:\!s)\mathbf{B}\bar{\mathbf{h}} \ ,$$

where $\tilde{\mathbf{h}} := \mathbf{h} - \mathbf{I}_n \otimes \mathbf{J}_n \mathbf{h}$. Taking the squared norm and using the orthogonality of eigenspaces yields

$$\left\| \mathbf{z}(t) - \hat{U}_n \mathbf{1}_n \right\|^2 = \left\| \frac{1}{t} \sum_{s=1}^{t} \mathbf{W}_2(t{:}s) \mathbf{B} \left( \mathbf{I}_n \otimes \tilde{\mathbf{W}}_1(s{:}) \right) \tilde{\mathbf{h}} + \mathbf{J}_n \mathbf{B} \left( \mathbf{I}_n \otimes \tilde{\mathbf{W}}_1(s{:}) \right) \tilde{\mathbf{h}} \right\|^2$$

$$= \left\| \frac{1}{t} \sum_{s=1}^{t} \tilde{\mathbf{W}}_2(t{:}s) \mathbf{B} \left( \mathbf{I}_n \otimes \mathbf{W}_1(s{:}) \right) \mathbf{h} \right\|^2 \tag{A}$$

$$+ \left\| \frac{1}{t} \sum_{s=1}^{t} \mathbf{J}_n \mathbf{B} \left( \mathbf{I}_n \otimes \tilde{\mathbf{W}}_1(s{:}) \right) \tilde{\mathbf{h}} \right\|^2 \ . \tag{B}$$

Let us start by analyzing the first term (A); one has

$$(\text{A}) = \frac{1}{t^2} \sum_{s=1}^{t} \sum_{r=1}^{t} \mathbf{h}^\top \left( \mathbf{I}_n \otimes \mathbf{W}_1(r{:}) \right)^\top \mathbf{B}^\top \tilde{\mathbf{W}}_2(t{:}r)^\top \tilde{\mathbf{W}}_2(t{:}s) \mathbf{B} \left( \mathbf{I}_n \otimes \mathbf{W}_1(s{:}) \right) \mathbf{h}$$

$$= \frac{1}{t^2} \sum_{s=1}^{t} \mathbf{h}^\top \left( \mathbf{I}_n \otimes \mathbf{W}_1(s{:}) \right)^\top \mathbf{B}^\top \tilde{\mathbf{W}}_2(t{:}s)^\top \tilde{\mathbf{W}}_2(t{:}s) \mathbf{B} \left( \mathbf{I}_n \otimes \mathbf{W}_1(s{:}) \right) \mathbf{h} \tag{A.1}$$

$$+ \frac{2}{t^2} \sum_{s=1}^{t-1} \sum_{r=s+1}^{t} \mathbf{h}^\top \left( \mathbf{I}_n \otimes \mathbf{W}_1(r{:}) \right)^\top \mathbf{B}^\top \tilde{\mathbf{W}}_2(t{:}r)^\top \tilde{\mathbf{W}}_2(t{:}s) \mathbf{B} \left( \mathbf{I}_n \otimes \mathbf{W}_1(s{:}) \right) \mathbf{h} \ . \tag{A.2}$$

For any $t \geq s > 0$ and any $\mathbf{x} \in \mathbb{R}^n$, one has

$$\mathbb{E}\left[ \mathbf{x}^\top \tilde{\mathbf{W}}_2(t{:}s)^\top \tilde{\mathbf{W}}_2(t{:}s) \mathbf{x} \middle| \mathcal{F}_{t-1} \right] = \mathbf{x}^\top \tilde{\mathbf{W}}_2(t-1{:}s)^\top \mathbb{E}\left[ \tilde{\mathbf{W}}_2(t)^2 \right] \tilde{\mathbf{W}}_2(t{:}s) \mathbf{x}$$

$$\leq \lambda_2(2) \mathbf{x}^\top \tilde{\mathbf{W}}_2(t-1{:}s)^\top \tilde{\mathbf{W}}_2(t-1{:}s) \mathbf{x} \ . \tag{15}$$

Using (15) on (A.1) yields the following result

$$(\text{A.1}) \leq \frac{1}{t^2} \sum_{s=1}^{t} \lambda_2(2)^{t-s} \left\| \mathbf{B} \left( \mathbf{I}_n \otimes \mathbf{W}_1(s{:}) \right) \mathbf{h} \right\|^2 \leq \frac{1}{t^2} \cdot \frac{\|\mathbf{h}\|}{1 - \lambda_2(2)} \ .$$

The second part (A.2) can be bounded in a similar fashion, using the above reasoning up to index $r$:

$$(\text{A.2}) \leq \frac{2}{t^2} \sum_{s=1}^{t-1} \sum_{r=s+1}^{t} \lambda_2(2)^{t-r} \mathbf{h}^\top \left( \mathbf{I}_n \otimes \mathbf{W}_1(r{:}) \right)^\top \mathbf{B}^\top \tilde{\mathbf{W}}_2(r{:}s) \mathbf{B} \left( \mathbf{I}_n \otimes \mathbf{W}_1(s{:}) \right) \mathbf{h}$$

$$\leq \frac{2 \|\mathbf{h}\|^2}{t^2} \sum_{s=1}^{t-1} \sum_{r=s+1}^{t} \lambda_2(2)^{t-r}$$

$$\leq \frac{2 \|\mathbf{h}\|^2}{t \left( 1 - \lambda_2(2) \right)} \ .$$

This concludes the analysis of the first term:

$$(\text{A}) \leq \frac{\|\mathbf{h}\|^2}{1 - \lambda_2(2)} \left( \frac{1}{t^2} + \frac{2}{t} \right) \ .$$

Let us now focus on the second term. It can be split in a similar way:

$$(\text{B}) = \frac{1}{t^2} \sum_{s=1}^{t} \sum_{r=1}^{t} \tilde{\mathbf{h}}^\top \left( \mathbf{I}_n \otimes \tilde{\mathbf{W}}_1(s{:}) \right)^\top \mathbf{B}^\top \mathbf{J}_n^\top \mathbf{J}_n \mathbf{B} \left( \mathbf{I}_n \otimes \tilde{\mathbf{W}}_1(s{:}) \right) \tilde{\mathbf{h}}$$

$$= \frac{1}{t^2} \sum_{s=1}^{t} \tilde{\mathbf{h}}^\top \left( \mathbf{I}_n \otimes \tilde{\mathbf{W}}_1(s{:}) \right)^\top \mathbf{B}^\top \mathbf{J}_n \mathbf{B} \left( \mathbf{I}_n \otimes \tilde{\mathbf{W}}_1(s{:}) \right) \tilde{\mathbf{h}} \tag{B.1}$$

$$+ \frac{2}{t^2} \sum_{s=1}^{t-1} \sum_{r=s+1}^{t} \tilde{\mathbf{h}}^\top \left( \mathbf{I}_n \otimes \tilde{\mathbf{W}}_1(r{:}) \right)^\top \mathbf{B}^\top \mathbf{J}_n \mathbf{B} \left( \mathbf{I}_n \otimes \tilde{\mathbf{W}}_1(s{:}) \right) \tilde{\mathbf{h}} \ . \tag{B.2}$$

---

**Algorithm 3** U1-GOSSIP algorithm for computing (6)

---

**Require:** Each node $k$ holds observation $\mathbf{x}_k$
1: $\mathbf{y}_k \leftarrow \mathbf{x}_k$   (Each node initializes its auxiliary observation)
2: $z_k \leftarrow 0$   (Each node initializes its estimator)
3: **for** $t = 1, 2, \dots$ **do**
4:     Draw $(i, j)$ uniformly at random from $E$
5:     Nodes $i$ and $j$ swap their auxiliary observations: $\mathbf{y}_i \leftrightarrow \mathbf{y}_j$
6:     **for** $k = 1, \dots, n$ **do**
7:         $z_k \leftarrow \frac{t-1}{t} z_k + \frac{1}{t} h(\mathbf{x}_k, \mathbf{y}_k)$
8:     **end for**
9: **end for**
10: **return** Each node $k$ has $z_k$

---

The first term (B.1) can be simply bounded by

$$(\text{B.1}) \leq \frac{\left\|\tilde{\mathbf{h}}\right\|^2}{t} \ .$$

Regarding the second term, one has

$$(\text{B.2}) \leq \frac{2}{t^2} \sum_{s=1}^{t-1} \sum_{r=s+1}^{t} \lambda_2(1)^{r-s} \left\|\tilde{\mathbf{h}}\right\|^2$$

$$\leq \frac{2 \left\|\tilde{\mathbf{h}}\right\|^2}{t \left(1 - \lambda_2(1)\right)} \ .$$

Combining our bounds on (A) and (B) yields

$$\mathbb{E} \left\| \mathbf{z}(t) - \hat{U}_n \mathbf{1}_n \right\|^2 \leq \frac{1}{t} \left( \left(1 + \frac{2|E|}{\lambda_{n-1}}\right) \left\|\tilde{\mathbf{h}}\right\|^2 + \frac{2|E|}{\lambda_{n-1}} \left(2 + \frac{1}{t}\right) \|\mathbf{h}\|^2 \right) \ .$$

The result holds by using the Cauchy-Schwarz inequality on the above result.

### C.3   U1-GOSSIP **algorithm**

The goal of the U1-GOSSIP algorithm is for each node to compute an estimate of its respective partial U-statistic. This process is summarized in Algorithm 3. Using reasoning similar to that for GOSTA, one can derive a convergence rate for the expected estimates, as stated in the following theorem.

**Theorem 4.** *Let us assume that $G = (V, E)$ is connected and non bipartite. Then, for $\mathbf{z}(t) = (z_1(t), \dots, z_n(t))^\top$ defined in Algorithm 3, we have that for all $k \in [n]$ and any $t > 0$,*

$$\left| \mathbb{E}[z_k(t)] - \hat{U}_n^{(k)} \right| \leq \frac{|E| \|\mathbf{H}\mathbf{e}_k\|}{2\lambda_{n-1} t} \ ,$$

*where $\lambda_{n-1}$ is the second smallest eigenvalue of the graph Laplacian $\mathbf{L}$.*

### C.4   U2-GOSSIP **algorithm**

U2-GOSSIP Pelckmans and Suykens (2009) is an alternative to GOSTA for computing $U$-statistics. In this algorithm, each node stores two auxiliary observations that are propagated using independent random walks. These two auxiliary observations will be used for estimating the $U$-statistic—see Algorithm 4 for details.

We can now state a convergence result for Algorithm 4.

**Theorem 5.** *Let us assume that $G$ is connected and non-bipartite. Then, for $\mathbf{z}(t)$ defined in Algorithm 4, we have that for all $k \in [n]$:*

$$\lim_{t \to +\infty} \mathbb{E}[z_k(t)] = \hat{U}_n(h) \ .$$

---

**Algorithm 4** U2-gossip Pelckmans and Suykens (2009)

---

**Require:** Each node $k$ holds observation $X_k$
1: $Y_k^{(1)} \leftarrow X_k, \quad Y_k^{(2)} \quad \leftarrow X_k, \quad Z_k \leftarrow 0$
2: **for** $t = 1, 2, \ldots$ **do**
3:     **for** $p = 1, \ldots, n$ **do**
4:         $Z_p \leftarrow \frac{t-1}{t} Z_p + \frac{1}{t} H(Y_p^{(1)}, Y_p^{(2)})$
5:     **end for**
6:     Draw $(i, j)$ uniformly at random from $E$
7:     Nodes $i$ and $j$ swap their first auxiliary observations: $Y_i^{(1)} \leftrightarrow Y_j^{(1)}$
8:     Draw $(k, l)$ uniformly at random from $E$
9:     Nodes $k$ and $l$ swap their second auxiliary observations: $Y_k^{(2)} \leftrightarrow Y_l^{(2)}$
10: **end for**

---

*Moreover, for any $t > 0$,*

$$\left\| \mathbb{E}[\mathbf{z}(t)] - \hat{U}_n(h)\mathbf{1}_n \right\| \le \frac{|E|}{t\lambda_{n-1}} \left( \frac{3 - \lambda_{n-1}/|E|}{2 - \lambda_{n-1}/|E|} \|\mathbf{H} - \overline{\mathbf{h}}\mathbf{1}_n^\top\|_{\mathrm{F}} + \|\overline{\mathbf{h}} - \hat{U}_n(h)\mathbf{1}_n\| \right) \ ,$$

*where $\lambda_{n-1}$ is the second smallest eigenvalue of $\mathbf{L}$.*

*Proof of Theorem 5.* Let $t > 0$, one can adapt GoSta analysis and write

$$\mathbf{z}(t) = \frac{1}{t} \sum_{s=1}^{t} \mathbf{W}_1'(s\colon)\mathbf{B}_n \left( \mathbf{I}_n \otimes \mathbf{W}_1(s\colon) \right) \mathbf{h} \ ,$$

where for any $1 \le s \le t$, $\mathbf{W}_1'(s)$ is an independent copy of $\mathbf{W}_1(s)$. Therefore, one has

$$\mathbb{E}[\mathbf{z}(t)] = \hat{U}_n(h) + \frac{1}{t} \sum_{s=1}^{t} \mathbf{J}_n \mathbf{B}_n \left( \mathbf{I}_n \otimes \tilde{\mathbf{W}}_1^s \right) \tilde{\mathbf{h}} + \tilde{\mathbf{W}}_1^s \overline{\mathbf{h}} + \tilde{\mathbf{W}}_1^s \mathbf{B}_n \left( \mathbf{I}_n \otimes \tilde{\mathbf{W}}_1^s \right) \tilde{\mathbf{h}} \ .$$

Using the eigenvalues properties of $\mathbf{W}_1$, one can write

$$\left\| \mathbb{E}[\mathbf{z}(t)] - \hat{U}_n(h)\mathbf{1}_n \right\| \le \frac{1}{t} \sum_{s=1}^{t} \lambda_2(1)^s \|\tilde{\mathbf{h}}\| + \lambda_2(1)^s \|\overline{\mathbf{h}}\| + \lambda_2(1)^{2s} \|\tilde{\mathbf{h}}\| \ .$$

Rearranging the terms yields

$$\left\| \mathbb{E}[\mathbf{z}(t)] - \hat{U}_n(h)\mathbf{1}_n \right\| \le \frac{2 + \lambda_2(1)}{1 - \lambda_2(1)^2} \|\tilde{\mathbf{h}}\| + \frac{1}{1 - \lambda_2(1)} \|\tilde{\mathbf{h}}\| \ ,$$

and the result holds.

$$\square$$

# D  Dual averaging

In this section, we focus on the dual averaging algorithm. First, Section D.1 introduces some centralized approaches for dual averaging and the key theoretical guarantees associated. Then, Section D.2 develops the proofs for such guarantees, alongside the proofs of the convergence rates presented in Section 4.

## D.1  Centralized dual averaging

### D.1.1  Deterministic Setting

In this section, we review the dual averaging optimization algorithm (Nesterov, 2009; Xiao, 2010) to solve Problem (7) in the centralized setting (where all data lie on the same machine). To explain the main idea

behind dual averaging, let us first consider the iterations of Stochastic Gradient Descent (SGD), assuming $\psi \equiv 0$ for simplicity:

$$\theta(t+1) = \theta(t) - \gamma(t)\mathbf{g}(t) \ ,$$

where $\mathbb{E}[\mathbf{g}(t)|\theta(t)] = \nabla f(\theta(t))$, and $(\gamma(t))_{t \geq 0}$ is a non-negative and non-increasing step size sequence. This update rule can be rewritten equivalently as follows:

$$\theta(t+1) = \underset{\theta \in \mathbb{R}^d}{\arg\min} \left\{ f(\theta(t)) + (\theta - \theta(t))^\top \mathbf{g}(t) + \frac{\|\theta - \theta(t)\|^2}{2\gamma(t)} \right\} \ ,$$

meaning that $\theta(t+1)$ is the minimizer of some quadratic approximation of $f$ around $\theta(t)$. Recursively and assuming that $\theta(0) = 0$, one can obtain:

$$\theta(t+1) = \underset{\theta \in \mathbb{R}^d}{\arg\min} \left\{ \theta^\top \left( \sum_{s=0}^{t} \gamma(s)\mathbf{g}(s) \right) + \frac{\|\theta\|}{2} \right\} \ . \tag{16}$$

For SGD to converge to an optimal solution, the step size sequence must satisfy $\lim_{t \to +\infty} \gamma(t) = 0$ and $\sum_{t=0}^{\infty} \gamma(t) = \infty$. As noticed by Nesterov (2009), an undesirable consequence is that new gradient estimates are given smaller weights than old ones in (16). Dual averaging aims at integrating all gradient estimates with the same weight.

Let $(\gamma(t))_{t \geq 0}$ be a positive and non-increasing step size sequence. The dual averaging algorithm maintains a sequence of primal iterates $(\theta(t))_{t > 0}$, and a sequence $(\mathbf{z}(t))_{t \geq 0}$ of dual variables which collects the sum of the unbiased gradient estimates seen up to time $t$. We initialize to $\theta(1) = \mathbf{z}(0) = 0$. At each step $t > 0$, we compute an unbiased estimate $\mathbf{g}(t)$ of $\nabla f(\theta(t))$. The most common choice is to take $\mathbf{g}(t) = \nabla f(\theta; \mathbf{x}_{I_t}, \mathbf{x}_{J_t})$ where $I_t$ and $J_t$ are drawn uniformly at random from $[n]$. We then set $\mathbf{z}(t+1) = \mathbf{z}(t) + \mathbf{g}(t)$ and generate the next iterate with the following rule:

$$\begin{cases} \theta(t+1) = \pi_t^\psi(\mathbf{z}(t+1)), \\ \pi_t^\psi(\mathbf{z}) := \underset{\theta \in \mathbb{R}^d}{\arg\min} \left\{ -\mathbf{z}^\top \theta + \frac{\|\theta\|^2}{2\gamma(t)} + t\psi(\theta) \right\} \end{cases} \ .$$

This particular formulation was introduced in (Xiao, 2009; 2010), extending the method introduced by Nesterov (2009) in the specific case of indicator functions. In this work, we borrow the notation from Xiao (2010). When it is clear from the context, we will drop the dependence in $\psi$ and simply write $\pi_t(\mathbf{z}) = \pi_t^\psi(\mathbf{z})$.

Note that $\pi_t(\cdot)$ is related to the proximal operator of a function $\phi : \mathbb{R}^d \to \mathbb{R}$ defined by

$$\text{prox}_\phi(\mathbf{x}) = \underset{\mathbf{z} \in \mathbb{R}^d}{\arg\min} \left\{ \frac{\|\mathbf{z} - \mathbf{x}\|^2}{2} + \phi(\mathbf{x}) \right\} \ .$$

Indeed, one can write:

$$\pi_t(\mathbf{z}) = \text{prox}_{t\gamma(t)\psi}(\gamma(t)\mathbf{z}) \ .$$

For many functions $\psi$ of practical interest, $\pi_t(\cdot)$ has a closed form solution. For instance, when $\psi = \|\cdot\|^2$, $\pi_t(\cdot)$ corresponds to a simple scaling, and when $\psi = \|\cdot\|_1$ it is a soft-thresholding operator. If $\psi$ is the indicator function of a closed convex set $\mathcal{C}$, then $\pi_t(\cdot)$ is the projection operator onto $\mathcal{C}$.

The dual averaging method is summarized in Algorithm 5. In order to perform a theoretical analysis of this algorithm, we introduce the following function. Let us define, for $t \geq 0$

$$V_t(\mathbf{z}) := \underset{\theta \in \mathbb{R}^d}{\max} \left\{ \mathbf{z}^\top \theta - \frac{\|\theta\|^2}{2\gamma(t)} - t\psi(\theta) \right\} \ .$$

Remark that with the assumption that $\psi(0) = 0$, then $V_t(0) = 0$. Strong convexity in $\theta$ of the objective function, ensures that the solution of the optimization problem is unique. The following lemma links the function $V_t$ and the algorithm update and is a simple application of the results from (Xiao, 2009, Lemma 10):

---

**Algorithm 5** Centralized dual averaging

---

**Require:** Step size $(\gamma(t))_{t\geq 1} > 0$
1: $\theta \leftarrow 0, \quad \bar{\theta} \leftarrow 0, \quad \mathbf{z} \leftarrow 0$
2: **for** $t = 1, \ldots, T$ **do**
3: $\quad \mathbf{z} \leftarrow \mathbf{z} + \nabla f(\theta)$
4: $\quad \theta \leftarrow \pi_t(\mathbf{z})$
5: $\quad \bar{\theta} \leftarrow \left(1 - \frac{1}{t}\right)\bar{\theta} + \frac{1}{t}\theta$
6: **end forreturn** $\bar{\theta}$

---

**Lemma 4.** *For any* $\mathbf{z} \in \mathbb{R}^d$*, one has:*

$$\pi_t(\mathbf{z}) = \nabla V_t(\mathbf{z}) \ ,$$

*and the following statements hold true: for any* $\mathbf{z}_1, \mathbf{z}_2 \in \mathbb{R}^d$

$$\|\pi_t(\mathbf{z}_1) - \pi_t(\mathbf{z}_2)\| \leq \gamma(t)\|\mathbf{z}_1 - \mathbf{z}_2\| \ ,$$

*and for any* $\mathbf{g}, \mathbf{z} \in \mathbb{R}^d$*,*

$$V_t(\mathbf{z} + \mathbf{g}) \leq V_t(\mathbf{z}) + \mathbf{g}^\top \nabla V_t(\mathbf{z}) + \frac{\gamma(t)}{2}\|\mathbf{g}\|^2 \ . \tag{17}$$

Moreover, adapting (Xiao, 2009, Lemma 11) we can state:

**Lemma 5.** *For any* $t \geq 1$ *and any non-increasing sequence* $(\gamma(t))_{t\geq 1}$*, we have*

$$V_t\left(-\mathbf{z}(t+1)\right) + \psi(\theta(t+1)) \leq V_{t-1}\left(-\mathbf{z}(t+1)\right) \ .$$

We also need a last technical result that we will use several times in the following:

**Lemma 6.** *Let* $\theta(t) = \pi_t(\sum_{s=1}^{t-1}\mathbf{g}(s))$*, and let* $(\gamma(t))_{t\geq 1}$ *be a non-increasing and non-negative sequence sequence (with the convention* $\gamma(0) = 0$*), then for any* $\theta \in \mathbb{R}^d$*:*

$$\frac{1}{T}\sum_{t=1}^{T}\mathbf{g}(t)^\top(\theta(t) - \theta) + \frac{1}{T}\sum_{t=1}^{T}(\psi(\theta(t)) - \psi(\theta)) \leq \frac{1}{T}\sum_{t=1}^{T}\frac{\gamma(t-1)}{2}\|\mathbf{g}(t)\|^2 + \frac{\|\theta\|^2}{2T\gamma(T)} \ . \tag{18}$$

*Proof.* Using the definition of $V_T$, one can get the following upper bound:

$$\sum_{t=1}^{T}\left(\mathbf{g}(t)^\top\theta + \psi(\theta)\right) = \mathbf{z}(T+1)^\top\theta + T\psi(\theta)$$

$$= \mathbf{z}(T+1)^\top\theta + T\psi(\theta) + \frac{\|\theta\|^2}{2\gamma(T)} - \frac{\|\theta\|^2}{2\gamma(T)}$$

$$\leq \frac{\|\theta\|^2}{2\gamma(T)} + V_T(-\mathbf{z}(T+1)) \ . \tag{19}$$

Then, one can check that with (17) and Lemma 5 that, for any $1 \leq t \leq T$:

$$V_t(-\mathbf{z}(t+1)) + \psi(\theta(t+1)) \leq V_{t-1}(-\mathbf{z}(t+1)) = V_{t-1}(-\mathbf{z}(t) - \mathbf{g}(t))$$

$$\leq V_{t-1}(-\mathbf{z}(t)) - \mathbf{g}(t)^\top\theta(t) + \frac{\gamma(t-1)}{2}\|\mathbf{g}(t)\|^2 \ .$$

From the last display, the following holds:

$$\mathbf{g}(t)^\top\theta(t) + \psi(\theta(t+1)) \leq V_{t-1}(-\mathbf{z}(t)) - V_t(-\mathbf{z}(t+1)) + \frac{\gamma(t-1)}{2}\|\mathbf{g}(t)\|^2 \ .$$

Summing the former for $t = 1, \ldots, T$ yields

$$\sum_{t=1}^{T} \mathbf{g}(t)^{\top} \theta(t) + \psi(\theta(t+1)) \leq V_0(-\mathbf{z}_0) - V_T(-\mathbf{z}_T) + \sum_{t=1}^{T} \frac{\gamma(t-1)}{2} \|\mathbf{g}_t\|^2 \ .$$

Remark that $V_0(0) = 0$ and $\psi(\theta(1)) - \psi(\theta(T+1)) = -\psi(\theta(T+1)) \leq 0$, so the previous display can be reduced to:

$$\sum_{t=1}^{T} \mathbf{g}(t)^{\top} \theta(t) + \psi(\theta(t)) + V_T(-\mathbf{z}(T+1)) \leq \sum_{t=1}^{T} \frac{\gamma(t-1)}{2} \|\mathbf{g}(t)\|^2 \ . \tag{20}$$

Combining with (19), the lemma holds true. □

Bounding the error of the dual averaging is provided in the next theorem, where we remind that $R_n = f + \psi$:

**Theorem 6** (Dual averaging). *Let $(\gamma(t))_{t \geq 1}$ be a non increasing sequence. Let $(\mathbf{z}(t))_{t \geq 1}$, $(\theta(t))_{t \geq 1}$, $(\bar{\theta}(t))_{t \geq 1}$ and $(\mathbf{g}(t))_{t \geq 1}$ be generated according to Algorithm 5. Assume that the function $f$ is $L_f$-Lipschitz and let $\boldsymbol{\theta^*} \in \mathbb{R}^d$ be a minimizer of $R_n$, i.e., $\boldsymbol{\theta^*} \in \arg\min_{\theta' \in \mathbb{R}^d} R_n(\theta')$. Then for any $T \geq 2$, one has:*

$$R_n(\bar{\theta}(T)) - R_n(\theta^*) \leq \frac{\|\theta^*\|^2}{2T\gamma(T)} + \frac{L_f^2}{2T} \sum_{t=1}^{T-1} \gamma(t) \ . \tag{21}$$

*Moreover, with $D > 0$ such that $\|\theta^*\| \leq D$, and choosing $\gamma(t) = \frac{D}{L_f \sqrt{2t}}$ yields:*

$$R_n(\bar{\theta}(T)) - R_n(\theta^*) \leq \frac{\sqrt{2}DL_f}{\sqrt{T}} \ .$$

*Proof.* Let $T \geq 2$. Using the convexity of $f$ and $\psi$, we can get:

$$R_n(\bar{\theta}(T)) - R_n(\theta^*) \leq \frac{1}{T} \sum_{t=1}^{T} f(\theta(t)) - f(\theta^*) + \psi(\bar{\theta}) - \psi(\theta^*)$$

$$\leq \frac{1}{T} \sum_{t=1}^{T} \mathbf{g}(t)^{\top}(\theta(t) - \theta^*) + \frac{1}{T} \sum_{t=1}^{T} (\psi(\theta(t)) - \psi(\theta^*))$$

$$\leq \frac{1}{T} \sum_{t=1}^{T} \frac{\gamma(t-1)}{2} \|\mathbf{g}(t)\|^2 + \frac{\|\theta\|^2}{2T\gamma(T)} \ .$$

where the second inequality holds since $\mathbf{g}(t) = \nabla f(\theta(t))$, and the third one is from an application of Lemma 6 with the choice $\theta = \theta^*$. We can conclude the proof provided that $\|\mathbf{g}(t)\| \leq L_f$, which is true whenever $f$ is $L_f$-Lipschitz. □

### D.1.2 Stochastic Dual Averaging

Similarly to sub-gradient descent algorithms, one can adapt dual averaging algorithm to a stochastic setting; this was studied extensively by Xiao (2009). Instead of updating the dual variable $\mathbf{z}(t)$ with the (full) gradient of $f$ at $\theta(t)$, one now only requires the *expected* value of the update to be the gradient, that is:

$$\mathbf{z}(t+1) = \mathbf{z}(t) + \mathbf{g}(t) \ ,$$

with $\mathbb{E}[\mathbf{g}(t)|\theta(t)] = \nabla f(\theta(t))$. As in the gradient descent case, convergence results still hold in expectation, as stated in Theorem 7.

**Theorem 7** (Stochastic dual averaging). *Let* $(\gamma(t))_{t\geq1}$ *be a non increasing sequence. Let* $(\mathbf{z}(t))_{t\geq1}$, $(\theta(t))_{t\geq1}$ *and* $(\mathbf{g}(t))_{t\geq1}$ *be generated according to stochastic dual averaging rules. Assume that the function* $f$ *is* $L_f$-*Lipschitz and that* $\theta^* \in \arg\min_{\theta'\in\mathbb{R}^d} R_n(\theta')$, *then for any* $T \geq 2$, *one has:*

$$\mathbb{E}_T\left[R_n(\bar{\theta}(T)) - R_n(\theta^*)\right] \leq \frac{\|\theta^*\|^2}{2T\gamma(T)} + \frac{L_f^2}{2T}\sum_{t=1}^{T-1}\gamma(t) \ , \tag{22}$$

*where* $\mathbb{E}_T$ *is the expectation over all possible sequence* $(\mathbf{g}(t))_{1\leq t\leq T}$.

*Moreover, if one knows that* $D > 0$ *such that* $\|\theta^*\| \leq D$, *then for* $\gamma(t) = \frac{D}{L_f\sqrt{2t}}$, *one has:*

$$\mathbb{E}_T\left[R_n(\bar{\theta}(T)) - R_n(\theta^*)\right] \leq \frac{\sqrt{2}DL_f}{\sqrt{T}} \ .$$

*Proof.* One only has to prove that the convexity inequality in Lemma 6 holds in expectation. The rest of the proof can be directly adapted from Theorem 6.

Let $T \geq 2$; using the convexity of $f$, one obtains:

$$\mathbb{E}_T[f(\bar{\theta}(T)) - f(\theta^*)] \leq \frac{1}{T}\sum_{t=1}^{T}\mathbb{E}_T[f(\theta(t)) - f(\theta^*)] \ .$$

For any $0 < t \leq T$, $\mathbb{E}[\theta(t)|\mathbf{g}(0),\ldots,\mathbf{g}(t-1)] = \theta(t)$. Therefore, we have:

$$\mathbb{E}_T[f(\theta(t)) - f(\theta^*)] = \mathbb{E}_{t-1}[f(\theta(t)) - f(\theta^*)] \ .$$

The vector $\mathbb{E}_t[\mathbf{g}(t)|\theta(t)]$ is the gradient of $f$ at $\theta(t)$, we can then use $f$ convexity to write:

$$\mathbb{E}_{t-1}[f(\theta(t)) - f(\theta^*)] \leq \mathbb{E}_{t-1}\left[(\theta(t) - \theta^*)^\top\mathbb{E}_t[\mathbf{g}(t)|\theta(t)]\right] \ .$$

Using properties of conditional expectation, we obtain:

$$\mathbb{E}_{t-1}\left[(\theta(t) - \theta^*)^\top\mathbb{E}_t[\mathbf{g}(t)|\theta(t)]\right] = \mathbb{E}_{t-1}\left[\mathbb{E}_t[(\theta(t) - \theta^*)^\top\mathbf{g}(t)|\theta(t)]\right] = \mathbb{E}_t[(\theta(t) - \theta^*)^\top\mathbf{g}(t)] \ .$$

Finally, we can write:

$$\mathbb{E}_T[f(\bar{\theta}(T) - f(\theta^*)] \leq \frac{1}{T}\sum_{t=1}^{T}\mathbb{E}_t[(\theta(t) - \theta^*)^\top\mathbf{g}(t)] = \mathbb{E}_T\left[\frac{1}{T}\sum_{t=1}^{T}(\theta(t) - \theta^*)^\top\mathbf{g}(t)\right] \ ,$$

Therefore, the convexity inequality holds in expectation and one can adapt the proof of Theorem 6 to conclude. $\square$

### D.1.3 Distributed dual averaging

Let $\mathbf{x}_1,\ldots,\mathbf{x}_n \in \mathbb{R}^d$. We consider functions $f$ of the form $f := (1/n)\sum_{i=1}^{n}f(\cdot;\mathbf{x}_i)$. In addition, we now focus on a distributed setting, where each node $i \in [n]$ holds one observation $\mathbf{x}_i$ for simplicity. The distributed dual averaging algorithm for solving (7) was first introduced by Duchi et al. (2010a) and consists in the following: each node $i \in [n]$ stores its own primal and dual sequences $(\boldsymbol{\theta}_i(t), z_i(t))_{1\leq i\leq n}$. We denote as $\mathbf{Z}(t)$ the matrix of dual variables $\mathbf{Z}(t) = (\mathbf{z}_1(t),\ldots,\mathbf{z}_n(t))^\top$. At iteration $t+1$, a node $i$ will perform the following update:

$$\begin{cases} \mathbf{z}_i(t+1) &= \mathbf{g}_i(t) + \sum_{j=1}^{n}\mathbf{W}_{ij}\mathbf{z}_j(t) \\ \boldsymbol{\theta}_i(t+1) &= \pi_t(\mathbf{z}_i(t+1)) \ , \end{cases} \tag{23}$$

where $\mathbf{W}$ is a doubly stochastic matrix such that

$$(i,j) \notin E \Rightarrow \mathbf{W}_{ij} = 0 \ .$$

---

**Algorithm 6** Distributed dual averaging

---

**Require:** Step size $(\gamma(t))_{t \geq 1} > 0$, weight matrix $\mathbf{W}$
1: $\boldsymbol{\theta}_i(0) \leftarrow 0, \quad \mathbf{z}_i(0) \quad \leftarrow 0 \quad$ for each node $i$
2: **for** $t = 1, \ldots, T$ **do**
3: $\quad \mathbf{Z}(t+1) \leftarrow \mathbf{W}\mathbf{Z}(t) + \mathbf{G}(t)$
4: $\quad$ **for** $i = 1, \ldots, n$ **do**
5: $\quad\quad \boldsymbol{\theta}_i(t+1) \leftarrow \pi_t(\mathbf{z}_i(t+1))$
6: $\quad$ **end for**
7: **end for return** $(\bar{\theta}_i(T))_{1 \leq i \leq n}$

---

Update (23) only differs in the dual update: gradients are now added to an average of neighbors dual variables. Let us point out that the dual update can be reformulated as follows:

$$\mathbf{Z}(t+1) = \mathbf{G}(t) + \mathbf{W}\mathbf{Z}(t) \ ,$$

where $\mathbf{G}(t) = (\mathbf{g}_1(t), \ldots, \mathbf{g}_n(t))^\top$.

This is detailed in Algorithm 6. The main convergence results of this method can be stated as follows: given a sequence of step sizes $(\gamma(t))_{t \geq 0}$, for any $i \in [n]$ and any $T > 0$, one has

$$f(\bar{\boldsymbol{\theta}}_i(T)) - f^\star \leq \frac{\|\boldsymbol{\theta}^\star\|^2}{2T\gamma T} + \frac{L^2}{2T} \sum_{t=1}^{T-1} \gamma(t) + \frac{3L}{T} \max_{1 \leq j \leq n} \sum_{t=1}^{T} \gamma(t)\|\mathbf{z}_j(t) - \bar{\mathbf{z}}(t)\| \ , \tag{24}$$

where $\bar{\mathbf{z}}(t) := (1/n) \sum_{j=1}^n \mathbf{z}_j(t)$ is the average dual variable over the network. The first two terms are related to the dual averaging method and the last one depends on the graph topology. Establishing convergence of a distributed version of dual averaging thus relies on finding a communication scheme such that the rightmost quantity in (24) is decreasing.

### D.1.4 Ergodic dual averaging

The previous analysis is sufficient for providing convergence rate of a decentralized optimization when the objective is separable in the observations. For pairwise objectives however, an additional look at the dual averaging is needed. Indeed, one key insight to the method we describe later on is that biased estimates of gradients are computed in opposition to unbiased estimates of the stochastic dual averaging. However, estimate bias decreases exponentially fast, so it should not penalize heavily the convergence rate. We thus study the bias influence using an ergodic analysis.

**Problem setting** We define $F : \mathcal{X} \times \Delta_n$ as follows:

$$\tilde{F} : \begin{cases} \mathcal{X} \times \Delta_n & \to & \mathbb{R} \\ (\theta, \boldsymbol{\xi}) & \mapsto & \sum_{i=1}^n \xi_i f_i(\theta) \end{cases} ,$$

where $\Delta_n$ is the simplex in $\mathbb{R}^n$, *i.e.*,

$$\Delta_n = \{\boldsymbol{\xi} \in \mathbb{R}_+^n, \|\boldsymbol{\xi}\|_1 = 1\} \ .$$

Our goal is to solve the following optimization problem:

$$\min_{\theta \in \mathcal{X}} f(\theta) = F\left(\theta, \frac{\mathbf{1}_n}{n}\right) = \frac{1}{n} \sum_{i=1}^n f_i(\theta) \ . \tag{25}$$

Throughout this section, we make the assumption that there exists $D > 0$, such that if $\theta \in \mathcal{X}$ then $\|\theta\| \leq D$. Using the dual averaging approach, one aims at finding an algorithm for solving problem (25) with "noisy" information, in a way to be defined later. In the dual averaging method with true gradient information, variables are updated as follows:

$$\begin{cases} \mathbf{z}(t+1) & = & \mathbf{z}(t) + \nabla f(\theta(t)) \\ \theta(t+1) & = & \pi_t(\mathbf{z}(t+1)) \end{cases} . \tag{26}$$

As mentioned previously, we focus here on a noisy setting, similar to ergodic mirror descent introduced in Duchi et al. (2012b). Let $(\boldsymbol{\xi}(t))_{t\geq 0}$ be a sequence of—non necessarily independent—random variables over $\Delta_n$. For $t \geq 0$, we denote as $P(t)$ the distribution of $\boldsymbol{\xi}(t)$ and we assume that there exists $P^\infty$ such that $\lim_{t\to+\infty} \|P(t) - P^\infty\|_{TV} = 0$ and

$$\mathbb{E}_{P^\infty}[F(\cdot, \boldsymbol{\xi})] = f .$$

We make the additional assumption that one may not access the true value of $\nabla_\theta F(., \mathbf{1}_n/n)$. Instead, at iteration $t$, one can only compute an estimate $\nabla_\theta F(., \boldsymbol{\xi}(t))$. The iterative process described in (26) can then be reformulated:

$$\begin{cases} \mathbf{z}(t+1) &=& \mathbf{z}(t) + \nabla_\theta F(\theta(t), \boldsymbol{\xi}(t)) \\ \theta(t+1) &=& \pi_t(\mathbf{z}(t+1)) \end{cases} . \tag{27}$$

Recall that, for any $t > 0$, the mixing time of the distribution $P(t)$ towards its limit $P^\infty$ is defined as follows:

$$\tau(t, \cdot) : \epsilon \mapsto \inf \left\{ s \geq 0, \|P(t+s|t) - P^\infty\|_{\mathrm{TV}} \leq \epsilon \right\} ,$$

where $\| \cdot \|_{\mathrm{TV}}$ is the total variation distance between two distributions and $P(t+s|t)$ is the distribution of $\boldsymbol{\xi}(t+s)$ conditioned on the natural filtration $\mathcal{F}_t := \sigma(\boldsymbol{\xi}(1), \dots, \boldsymbol{\xi}(t))$.

**Convergence analysis**  As mentioned in Section 4.2, the convergence analysis of such a setting relies on one key observation, described in Duchi et al. (2012b): for any $0 \leq \tau < T$ and any $\theta^* \in \mathcal{X}$, the regret can be decomposed as follows

$$\sum_{t=1}^{T} (f(\theta(t)) - f(\theta^*)) = \sum_{t=1}^{T-\tau} (f(\theta(t)) - f(\theta^*) + F(\theta(t), \boldsymbol{\xi}(t+\tau)) - F(\theta^*, \boldsymbol{\xi}(t+\tau))) \tag{28}$$

$$+ \sum_{t=1}^{T-\tau} (F(\theta(t), \boldsymbol{\xi}(t+\tau)) - F(\theta(t+\tau), \boldsymbol{\xi}(t+\tau))) \tag{29}$$

$$+ \sum_{t=\tau+1}^{T} (F(\theta(t), \boldsymbol{\xi}(t)) - F(\theta^*, \boldsymbol{\xi}(t))) \tag{30}$$

$$+ \sum_{t=T-\tau+1}^{T} (f(\theta(t)) - f(\theta^*)) . \tag{31}$$

Following the reasoning of Duchi et al. (2012b), we will provide a bound on each term of the decomposition—some bounds actually being expected bounds (see Section D.2 for detailed proofs).

**Lemma 7** (Error after mixing). *Let $\theta$ be a $\mathcal{F}_t$-mesurable variable. Then for any $\theta^* \in \mathcal{X}$ and any $\tau > 0$, one has:*

$$\mathbb{E}[f(\theta) - f(\theta^*) + F(\theta, \boldsymbol{\xi}(t+\tau)) - F(\theta^*, \boldsymbol{\xi}(t+\tau))|\mathcal{F}_t] \leq 2LD\|P(t+\tau|t) - P^\infty\|_{\mathrm{TV}} .$$

**Lemma 8** (Consecutive iterates bound). *Let $(\theta(t))_{t\geq 0}$ be generated according to (10). Then, for any $t \geq 0$:*

$$\|\theta(t+1) - \theta(t)\| \leq 3L_f \left(1 + \frac{1}{2t+1}\right) (\Gamma(t+1) - \Gamma(t)) ,$$

*where for $t \geq 0$, $\Gamma(t) = t\gamma(t)$. In addition, if $\gamma(t) \propto t^\alpha$ for some $\alpha \in (-1, 0)$, then for any $t \geq 0$:*

$$\|\theta(t+1) - \theta(t)\| \leq 3L_f \left(1 + \frac{1}{2t+1}\right)(\alpha+1)\gamma(t) \leq 6L_f\gamma(t) .$$

When $\alpha = 1/2$, the bound is equivalent to $\frac{3}{2}L_f\gamma(t)$ when $t$ goes to infinity. This is similar to the $L_f\gamma(t)$ bound of other first order methods (gradient descent, mirror descent, *etc.*).

Lemma 8 provides a bound over the distance of two consecutive iterates. We now use this result to control term (29) in the regret decomposition.

**Lemma 9** (Gap with noisy objectives)**.** *Let $\tau \geq 0$. If $(\theta(t))_{t \geq 0}$ is generated according to (10), then for any $t \geq 0$:*

$$F(\theta(t), \boldsymbol{\xi}(t+\tau)) - F(\theta(t+\tau), \boldsymbol{\xi}(t+\tau)) \leq 3L^2 \left(1 + \frac{1}{2t+1}\right) \left(\Gamma(t+\tau) - \Gamma(t)\right) \ .$$

*Moreover, if $\gamma(t) \propto t^\alpha$ for some $\alpha \in (-1, 0)$, one has:*

$$F(\theta(t), \boldsymbol{\xi}(t+\tau)) - F(\theta(t+\tau), \boldsymbol{\xi}(t+\tau)) \leq 3L^2 \tau \left(1 + \frac{1}{2t+1}\right) (1+\alpha)\gamma(t) \leq 6L^2 \tau \gamma(t) \ .$$

Finally, we bound the term (30), which corresponds to the optimization regret. This bound is a quite straightforward adpatation from the regular dual averaging algorithm bound in Nesterov (2009).

**Lemma 10** (Optimization error)**.** *For any $\theta^* \in \mathcal{X}$, one has:*

$$\sum_{t=\tau+1}^{T} F(\theta(t), \boldsymbol{\xi}(t)) - F(\theta^*, \boldsymbol{\xi}(t)) \leq \frac{\|\theta^*\|^2}{2\gamma(T)} + \frac{L^2}{2} \sum_{t=\tau+1}^{T} \gamma(t) \ .$$

*Proof.* For any $\boldsymbol{\xi} \in \Delta_n$, $F(\cdot, \boldsymbol{\xi})$ is convex. Therefore, one has for any $\theta^* \in \mathcal{X}$:

$$\sum_{t=\tau+1}^{T} F(\theta(t), \boldsymbol{\xi}(t)) - F(\theta^*, \boldsymbol{\xi}(t)) \leq \sum_{t=\tau+1}^{T} \nabla_\theta F(\theta(t), \boldsymbol{\xi}(t))^\top (\theta(t) - \theta^*) \ .$$

One can then conclude using the definition of $(\theta(t))_{t \geq 0}$ and the proof of dual averaging convergence in a standard setting—see (Nesterov, 2009, Theorem 1) for instance. □

From now on, we assume that there exists $\alpha \in (-1, 0)$ such that $\gamma(t) \propto \alpha$. This allows for easier convergence analysis but a more general analysis can still be performed using the bound provided in Lemma 9. We can now apply previous results to the expected regret; for any $\tau \geq 0$, one has:

$$\sum_{t=1}^{t} \mathbb{E}[(f(\theta(t)) - f(\theta^*))] \leq 2LD \sum_{t=1}^{T-\tau} \|P(t+\tau|t) - P^\infty\|_{TV}$$

$$+ 3L^2 \sum_{t=1}^{T-\tau} \left(1 + \frac{1}{2t+1}\right)(1+\alpha)\gamma(t)$$

$$+ \frac{\|\theta^*\|^2}{2\gamma(T)} + \frac{L^2}{2} \sum_{t=\tau+1}^{T} \gamma(t)$$

$$+ \sum_{t=T-\tau+1}^{T} \mathbb{E}[(f(\theta(t)) - f(\theta^*))] \ .$$

We made the assumption $\theta \leq D$, so $f(\theta(t)) - f(\theta^*) \leq 2LD$, and one has the following bound:

$$\sum_{t=1}^{t} \mathbb{E}[(f(\theta(t)) - f(\theta^*))] \leq 2LD \sum_{t=1}^{T-\tau} \|P(t+\tau|t) - P^\infty\|_{TV}$$

$$+ 6L^2 \sum_{t=1}^{T-\tau} \gamma(t) + \frac{\|\theta^*\|^2}{2\gamma(T)} + \frac{L^2}{2} \sum_{t=\tau+1}^{T} \gamma(t) + \tau LD \ .$$

Let us assume that the mixing times are uniform, that is for any $\epsilon > 0$, there exists $\tau(\epsilon)$ such that:

$$\forall t \geq 0, \ \tau(t, \epsilon) \leq \tau(\epsilon) \ .$$

Therefore, one has for any $\epsilon > 0$:

$$\sum_{t=1}^{t} \mathbb{E}[(f(\theta(t)) - f(\theta^*))] \leq 2LD(T\epsilon + \tau(\epsilon)) + \frac{L^2}{2}\big(1 + 12\tau(\epsilon)\big)\sum_{t=1}^{T}\gamma(t) + \frac{\|\theta^*\|^2}{2\gamma(T)} \;,$$

and we can write the following theorem.

**Theorem 8** (Ergodic dual averaging)**.** *Let $(\theta(t))_{t\geq 0}$ be generated according to (10) and let $\theta^* \in \mathcal{X}$ be a minimizer of the problem (25). We make the following assumptions:*

1. *There exists $\alpha \in (-1, 0)$ such that $\gamma(t) \propto t^\alpha$.*

2. *There exists $D > 0$ such that for any $\theta \in \mathcal{X}$, $\|\theta\| \leq D$.*

3. *For any $\epsilon > 0$, there exists $\tau(\epsilon)$ such that for any $t \geq 0$, $\tau(t, \epsilon) \leq \tau(\epsilon)$.*

*Then, for any $\epsilon > 0$:*

$$\mathbb{E}[(f(\bar{\theta}(T))] - f(\theta^*)] \leq 2LD\left(\epsilon + \frac{\tau(\epsilon)}{T}\right) + \frac{L^2}{2T}\big(1 + 12\tau(\epsilon)\big)\sum_{t=1}^{T}\gamma(t) + \frac{\|\theta^*\|^2}{2T\gamma(T)} \;, \tag{32}$$

*where $\bar{\theta}(T) = \frac{1}{T}\sum_{t=1}^{T}\theta(t)$ is the iterates average at time $T$.*

Note that if one is able to compute $\nabla F(\cdot, \mathbf{1}_n/n)$ at every iteration, then $\tau(\epsilon) = 0$ for any $\epsilon > 0$ and one can recover the dual averaging convergence rate when $\epsilon \to 0$ in (32). This upper-bound evidences the need to compare the mixing time to the optimization rate: if $\tau(\epsilon) \ll \sqrt{T}$ then similar bounds are preserved.

### D.2 Proofs

This section is organized as follows. First, we establish proofs in Section D.2.1 for each lemma involved in the ergodic dual averaging analysis. Then, we perform the analysis of the pairwise decentralized dual averaging in the synchronous case. Finally, we tackle the proof of convergence in the fully asynchronous setting.

#### D.2.1 Ergodic dual averaging

**Error after mixing (Lemma 7)**

*Proof.* Let $\boldsymbol{\theta}$ be a $\mathcal{F}_t$-mesurable variable, $\boldsymbol{\theta}^* \in \mathcal{X}$ and $\tau \geq 0$. By definition of $P(t + \tau | t)$ and $P^\infty$, the LHS can be rewritten as follows:

$$\int_{\boldsymbol{\xi}\in\Delta_n} \big(F(\boldsymbol{\theta}, \boldsymbol{\xi}) - F(\boldsymbol{\theta}^*, \boldsymbol{\xi})\big)\mathrm{d}P^\infty(\boldsymbol{\xi}) - \int_{\boldsymbol{\xi}\in\Delta_n} \big(F(\boldsymbol{\theta}, \boldsymbol{\xi}) - F(\boldsymbol{\theta}^*, \boldsymbol{\xi})\big)\mathrm{d}P(t + \tau | t)(\boldsymbol{\xi}) \;. \tag{33}$$

Both expectations in (33) only differ from the probability measures involved; the above quantity can thus be bounded by:

$$\int_{\boldsymbol{\xi}\in\Delta_n} \big(F(\boldsymbol{\theta}, \boldsymbol{\xi}) - F(\boldsymbol{\theta}^*, \boldsymbol{\xi})\big)|\mathrm{d}P^\infty(\boldsymbol{\xi}) - \mathrm{d}P(t + \tau | t)(\boldsymbol{\xi})| \;.$$

Using the fact that for any $\boldsymbol{\xi} \in \Delta_n$, $F(\cdot, \boldsymbol{\xi})$ is $L_f$-Lipschitz, one has, for any $\boldsymbol{\theta}, \boldsymbol{\theta}^* \in \mathcal{X}$:

$$|F(\boldsymbol{\theta}, \boldsymbol{\xi}) - F(\boldsymbol{\theta}^*, \boldsymbol{\xi})| \leq L_f\|\boldsymbol{\theta} - \boldsymbol{\theta}^*\| \leq L_f D \;,$$

the last inequality deriving from the definition of $D$. Then the result holds using the definition of the total variation norm. □

**Consecutive iterates bound (Lemma 8)**

*Proof.* Let $(\mathbf{z}(t), \boldsymbol{\theta}(t))_{t \geq 0}$ be generated according to (10) for some positive, non-increasing sequence $(\gamma(t))_{t \geq 0}$. For any $t \geq 0$, we aim at bounding $\|\boldsymbol{\theta}(t+1) - \boldsymbol{\theta}(t)\|$. Let $s(t) \in \partial \psi(\boldsymbol{\theta}(t))$ and $s(t+1) \in \partial \psi(\boldsymbol{\theta}(t+1))$. The respective optimality conditions on $\boldsymbol{\theta}(t)$ and $\boldsymbol{\theta}(t+1)$ lead to the following inequalities:

$$
\begin{cases}
\big(\gamma(t)\mathbf{z}(t) - \boldsymbol{\theta}(t) - \Gamma(t)s(t)\big)^{\top} \big(\boldsymbol{\theta}(t+1) - \boldsymbol{\theta}(t)\big) & \leq & 0 \\
\big(\gamma(t+1)\mathbf{z}(t+1) - \boldsymbol{\theta}(t+1) - \Gamma(t+1)s(t+1)\big)^{\top} \big(\boldsymbol{\theta}(t) - \boldsymbol{\theta}(t+1)\big) & \leq & 0
\end{cases}. \tag{34}
$$

Then, using convexity of $\psi$ and the property of the subgradient leads to:

$$
\begin{cases}
s(t+1)^{\top}(\boldsymbol{\theta}(t+1) - \boldsymbol{\theta}(t)) & \geq & \psi(\boldsymbol{\theta}(t+1)) - \psi(\boldsymbol{\theta}(t)) \\
s(t)^{\top}(\boldsymbol{\theta}(t) - \boldsymbol{\theta}(t+1)) & \geq & \psi(\boldsymbol{\theta}(t)) - \psi(\boldsymbol{\theta}(t+1))
\end{cases}. \tag{35}
$$

Summing both inequalities in (34) and using (35), one obtains:

$$
\|\boldsymbol{\theta}(t+1) - \boldsymbol{\theta}(t)\|^2 \leq \big(\gamma(t+1)\mathbf{z}(t+1) - \gamma(t)\mathbf{z}(t)\big)^{\top} \big(\boldsymbol{\theta}(t+1) - \boldsymbol{\theta}(t)\big) \\
+ \big(\Gamma(t+1) - \Gamma(t)\big)\big(\psi(\boldsymbol{\theta}(t+1)) - \psi(\boldsymbol{\theta}(t))\big) . \tag{36}
$$

The optimality of $\boldsymbol{\theta}(t+1)$ ensures the following relation:

$$
\boldsymbol{\theta}(t+1)^{\top}\frac{\mathbf{z}(t+1)}{t+1} - \frac{\|\boldsymbol{\theta}(t+1)\|^2}{2\Gamma(t+1)} - \psi(\boldsymbol{\theta}(t+1)) \geq \boldsymbol{\theta}(t)^{\top}\frac{\mathbf{z}(t+1)}{t+1} - \frac{\|\boldsymbol{\theta}(t)\|^2}{2\Gamma(t+1)} - \psi(\boldsymbol{\theta}(t)) .
$$

We can reformulate this last inequality to provide an upper bound on $\psi(\boldsymbol{\theta}(t+1)) - \psi(\boldsymbol{\theta}(t))$:

$$
\begin{aligned}
\psi(\boldsymbol{\theta}(t+1)) - \psi(\boldsymbol{\theta}(t)) &\leq (\boldsymbol{\theta}(t+1) - \boldsymbol{\theta}(t))^{\top}\left(\frac{\mathbf{z}(t+1)}{t+1} + \frac{\boldsymbol{\theta}(t) - \boldsymbol{\theta}(t+1)}{2\Gamma(t+1)}\right) \\
&\quad + (\boldsymbol{\theta}(t+1) - \boldsymbol{\theta}(t))^{\top}\frac{\boldsymbol{\theta}(t+1)}{\Gamma(t+1)} \\
&\leq \|\boldsymbol{\theta}(t+1) - \boldsymbol{\theta}(t)\|\left(\frac{\|\mathbf{z}(t+1)\|}{t+1} + \frac{\|\boldsymbol{\theta}(t) - \boldsymbol{\theta}(t+1)\|}{2\Gamma(t+1)}\right) \\
&\quad + \|\boldsymbol{\theta}(t+1) - \boldsymbol{\theta}(t)\|\frac{\|\boldsymbol{\theta}(t+1)\|}{\Gamma(t+1)} ,
\end{aligned}
$$

where the last inequality is derived from Cauchy-Schwarz relation. Since $\pi_{t+1}$ is $\gamma(t+1)$-Lipschitz and $\pi_{t+1}(0) = 0$, one has $\|\boldsymbol{\theta}(t+1)\| \leq \gamma(t+1)\|\mathbf{z}(t+1)\|$. Moreover, by definition of $\mathbf{z}(t+1)$ and since all $f_i$ are $L$-Lipschitz, one has $\|\mathbf{z}(t+1)\| \leq (t+1)L$. These two last results lead the following bound:

$$
\psi(\boldsymbol{\theta}(t+1)) - \psi(\boldsymbol{\theta}(t)) \leq 2L\|\boldsymbol{\theta}(t+1) - \boldsymbol{\theta}(t)\| + \frac{\|\boldsymbol{\theta}(t) - \boldsymbol{\theta}(t+1)\|^2}{2\Gamma(t+1)} .
$$

Now, we can use this bound in inequality (36):

$$
\left(1 - \frac{\Gamma(t+1) - \Gamma(t)}{2\Gamma(t+1)}\right)\|\boldsymbol{\theta}(t+1) - \boldsymbol{\theta}(t)\| \leq \|\gamma(t+1)\mathbf{z}(t+1) - \gamma(t)\mathbf{z}(t)\| + 2(\Gamma(t+1) - \Gamma(t))L .
$$

The first term in the RHS can be simply bounded as follows:

$$
\begin{aligned}
\|\gamma(t+1)\mathbf{z}(t+1) - \gamma(t)\mathbf{z}(t)\| &\leq \gamma(t+1)\|\mathbf{z}(t+1) - \mathbf{z}(t)\| + (\gamma(t+1) - \gamma(t))\|\mathbf{z}(t)\| \\
&\leq (\Gamma(t+1) - \Gamma(t))L .
\end{aligned}
$$

Since $\Gamma(t+1)$ and $\Gamma(t)$ are both positive, one has:

$$
1 - \frac{\Gamma(t+1) - \Gamma(t)}{2\Gamma(t+1)} = \frac{\Gamma(t+1) + \Gamma(t)}{2\Gamma(t+1)} \geq 0 ,
$$

which finally leads to:

$$\|\boldsymbol{\theta}(t+1) - \boldsymbol{\theta}(t)\| \leq 3L\big(\Gamma(t+1) - \Gamma(t)\big)\frac{2\Gamma(t+1)}{\Gamma(t+1) + \Gamma(t)} \leq 3L\big(\Gamma(t+1) - \Gamma(t)\big)\left(1 + \frac{1}{2t+1}\right) \ .$$

We make the additional assumption that $\gamma(t) \propto t^\alpha$ for some $\alpha \in (-1, 0)$. This is not a particularly restrictive assumption since the dual averaging algorithm imposes that:

1. $\lim_{t\to\infty} \gamma(t) = 0$, hence $\alpha < 0$.

2. $\lim_{t\to\infty} t\gamma(t) = +\infty$, hence $\alpha + 1 > 0$.

With this assumption and Taylor-Lagrange formula yields:

$$\Gamma(t+1) - \Gamma(t) \leq (\alpha + 1)\gamma(t) \ ,$$

and the final result holds. $\qquad\square$

**Gap with noisy objectives (Lemma 9)**

*Proof.* Let $\tau, t \geq 0$. One has:

$$F(\boldsymbol{\theta}(t), \boldsymbol{\xi}(t+\tau)) - F(\boldsymbol{\theta}(t+\tau), \boldsymbol{\xi}(t+\tau)) \leq L\|\boldsymbol{\theta}(t) - \boldsymbol{\theta}(t+\tau)\|$$

$$\leq L\sum_{s=0}^{\tau-1} \|\boldsymbol{\theta}(t+s) - \boldsymbol{\theta}(t+s+1)\| \ .$$

Using Lemma 8, one has for any $0 \leq s \leq \tau - 1$:

$$\|\boldsymbol{\theta}(t+s) - \boldsymbol{\theta}(t+s+1)\| \leq 3L\left(1 + \frac{1}{2(t+s)+1}\right)\big(\Gamma(t+s+1) - \Gamma(t+s)\big)$$

$$\leq 3L\left(1 + \frac{1}{2t+1}\right)\big(\Gamma(t+s+1) - \Gamma(t+s)\big) \ .$$

Summing over $s$ leads to:

$$\sum_{s=0}^{\tau-1} \|\boldsymbol{\theta}(t+s) - \boldsymbol{\theta}(t+s+1)\| \leq 3L\left(1 + \frac{1}{2t+1}\right)\big(\Gamma(t+\tau) - \Gamma(t)\big) \ ,$$

and the first claim holds. We now make the assumption that $\gamma(t) \propto t^\alpha$, with $\alpha \in (-1, 0)$. As denoted in the proof of Lemma 8, one has:

$$\Gamma(t+\tau) - \Gamma(t) \leq \tau(1+\alpha)\gamma(t) \ ,$$

so the second claim also holds. $\qquad\square$

### D.2.2 Synchronous Pairwise Gossip Dual Averaging

In this section, we focus on the synchronous setting. First, we establish a result on the expected dispersion of the dual variables over the network. We then use this result to detail the rate of the decentralized dual averaging, both for separable and pairwise objectives. Finally, we use the ergodic dual averaging to provide an explicit rate of convergence.

In Duchi et al. (2012a), the following convergence rate for distributed dual averaging is established:

$$\mathbb{E}[R_n(\bar{\boldsymbol{\theta}}_i(T))] - R_n(\boldsymbol{\theta}^*) \leq \frac{1}{2T\gamma(T)}\|\boldsymbol{\theta}^*\|^2 + \frac{L_f^2}{2T}\sum_{t=2}^{T}\gamma(t-1)$$

$$+ \frac{L_f}{nT}\sum_{t=2}^{T}\gamma(t-1)\sum_{j=1}^{n}\Big(\|\mathbf{z}_i(t) - \mathbf{z}_j(t)\| + \|\bar{\mathbf{z}}(t) - \mathbf{z}_j(t)\|\Big) \ .$$

The first part is an optimization term, which is exactly the same as in the centralized setting. Then, the second part is a network-dependent term which depends on the global variation of the dual variables; the following lemma provides an explicit dependence between this term and the topology of the network.

**Lemma 11.** *Let $(\mathbf{G}(t))_{t \geq 1}$ and $(\mathbf{Z}(t))_{t \geq 1}$ respectively be the gradients and the gradients cummulative sums of the distributed dual averaging algorithm. If $G$ is connected and non bipartite, then one has for $t \geq 1$:*

$$\frac{1}{n} \sum_{i=1}^{n} \mathbb{E} \|\mathbf{z}_i(t) - \bar{\mathbf{z}}(t)\| \leq \frac{L}{1 - \sqrt{1 - \lambda_{n-1}/|E|}} \ ,$$

*where $\lambda_{n-1}$ is the second smallest eigenvalue of the graph Laplacian $\mathbf{L}$.*

*Proof.* For $t \geq 1$, let $\mathbf{W}(t)$ be the random matrix such that if $(i, j) \in E$ is picked at $t$, then

$$\mathbf{W}(t) = \mathbf{I}_n - \frac{1}{2}(\mathbf{e}_i - \mathbf{e}_j)(\mathbf{e}_i - \mathbf{e}_j)^\top \ .$$

As denoted in Duchi et al. (2012a), the update rule for $\mathbf{Z}$ can be expressed as follows:

$$\mathbf{Z}(t+1) = \mathbf{G}(t) + \mathbf{W}(t)\mathbf{Z}(t),$$

for any $t \geq 1$, reminding that $\mathbf{G}(0) = 0, \mathbf{Z}(1) = 0$. Therefore, one can obtain recursively

$$\mathbf{Z}(t) = \sum_{s=0}^{t} \mathbf{W}(t:s)\mathbf{G}(s),$$

where $\mathbf{W}(t:s) = \mathbf{W}(t) \dots \mathbf{W}(s+1)$, with the convention $\mathbf{W}(t:t) = \mathbf{I}_n$. For any $t \geq 1$, let $\tilde{\mathbf{W}}(t)$ be defined as follows:

$$\tilde{\mathbf{W}}(t) := \mathbf{W}(t) - \mathbf{J}_n \ .$$

One can notice that for any $0 \leq s \leq t$, $\tilde{\mathbf{W}}(t:s) = \mathbf{W}(t:s) - \mathbf{J}_n$ and write:

$$\mathbf{Z}(t) - \mathbf{1}_n \hat{\mathbf{z}}(t)^\top = \sum_{s=0}^{t} \tilde{\mathbf{W}}(t:s)\mathbf{G}(s).$$

We now take the expected value of the Frobenius norm:

$$\mathbb{E}\left[\left\|\mathbf{Z}(t) - \mathbf{1}_n \hat{\mathbf{z}}(t)^\top\right\|_F\right] \leq \sum_{s=0}^{t} \mathbb{E}\left[\|\mathbf{W}(t:s)\mathbf{G}(s)\|_F\right] \leq \sum_{s=0}^{t} \sqrt{\mathbb{E}\left[\|\mathbf{W}(t:s)\mathbf{G}(s)\|_F^2\right]}$$

$$= \sum_{i=1}^{n} \sum_{s=0}^{t} \sqrt{\mathbb{E}\left[\mathbf{g}^{(i)}(s)^\top \tilde{\mathbf{W}}(t:s)^\top \tilde{\mathbf{W}}(t:s)\mathbf{g}^{(i)}(s)\right]},$$

where $\mathbf{g}^{(i)}(s)$ is the $i$-th column of matrix $\mathbf{G}(s)$. Conditioning over $\mathcal{F}_{t-1}$ leads to:

$$\mathbb{E}\left[\mathbf{g}^{(i)}(s)^\top \tilde{\mathbf{W}}(t:s)^\top \tilde{\mathbf{W}}(t:s)\mathbf{g}^{(i)}(s)\right] \leq \lambda_2(2)\mathbb{E}\left[\mathbf{g}^{(i)}(s)^\top \tilde{\mathbf{W}}(t-1:s)\mathbf{g}^{(i)}(s)\right] \ .$$

where $\lambda_2(2)$ is defined in Section A. Using the fact that for any $s \geq 0$, $\|\mathbf{G}(s)\|_F^2 \leq nL^2$, one has:

$$\mathbb{E}\left[\left\|\mathbf{Z}(t) - \mathbf{1}_n \hat{\mathbf{z}}(t)^\top\right\|_F\right] \leq \sqrt{n}L \sum_{s=0}^{t} \lambda_2(2)^{\frac{t-s}{2}} \leq \frac{\sqrt{n}L}{1 - \sqrt{\lambda_2(2)}} \ .$$

Finally, using the bounds between $\ell_1$ and $\ell_2$-norms yields:

$$\frac{1}{n} \sum_{i=1}^{n} \mathbb{E}\|\mathbf{z}_i(t) - \hat{\mathbf{z}}(t)\| \leq \frac{1}{\sqrt{n}} \mathbb{E}\left\|\mathbf{Z}(t) - \mathbf{1}_n \hat{\mathbf{z}}(t)^\top\right\|_F \leq \frac{L}{1 - \sqrt{\lambda_2(2)}} \ .$$

$\square$

With this bound on dual variables, the convergence rate can be reformulated as follows.

**Corollary 2.** *Let $\mathcal{G} = ([n], E)$ be a connected and non bipartite graph. Let $(\gamma(t))_{t \geq 1}$ be a non-increasing and non-negative sequence. For $i \in [n]$, let $(\mathbf{g}_i(t))_{t \geq 1}$, $(\mathbf{z}_i(t))_{t \geq 1}$ and $(\boldsymbol{\theta}_i(t))_{t \geq 1}$ be generated according to the distributed dual averaging algorithm. For $\boldsymbol{\theta}^* \in \arg\min_{\boldsymbol{\theta}' \in \mathbb{R}^d} R_n(\boldsymbol{\theta}')$, $i \in [n]$ and $T \geq 2$, one has:*

$$\mathbb{E}[R_n(\bar{\boldsymbol{\theta}}_i(T))] - R_n(\boldsymbol{\theta}^*) \leq \frac{1}{2T\gamma(T)}\|\boldsymbol{\theta}^*\|^2 + \frac{L^2}{2T}\sum_{t=1}^{T-1}\gamma(t) + \frac{3L^2}{T\left(1 - \sqrt{\lambda_2(2)}\right)}\sum_{t=1}^{T-1}\gamma(t),$$

*where $\lambda_2(2) < 1$ is the second largest eigenvalue of $\mathbf{W}_2$.*

We now focus on gossip dual averaging for pairwise functions, as shown in Algorithm 2. The key observation is that, at each iteration, the descent direction is stochastic but also a *biased* estimate of the gradient. That is, instead of updating a dual variable $\mathbf{z}_i(t)$ with $\mathbf{g}_i(t)$ such that $\mathbb{E}[\mathbf{g}_i(t)|\boldsymbol{\theta}_i(t)] = \nabla f_i(\boldsymbol{\theta}_i(t))$, we perform some update $\mathbf{d}_i(t)$, and we denote by $\boldsymbol{\epsilon}_i(t)$ the quantity such that $\mathbb{E}[\mathbf{d}_i(t) - \boldsymbol{\epsilon}_i(t)|\boldsymbol{\theta}_i(t)] = \mathbb{E}[\mathbf{g}_i(t)|\boldsymbol{\theta}_i(t)] = \nabla f_i(\boldsymbol{\theta}_i(t))$. We now prove Proposition 1, which upper-bound the error with an additional bias-dependent term.

*Proof.* We can apply the same arguments as in the proofs of centralized and distributed dual averaging, so for $T > 0$ and $i \in [n]$:

$$\mathbb{E}_T[R_n(\bar{\boldsymbol{\theta}}_i(T))] - R_n(\boldsymbol{\theta}^*) \leq \frac{L}{nT}\sum_{t=2}^{T}\gamma(t-1)\sum_{j=1}^{n}\mathbb{E}\Big[\|\mathbf{z}_i(t) - \mathbf{z}_j(t)\|\Big]$$

$$+ \frac{L}{nT}\sum_{t=2}^{T}\gamma(t-1)\sum_{j=1}^{n}\|\hat{\mathbf{z}}(t) - \mathbf{z}_j(t)\| + \frac{1}{T}\sum_{t=2}^{T}\mathbb{E}[(\hat{\boldsymbol{\omega}}(t) - \boldsymbol{\theta}^*)^\top\hat{\mathbf{g}}(t)]$$

$$\leq \frac{3L}{nT}\sum_{t=2}^{T}\gamma(t-1)\sum_{j=1}^{n}\|\hat{\mathbf{z}}(t) - \mathbf{z}_j(t)\| + \frac{1}{T}\sum_{t=2}^{T}\mathbb{E}[(\hat{\boldsymbol{\omega}}(t) - \boldsymbol{\theta}^*)^\top\hat{\mathbf{g}}(t)] .$$

The first term can be bound using Lemma 11. The second term however can no longer be bound using Lemma 6, since the updates are performed with $\mathbf{d}_j(t)$ and not $\mathbf{g}_j(t) = \mathbf{d}_j(t) - \boldsymbol{\epsilon}_j(t)$. With the definition of $\mathbf{d}_j(t)$, the former yields:

$$\frac{1}{T}\sum_{t=2}^{T}\mathbb{E}[(\hat{\boldsymbol{\omega}}(t) - \boldsymbol{\theta}^*)^\top\hat{\mathbf{g}}(t)] = \frac{1}{T}\sum_{t=2}^{T}\mathbb{E}[(\hat{\boldsymbol{\omega}}(t) - \boldsymbol{\theta}^*)^\top(\hat{\mathbf{d}}(t) - \hat{\boldsymbol{\epsilon}}(t))] .$$

Now Lemma 6 can be applied to the first term in the right hand side and the result holds. $\qquad\square$

We now focus on the proof of 2. This results is based both on Proposition 1 and ergodic dual averaging presented in Section D.1.4.

*Proof.* Throughout this proof, we assume $\psi \equiv 0$ for simplicity; similar results can however be obtained in the case $\psi \not\equiv 0$. Let $i \in [n]$ and $T \geq 1$. One has:

$$R_n(\bar{\boldsymbol{\theta}}_i(T)) - R_n(\boldsymbol{\theta}^*) \leq \frac{1}{nT}\sum_{t=1}^{T}\sum_{j=1}^{n}L\gamma(t)\|\mathbf{z}_i(t) - \mathbf{z}_j(t)\| + \frac{1}{nT}\sum_{t=1}^{T}\sum_{j=1}^{n}\big(f_j(\boldsymbol{\theta}_j(t)) - f_j(\boldsymbol{\theta}^*)\big)$$

$$\leq \frac{2L}{nT}\sum_{t=1}^{T}\sum_{j=1}^{n}\gamma(t)\|\mathbf{z}_i(t) - \hat{\mathbf{z}}(t)\| + \frac{1}{nT}\sum_{t=1}^{T}\sum_{j=1}^{n}\big(f_j(\boldsymbol{\theta}_j(t)) - f_j(\boldsymbol{\theta}^*)\big) .$$

The first term in the right hand side can be bounded using Lemma 11, so we only need to handle the last term. To do so, we adapt the proof of convergence for the ergodic dual averaging to the partial objectives. For $j \in [n]$, let us define $F_j : \mathbb{R}^d \times \Delta_n \to \mathbb{R}$ such that for any $(\boldsymbol{\theta}, \boldsymbol{\xi}) \in \mathbb{R}^d \times \Delta_n$:

$$F_j(\boldsymbol{\theta}, \boldsymbol{\xi}) = \sum_{k=1}^{n}\xi_k f(\boldsymbol{\theta}; \mathbf{x}_j, \mathbf{x}_k).$$

Moreover, for $t \geq 1$, we define $\boldsymbol{\xi}_j(t) \in \{0,1\}^n \cap \Delta_n$ such that for $k \in [n]$, $\boldsymbol{\xi}_j(t)^\top \mathbf{e}_k = 1$ if and only if $\mathbf{y}_j(t) = \mathbf{x}_k$. Deriving the decomposition of the ergodic dual averaging proof yields:

$$\sum_{t=1}^{T}(f_j(\boldsymbol{\theta}_j(t)) - f_j(\boldsymbol{\theta}^*)) =$$

$$\sum_{t=1}^{T-\tau}(f_j(\boldsymbol{\theta}_j(t)) - f_j(\boldsymbol{\theta}^*) + F_j(\boldsymbol{\theta}_j(t), \boldsymbol{\xi}_j(t+\tau)) - F_j(\boldsymbol{\theta}^*, \boldsymbol{\xi}_j(t+\tau))) \tag{37}$$

$$+ \sum_{t=1}^{T-\tau}(F_j(\boldsymbol{\theta}_j(t), \boldsymbol{\xi}_j(t+\tau)) - F_j(\boldsymbol{\theta}_j(t+\tau), \boldsymbol{\xi}_j(t+\tau))) \tag{38}$$

$$+ \sum_{t=\tau+1}^{T}(F_j(\boldsymbol{\theta}_j(t), \boldsymbol{\xi}_j(t)) - F_j(\boldsymbol{\theta}^*, \boldsymbol{\xi}_j(t))) \tag{39}$$

$$+ \sum_{t=T-\tau+1}^{T}(f_j(\boldsymbol{\theta}_j(t)) - f_j(\boldsymbol{\theta}^*)). \tag{40}$$

Bounding the term (37) requires the knowledge of the total variation distance between the random walk associated to $j$ after $\tau$ algorithm steps and the uniform. (Chung, 1997, Theorem 1.18) states that such norm for one random walk is upper-bounded as follows:

$$\|P(t+\tau|t) - P_\infty\|_{\text{TV}} \leq \frac{|E|\tilde{\lambda}_2(2)^\tau}{2\min_{k\in[n]} d_k},$$

where $\tilde{\lambda}_2(2)$ is such that $\tilde{\lambda}_2(2) \leq 1 - \frac{\lambda_{n-1}}{\max_{k\in[n]} d_k}$. However, in this case, the random walk associated to the $j$-th auxiliary observation will not necessarily be propagated $\tau$ times during $\tau$ algorithm steps. Since we are interested in expected bound, we bound the expected total variation norm as follows:

$$\mathbb{E}\|P_j(t+\tau|t) - P_\infty\|_{\text{TV}} \leq \sum_{s=0}^{\tau} \mathbb{P}\left(\sum_{r=t}^{t+\tau} \delta_j(r) = s\right) \frac{|E|\tilde{\lambda}_2(2)^s}{2\min_{k\in[n]} d_k}$$

$$\leq \frac{|E|}{2\min_{k\in[n]} d_k} \sum_{s=0}^{\tau} \binom{\tau}{s} p_j^s(1-p_j)^{\tau-s}\tilde{\lambda}_2(2)^s$$

$$= \frac{|E|}{2\min_{k\in[n]} d_k}\left((1-p_j) + p_j\tilde{\lambda}_2(2)\right)^\tau \leq c(G) \cdot c'(G)^\tau,$$

where $c(G) := \frac{|E|}{2\min_{k\in[n]} d_k}$ and $c'(G) := \max_{k\in[n]}\left\{(1-p_k) + p_k\tilde{\lambda}_2(2)\right\}$.

The term (38) provides an upper-bound similar to the centralized ergodic case, as it only depends on the Lipschitz constant $L$ and the step size sequence. When averaging over all $j \in [n]$, the term (39) can be upper-bounded as follows:

$$\frac{1}{n}\sum_{j=1}^{n}\sum_{t=\tau+1}^{T}(F_j(\boldsymbol{\theta}_j(t), \boldsymbol{\xi}_j(t)) - F_j(\boldsymbol{\theta}^*, \boldsymbol{\xi}_j(t)))$$

$$\leq \frac{1}{n}\sum_{j=1}^{n}\sum_{t=\tau+1}^{T}(F_j(\boldsymbol{\theta}_j(t), \boldsymbol{\xi}_j(t)) - F_j(\widehat{\boldsymbol{\omega}}(t), \boldsymbol{\xi}_j(t))) + \sum_{t=1}^{T}(\widehat{\boldsymbol{\omega}}(t) - \boldsymbol{\theta}^*)^\top \widehat{\mathbf{d}}(t)$$

$$\leq \frac{L}{n}\sum_{j=1}^{n}\sum_{t=\tau+1}^{T}\gamma(t)\|\mathbf{z}_j(t) - \widehat{\mathbf{z}}(t)\| + \sum_{t=1}^{T}(\widehat{\boldsymbol{\omega}}(t) - \boldsymbol{\theta}^*)^\top \widehat{\mathbf{d}}(t),$$

which can then be bounded similarly to Proposition 1. Finally, the last term (40) is bounded by $2\tau LD$, as in the centralized case. $\square$

---

**Algorithm 7** Gossip dual averaging for pairwise function in asynchronous setting

---

**Require:** Step size $(\gamma(t))_{t\geq 0} > 0$, probabilities $(p_k)_{k\in[n]}$
1: $\mathbf{y}_i \leftarrow \mathbf{x}_i, \quad \mathbf{z}_i \quad \leftarrow 0, \quad \boldsymbol{\theta}_i \leftarrow 0, \quad \bar{\theta}_i \quad \leftarrow 0, \quad m_i \leftarrow 0 \quad$ for each node $i \in [n]$
2: **for** $t = 1, \ldots, T$ **do**
3: $\quad$ Draw $(i, j)$ uniformly at random in $E$
4: $\quad$ Swap auxiliary observations: $\mathbf{y}_i \leftrightarrow \mathbf{y}_j$
5: $\quad$ **for** $k \in \{i, j\}$ **do**
6: $\quad\quad \mathbf{z}_k \leftarrow \frac{\mathbf{z}_i + \mathbf{z}_j}{2}$
7: $\quad\quad \mathbf{z}_k \leftarrow \frac{1}{p_k} \nabla_{\boldsymbol{\theta}} f(\boldsymbol{\theta}_k; \mathbf{x}_k, \mathbf{y}_k)$
8: $\quad\quad m_k \leftarrow m_k + \frac{1}{p_k}$
9: $\quad\quad \boldsymbol{\theta}_k \leftarrow \pi_{m_k}(\mathbf{z}_k)$
10: $\quad\quad \bar{\theta}_k \leftarrow \left(1 - \frac{1}{m_k p_k}\right) \bar{\theta}_k$
11: $\quad$ **end for**
12: **end for**
13: **return** Each node $k \in [n]$ has $\bar{\theta}_k$

---

### D.2.3 Asynchronous Pairwise Dual Averaging

For any variant of gradient descent over a network with a decreasing step size, there is a need for a common time scale to perform the suitable decrease. In the synchronous setting, this time scale information can be shared easily among nodes by assuming the availability of a global clock. This is convenient for theoretical considerations, but is unrealistic in many practical (asynchronous) scenarios. As in the decentralized estimation framework considered in the appendix, we place ourselves here in a fully asynchronous setting where each node has a local clock, ticking at a Poisson rate of 1, independently from the others and we use the time estimators $(m_k)_{1\leq k\leq n}$ defined as follows:

$$m_k(t) = \begin{cases} m_k(t-1) + p_k^{-1} & \text{if } k \text{ is picked at iteration } t \\ m_k(t-1) & \text{otherwise} \end{cases}$$

Using these estimators, we can now adapt Algorithm 2 to the fully asynchronous case, as shown in Algorithm 7. The update step slightly differs from the synchronous case: the partial gradient has a weight $1/p_k$ instead of 1 so that all partial functions asymptotically count in equal way in every gradient accumulator. In contrast, uniform weights would penalize partial gradients from low degree nodes since the probability of being drawn is proportional to the degree. This weighting scheme is essential to ensure the convergence to the global solution. The model averaging step also needs to be altered: in absence of any global clock, the weight $1/t$ cannot be used and is replaced by $1/(m_k p_k)$, where $m_k p_k$ corresponds to the average number of times that node $k$ has been selected so far.

The following result is the analogue of Proposition 1 in the asynchronous framework.

**Proposition 3.** *Let $G = ([n], E)$ be a connected and non bipartite graph. Let $(\gamma(t))_{t\geq 1}$ be defined as $\gamma(t) = c/t^{1/2+\alpha}$ for some constant $c > 0$ and $\alpha \in (0, 1/2)$. For $i \in [n]$, let $(\mathbf{z}_i(t))_{t\geq 1}$ and $(\boldsymbol{\theta}_i(t))_{t\geq 1}$ be generated as described in Algorithm 7. Then, there exists some constant $C < +\infty$ such that, for $\boldsymbol{\theta}^* \in \arg\min_{\boldsymbol{\theta}' \in \mathbb{R}^d} F(\boldsymbol{\theta}')$, $i \in [n]$ and $T > 0$,*

$$F(\bar{\theta}_i(T)) - F(\boldsymbol{\theta}^*) \leq C \max(T^{-\alpha/2}, T^{\alpha-1/2}) + \frac{1}{T} \sum_{t=2}^{T} \mathbb{E}[(\widehat{\boldsymbol{\omega}}(t) - \boldsymbol{\theta}^*)^\top \widehat{\boldsymbol{\epsilon}}(t)] .$$

The main difficulty in the asynchronous setting is that each node $i$ has to use a time estimate $m_i$ instead of the global clock reference (that is no longer available in such a context). Even if the time estimate is unbiased, its variance puts an additional error term in the convergence rate. However, for an iteration $T$ *large enough*, one can bound these estimates as stated bellow.

**Lemma 12.** *There exists $T_1 > 0$ such that for any $t \geq T_1$, any $k \in [n]$ and any $q > 0$,*

$$t^- := t - t^{\frac{1}{2}+q} \leq m_k(t) \leq t + t^{\frac{1}{2}+q} =: t^+ \quad a.s.$$

*Proof.* Let $k \in [n]$. For $t \geq 1$, let us define $\delta_k(t)$ such that $\delta_k(t) = 1$ if $k$ is picked at iteration $t$ and $\delta_k(t) = 0$ otherwise. Then one has $m_k(t) = (1/p_k) \sum_{s=1}^{t} \delta_k(t)$. Since $(\delta_k(t))_{t \geq 1}$ is a Bernoulli process of parameter $1/p_k$, by the law of iterative logarithms Dudley (2010), (Nedić, 2011, Lemma 3) one has with probability 1 and for any $q > 0, \lim_{t \to +\infty} \frac{|m_k(t) - t|}{t^{\frac{1}{2} + q}} = 0$, and the result holds. $\qquad\square$

**Theorem 9.** *Let $G = ([n], E)$ be a connected and non bipartite graph. Let $(\gamma(t))_{t \geq 1}$ be defined as $\gamma(t) = c/t^{1/2+\alpha}$ for some constant $c > 0$ and $\alpha \in (0, 1/2)$. For $i \in [n]$, let $(\mathbf{z}_i(t))_{t \geq 1}$ and $(\boldsymbol{\theta}_i(t))_{t \geq 1}$ be generated as stated previously. For $\boldsymbol{\theta}^* \in \arg\min_{\boldsymbol{\theta}' \in \mathbb{R}^d} R_n(\boldsymbol{\theta}')$, $i \in [n]$ and $T > 0$, one has for some $C$:*

$$R_n(\bar{\boldsymbol{\theta}}_i(T)) - R_n(\boldsymbol{\theta}^*) \leq C \max(T^{-\alpha/2}, T^{\alpha-1/2}) + \frac{1}{T} \sum_{t=1}^{T} \mathbb{E}[\widehat{\boldsymbol{\epsilon}}(t)^\top \widehat{\boldsymbol{\omega}}(t)] \ .$$

*Proof.* In the asynchronous case, for $i \in [n]$ and $t \geq 1$, one has

$$\bar{\boldsymbol{\theta}}_i(T) = \frac{1}{m_i(T)} \sum_{t=1}^{T} \frac{\delta_i(t)}{p_i} \boldsymbol{\theta}_i(t) \ .$$

Then, using the convexity of $R_n$, one has:

$$\mathbb{E}_T[R_n(\bar{\boldsymbol{\theta}}_i(T)] - R_n(\boldsymbol{\theta}^*) \leq \mathbb{E}\left[ \frac{1}{m_i(T)} \sum_{t=1}^{T} \frac{\delta_i(t)}{p_i} R_n(\boldsymbol{\theta}_i(t)) \right] - R_n(\boldsymbol{\theta}^*) \ . \tag{41}$$

By Lemma 12, one has for $q > 0$

$$\mathbb{E}[R_n(\bar{\boldsymbol{\theta}}_i(T)) - R_n(\boldsymbol{\theta}^*)] \leq \frac{1}{T^-} \sum_{t=1}^{T} \mathbb{E}\left[ \frac{\delta_i(t)}{p_i} R_n(\boldsymbol{\theta}_i(t)) \right] - R_n(\boldsymbol{\theta}^*) \ .$$

Similarly to the synchronous case, one can write

$$\mathbb{E}\left[ \frac{\delta_i(t)}{p_i} f(\boldsymbol{\theta}_i(t)) \right] = \sum_{j=1}^{n} \frac{1}{n} \mathbb{E}\left[ \frac{\delta_i(t)}{p_i} f_j(\boldsymbol{\theta}_i(t)) \right]$$

$$= \frac{1}{n} \sum_{j=1}^{n} \mathbb{E}\left[ \frac{\delta_i(t)}{p_i} (f_j(\boldsymbol{\theta}_i(t)) - f_j(\boldsymbol{\theta}_j(t))) \right] + \frac{1}{n} \sum_{j=1}^{n} \mathbb{E}\left[ \frac{\delta_i(t)}{p_i} f_j(\boldsymbol{\theta}_j(t)) \right] \ .$$

In order to use the gradient inequality, we need to introduce $\delta_j(t)f_j(\boldsymbol{\theta}_j(t))$ instead of $\delta_i(t)f_j(\boldsymbol{\theta}_j(t))$. For $j \in [n]$, one has:

$$\frac{1}{T^-} \sum_{t=1}^{T} \mathbb{E}\left[ \frac{\delta_i(t)}{p_i} f_j(\boldsymbol{\theta}_j(t)) \right] = \frac{1}{T^-} \sum_{t=1}^{T} \mathbb{E}\left[ \left( \frac{\delta_i(t)}{p_i} - \frac{\delta_j(t)}{p_j} \right) f_j(\boldsymbol{\theta}_j(t)) + \frac{\delta_j(t)}{p_j} f_j(\boldsymbol{\theta}_j(t)) \right] \ .$$

Let $N_j = \sum_{t=1}^{T} \delta_j(t)$ and $1 \leq t_1 < \ldots < t_{N_j} \leq T$ be such that $\delta_j(t_k) = 1$ for $k \in [N_j]$. One can write

$$\frac{1}{T^-} \sum_{t=1}^{T} \mathbb{E}\left[ \left( \frac{\delta_i(t)}{p_i} - \frac{\delta_j(t)}{p_j} \right) f_j(\boldsymbol{\theta}_j(t)) \right] \tag{42}$$

$$= \frac{1}{T^-} \mathbb{E}\left[ \sum_{k=1}^{N_j - 1} \left( \left( \sum_{t=t_k}^{t_{k+1}-1} \frac{\delta_i(t)}{p_i} \right) - \frac{1}{p_j} \right) f_j(\boldsymbol{\theta}_j(t_k)) \right] + \frac{1}{T^-} \mathbb{E}\left[ \left( \sum_{t=0}^{t_1} \frac{\delta_i(t)}{p_i} \right) f_j(\boldsymbol{\theta}_j(0)) \right]$$

$$+ \frac{1}{T^-} \mathbb{E}\left[ \left( \left( \sum_{t=t_{N_j}}^{T} \frac{\delta_i(t)}{p_i} \right) - \frac{1}{p_j} \right) f_j(\boldsymbol{\theta}_j(t_{N_j})) \right]$$

$$\leq \frac{1}{T^-} \mathbb{E}\left[ \sum_{k=1}^{N_j - 1} \left( \left( \sum_{t=t_k}^{t_{k+1}-1} \frac{\delta_i(t)}{p_i} \right) - \frac{1}{p_j} \right) f_j(\boldsymbol{\theta}_j(t_k)) \right] + \frac{f_j(0)}{p_i p_j T^-} + \frac{L_f^2 \mathbb{E}[\gamma(t_{N_j} - 1)]}{p_i p_j} \ .$$

We need to study the behavior of $\delta_i$ and $\delta_j$ in the first term of the r.h.s. :

$$\mathbb{E}\left[\sum_{k=1}^{N_j-1}\left(\sum_{t=t_k}^{t_{k+1}-1}\frac{\delta_i(t)}{p_i}-\frac{1}{p_j}\right)f_j(\boldsymbol{\theta}_j(t_k))\right]$$

$$=\mathbb{E}\left[\sum_{k=1}^{N_j-1}\left(\mathbb{E}\left[\sum_{t=t_k}^{t_{k+1}-1}\frac{\delta_i(t)}{p_i}\Big|t_k,t_{k+1}\right]-\frac{1}{p_j}\right)f_j(\boldsymbol{\theta}_j(t_k))\right]\ .$$

$\delta_i(t)$ will not have the same dependency in $t_k$ whether $i$ and $j$ are connected or not. Let us first assume that $(i,j)\in E$. Then,

$$\mathbb{E}[\delta_i(t_k)|t_k]=\mathbb{E}[\delta_i(t)|\delta_j(t)=1]=\frac{1}{d_j}\ .$$

Also, for $t_k<t<t_{k+1}$, we get:

$$\mathbb{E}[\delta_i(t)|t_k]=\mathbb{E}[\delta_i(t)|\delta_j(t)=0]=\frac{p_i-2/|E|}{1-p_j}\ .$$

Finally, if $(i,j)\in E$, we obtain

$$\mathbb{E}\left[\sum_{t=t_k}^{t_{k+1}-1}\frac{\delta_i(t)}{p_i}\Big|t_k,t_{k+1}\right]=\left(\frac{1}{d_j}+(t_{k+1}-t_k-1)\frac{p_i-2/|E|}{1-p_j}\right)\frac{1}{p_i}\ .$$

Before using this relation in the full expectation, let us denote that since $t_{k+1}-t_k$ is independent from $t_k$, one can write

$$\mathbb{E}\left[\left(\frac{1}{d_j}+(t_{k+1}-t_k-1)\frac{p_i-2/|E|}{1-p_j}\right)\frac{1}{p_i}\mid t_k\right]=\left(\frac{1}{d_j}+\left(\frac{1-p_j}{p_j}\right)\frac{p_i-2/|E|}{1-p_j}\right)\frac{1}{p_i}=\frac{1}{p_j}\ .$$

We can now use this relation in the full expectation

$$\mathbb{E}_T\left[\left(\frac{\delta_i(t)}{p_i}-\frac{\delta_j(t)}{p_j}\right)f_j(\boldsymbol{\theta}_j(t))\right]$$

$$=\mathbb{E}\left[\sum_{k=1}^{N_j-1}\left(\mathbb{E}\left[\mathbb{E}\left[\sum_{t=t_k}^{t_{k+1}-1}\frac{\delta_i(t)}{p_i}\Big|t_{k+1}-t_k\right]\Big|t_k\right]-\frac{1}{p_j}\right)f_j(\boldsymbol{\theta}_j(t_k))\right]=0\ . \tag{43}$$

Similarly if $(i,j)\notin E$, one has

$$\mathbb{E}[\delta_i(t_k)|t_k]=\mathbb{E}[\delta_i(t)|\delta_j(t)=1]=0\ ,$$

and for $t_k<t<t_{k+1}$,

$$\mathbb{E}[\delta_i(t)|t_k]=\mathbb{E}[\delta_i(t)|\delta_j(t)=0]=\frac{p_i}{1-p_j}\ ,$$

so the result of Equation (43) holds in this case. We have just shown that for every $j\in[n]$, we can use $\delta_j(t)f_j(\boldsymbol{\theta}_j(t))/p_j$ instead of $\delta_i(t)f_j(\boldsymbol{\theta}_j(t))/p_i$ . Combining (41) and (42) yields:

$$\mathbb{E}[R_n(\bar{\boldsymbol{\theta}}_i(T))]-R_n(\boldsymbol{\theta}^*)\leq\frac{1}{nT^-}\sum_{t=2}^{T}\sum_{j=1}^{n}\mathbb{E}\left[\frac{\delta_i(t)}{p_i}(f_j(\boldsymbol{\theta}_i(t))-f_j(\boldsymbol{\theta}_j(t)))\right]$$

$$+\frac{1}{nT^-}\sum_{t=2}^{T}\sum_{j=1}^{n}\mathbb{E}\left[\frac{\delta_j(t)}{p_j}\left(f_j(\boldsymbol{\theta}_j(t))-f_j(\boldsymbol{\theta}^*)\right)\right]$$

$$+\frac{1}{T^-}\sum_{t=2}^{T}\mathbb{E}\left[\frac{\delta_i(t)}{p_i}(\psi(\boldsymbol{\theta}_i(t))-\psi(\boldsymbol{\theta}^*))\right]$$

$$+\frac{f_j(0)}{p_ip_jT^-}+\frac{L_f^2\mathbb{E}[\gamma(t_{N_j}-1)]}{p_ip_j}\ .$$

Let us focus on the second term of the right hand side. For $t \geq 2$, one can write

$$\frac{1}{n}\sum_{j=1}^{n}\mathbb{E}\left[\frac{\delta_j(t)}{p_j}\left(f_j(\boldsymbol{\theta}_j(t)) - f_j(\boldsymbol{\theta}^*)\right)\right] \leq \frac{1}{n}\sum_{j=1}^{n}\mathbb{E}\left[\frac{\delta_j(t)}{p_j}\mathbf{g}_j(t)^\top(\boldsymbol{\theta}_j(t) - \boldsymbol{\theta}^*)\right]$$

$$= \frac{1}{n}\sum_{j=1}^{n}\mathbb{E}\left[\frac{\delta_j(t)}{p_j}\mathbf{g}_j(t)^\top(\boldsymbol{\theta}_j(t) - \widehat{\boldsymbol{\omega}}(t))\right]$$

$$+ \frac{1}{n}\sum_{j=1}^{n}\mathbb{E}\left[\frac{\delta_j(t)}{p_j}\mathbf{g}_j(t)^\top(\widehat{\boldsymbol{\omega}}(t) - \boldsymbol{\theta}^*)\right] \quad. \tag{44}$$

Here we control the term from (44) using $\widehat{\boldsymbol{\omega}}(t) := \pi_{m_i(t)}(\widehat{\mathbf{z}}(t))$

$$\frac{1}{n}\sum_{j=1}^{n}\mathbb{E}\left[\frac{\delta_j(t)}{p_j}\mathbf{g}_j(t)^\top(\widehat{\boldsymbol{\omega}}(t) - \boldsymbol{\theta}^*)\right] = \mathbb{E}\left[\left(\frac{1}{n}\sum_{j=1}^{n}\frac{\delta_j(t)}{p_j}\mathbf{g}_j(t)\right)^\top(\widehat{\boldsymbol{\omega}}(t) - \boldsymbol{\theta}^*)\right]$$

$$= \mathbb{E}\left[\widehat{\mathbf{g}}(t)^\top(\widehat{\boldsymbol{\omega}}(t) - \boldsymbol{\theta}^*)\right] \quad,$$

and the reasoning of the synchronous case can be applied to obtain

$$\frac{1}{nT^-}\sum_{t=2}^{T}\sum_{j=1}^{n}\mathbb{E}\left[\frac{\delta_j(t)}{p_j}\mathbf{g}_j(t)^\top(\widehat{\boldsymbol{\omega}}(t) - \boldsymbol{\theta}^*)\right]$$

$$\leq \frac{L^2}{2T^-}\sum_{t=2}^{T}\gamma(t-1) + \frac{\|\boldsymbol{\theta}^*\|^2}{2\gamma(T)} + \frac{1}{T}\sum_{t=2}^{T}\mathbb{E}[\widehat{\boldsymbol{\epsilon}}(t)^\top\widehat{\boldsymbol{\omega}}(t)] + \frac{1}{T^-}\sum_{t=2}^{T}(\psi(\boldsymbol{\theta}^*) - \mathbb{E}[\psi(\widehat{\boldsymbol{\omega}}(t))]). \tag{45}$$

We have:

$$\frac{1}{T^-}\sum_{t=2}^{T}\mathbb{E}\left[\frac{\delta_i(t)}{p_i}(\psi(\boldsymbol{\theta}_i(t)) - \psi(\boldsymbol{\theta}^*))\right] + \frac{1}{T^-}\sum_{t=2}^{T}(\psi(\boldsymbol{\theta}^*) - \mathbb{E}[\psi(\widehat{\boldsymbol{\omega}}(t))])$$

$$= \frac{1}{T^-}\sum_{t=2}^{T}\mathbb{E}\left[\frac{\delta_i(t)}{p_i}\psi(\boldsymbol{\theta}_i(t)) - \psi(\widehat{\boldsymbol{\omega}}(t))\right]$$

$$= \frac{1}{T^-}\sum_{t=2}^{T}\mathbb{E}\left[\frac{\delta_i(t)}{p_i}(\psi(\boldsymbol{\theta}_i(t)) - \psi(\widehat{\boldsymbol{\omega}}(t)))\right] + \frac{1}{T^-}\sum_{t=2}^{T}\mathbb{E}\left[(\frac{\delta_i(t)}{p_i} - 1)\psi(\widehat{\boldsymbol{\omega}}(t))\right]$$

$$= \frac{1}{T^-}\sum_{t=2}^{T}\mathbb{E}\left[\frac{\delta_i(t)}{p_i}(\psi(\boldsymbol{\theta}_i(t)) - \psi(\widehat{\boldsymbol{\omega}}(t)))\right] \quad,$$

where we have used for the last term the same arguments as in (43) to state $\frac{1}{T^-}\sum_{t=2}^{T}\mathbb{E}\left[(\frac{\delta_i(t)}{p_i} - 1)\psi(\widehat{\boldsymbol{\omega}}(t))\right] = 0$. Then, one can use the fact that $\pi_t$ is $\gamma(t)$-Lipschitz to write:

$$\frac{1}{p_iT^-}\sum_{t=2}^{T}\mathbb{E}\left[2L\gamma(m_i(t-1))\|\widehat{\mathbf{z}}(t) - \mathbf{z}_i(t)\| + \frac{\gamma(m_i(t-1))\|\widehat{\mathbf{z}}(t) - \mathbf{z}_i(t)\|^2}{2(m_i(t-1))}\right] \quad.$$

Provided that $\gamma(t) \leq C/\sqrt{t}$ for some constant $C$, then using Lemma 12 we can bound this term by $C'/\sqrt{T}$. Let us now control the following term:

$$\frac{1}{n}\sum_{j=1}^{n}\mathbb{E}\left[\frac{\delta_j(t)}{p_j}\mathbf{g}_j(t)^{\top}(\boldsymbol{\theta}_j(t) - \widehat{\boldsymbol{\omega}}(t))\right]$$

$$\leq \frac{L}{np_j}\sum_{j=1}^{n}\mathbb{E}\left[\|\boldsymbol{\theta}_j(t) - \widehat{\boldsymbol{\omega}}(t)\|\right] \tag{46}$$

$$\leq \frac{L}{np_j}\sum_{j=1}^{n}\mathbb{E}\left[\|\boldsymbol{\theta}_j(t) - \tilde{\boldsymbol{\theta}}_j(t)\| + \|\tilde{\boldsymbol{\theta}}_j(t) - \widehat{\boldsymbol{\omega}}(t)\|\right]$$

$$\leq \frac{L}{np_j}\sum_{j=1}^{n}\mathbb{E}\left[\gamma(m_j(t-1))\|z_j(t) - \widehat{\mathbf{z}}(t)\| + \|\tilde{\boldsymbol{\theta}}_j(t) - \widehat{\boldsymbol{\omega}}(t)\|\right] .$$

where $\tilde{\boldsymbol{\theta}}_j(t) = \pi_{m_j(t-1)}(-\bar{\mathbf{z}}(t))$. We can apply Lemma 13 with the choice $\boldsymbol{\theta}_1 = \tilde{\boldsymbol{\theta}}_j(t)$, $\boldsymbol{\theta}_2 = \widehat{\boldsymbol{\omega}}(t)$, $t_1 = m_j(t)$, $t_2 = m_i(t)$ and $z = \widehat{\mathbf{z}}(t)$:

$$\|\widehat{\boldsymbol{\omega}}(t) - \tilde{\boldsymbol{\theta}}_j(t)\|$$
$$\leq \|\widehat{\mathbf{z}}(t)\||\gamma(m_i(t)) - \gamma(m_j(t))|$$
$$+ \|\widehat{\mathbf{z}}(t)\|\left(\frac{3}{2} + \max\left(\frac{\gamma(m_j(t))}{\gamma(m_i(t))}, \frac{\gamma(m_i(t))}{\gamma(m_j(t))}\right)\right)\left|\gamma(m_j(t)) - \frac{m_i(t)}{m_j(t)}\gamma(m_i(t))\right|\right)$$
$$+ \|\widehat{\mathbf{z}}(t)\|\left(\frac{3}{2} + \max\left(\frac{\gamma(m_j(t))}{\gamma(m_i(t))}, \frac{\gamma(m_i(t))}{\gamma(m_j(t))}\right)\right)\left|\gamma(m_i(t)) - \frac{m_j(t)}{m_i(t)}\gamma(m_j(t))\right|\right) .$$

We use Lemma 12 with the choice $q = \alpha/2$, so we can bound for $t$ large enough the former expression by a term of order $\|\widehat{\mathbf{z}}(t)\||\gamma(m_i(t)) - \gamma(m_j(t))|$. Note also that $\|\widehat{\mathbf{z}}(t)\| \leq L\max_{k=1,\dots,n} m_k(t)$, so for $t$ large enough we obtain:

$$\|\widehat{\boldsymbol{\omega}}(t) - \tilde{\boldsymbol{\theta}}_j(t)\| \leq Lt^{+}|\gamma(t^{-}) - \gamma(t^{+})| .$$

With the additional constraint that the step size is of the form $\gamma(t) = Ct^{-1/2-\alpha}$, the term $\|\widehat{\boldsymbol{\omega}}(t) - \tilde{\boldsymbol{\theta}}_j(t)\|$ is bounded by $C't^{-\alpha/2}$ for $t$ large enough, and so is $(1/n)\sum_{j=1}^{n}\mathbb{E}\left[\frac{\delta_j(t)}{p_j}g_j(t)^{\top}(\boldsymbol{\theta}_j(t) - \widehat{\boldsymbol{\omega}}(t))\right]$.

To control the objectives, we use that $f_j$ is $L$-Lipschitz

$$|f_j(\boldsymbol{\theta}_i(t)) - f_j(\boldsymbol{\theta}_j(t))| \leq L\|\boldsymbol{\theta}_i(t) - \boldsymbol{\theta}_j(t)\| \leq L(\|\boldsymbol{\theta}_i(t) - \widehat{\boldsymbol{\omega}}(t)\| + \|\widehat{\boldsymbol{\omega}}(t) - \boldsymbol{\theta}_j(t)\|) ,$$

and we use now the same control as for (46), hence the result.

$\square$

**Lemma 13.** *Let $\gamma : \mathbb{R}_+ \to \mathbb{R}_+$ be a non-increasing positive function and let $\mathbf{z} \in \mathbb{R}^d$. For any $t_1, t_2 > 0$, one has*

$$\|\boldsymbol{\theta}_2 - \boldsymbol{\theta}_1\| \leq \|\mathbf{z}\||\gamma(t_2) - \gamma(t_1)| + \|\mathbf{z}\|\left(\frac{3}{2} + \max(\frac{\gamma(t_1)}{\gamma(t_2)}, \frac{\gamma(t_2)}{\gamma(t_1)})\right)\left(\frac{1}{t_1} + \frac{1}{t_2}\right)|t_1\gamma(t_1) - t_2\gamma(t_2)| ,$$

*where, for $i \in \{1, 2\}$,*

$$\boldsymbol{\theta}_i = \pi_{t_i}(\mathbf{z}) := \arg\max_{\boldsymbol{\theta}\in\mathbb{R}^d}\left\{\mathbf{z}^{\top}\boldsymbol{\theta} - \frac{\|\boldsymbol{\theta}\|^2}{2\gamma(t_i)} - t_i\psi(\boldsymbol{\theta})\right\} .$$

*Proof.* Using the optimality of the minimizer, for any $s_1 \in \partial\psi(\boldsymbol{\theta}_1)$ (resp. $s_2 \in \partial\psi(\boldsymbol{\theta}_2)$):

$$(\gamma(t_1)\mathbf{z} - t_1\gamma(t_1)s_1 - \boldsymbol{\theta}_1)^{\top}(\boldsymbol{\theta}_2 - \boldsymbol{\theta}_1) \leq 0$$
$$(\gamma(t_2)\mathbf{z} - t_2\gamma(t_2)s_2 - \boldsymbol{\theta}_2)^{\top}(\boldsymbol{\theta}_1 - \boldsymbol{\theta}_2) \leq 0$$

Re-arranging the terms, and using properties of sub-gradients yields:

$$\|\boldsymbol{\theta}_2 - \boldsymbol{\theta}_1\|^2 \leq (\gamma(t_2) - \gamma(t_1))\mathbf{z}^\top (\boldsymbol{\theta}_2 - \boldsymbol{\theta}_1) + (t_1\gamma(t_1)s_1 - t_2\gamma(t_2)s_2)^\top (\boldsymbol{\theta}_2 - \boldsymbol{\theta}_1) \tag{47}$$
$$\leq (\gamma(t_2) - \gamma(t_1))\mathbf{z}^\top (\boldsymbol{\theta}_2 - \boldsymbol{\theta}_1) + (t_1\gamma(t_1) - t_2\gamma(t_2))(\psi(\boldsymbol{\theta}_2) - \psi(\boldsymbol{\theta}_1))$$

Also, using the definition of $\boldsymbol{\theta}_1$ and $\boldsymbol{\theta}_2$, one has:

$$|\psi(\boldsymbol{\theta}_1) - \psi(\boldsymbol{\theta}_1)| \leq \|\mathbf{z}\| \|\boldsymbol{\theta}_1 - \boldsymbol{\theta}_2\| \left(\frac{3}{2} + \max(\frac{\gamma(t_1)}{\gamma(t_2)}, \frac{\gamma(t_2)}{\gamma(t_1)})\right) \left(\frac{1}{t_1} + \frac{1}{t_2}\right). \tag{48}$$

With relations (47) and (48) we bound the distance between $\boldsymbol{\theta}_1$ and $\boldsymbol{\theta}_2$ as follows:

$$\|\boldsymbol{\theta}_2 - \boldsymbol{\theta}_1\| \leq \|\mathbf{z}\| |\gamma(t_2) - \gamma(t_1)| + \|\mathbf{z}\| \left(\frac{3}{2} + \max(\frac{\gamma(t_1)}{\gamma(t_2)}, \frac{\gamma(t_2)}{\gamma(t_1)})\right) \left(\frac{1}{t_1} + \frac{1}{t_2}\right) |t_1\gamma(t_1) - t_2\gamma(t_2)|$$

$\square$

### D.3 Proof of Theorem 3

In this section, we state the proof of Theorem 3.

*Proof.* Let $\mathcal{G} = ([n], \mathcal{E})$ be a connected graph. We consider decentralized first-order methods that, at each iteration, can query local (sub)gradients and exchange messages along $\mathcal{E}$. For simplicity, we assume unit per-hop latency $\tau = 1$; heterogeneous latencies only change the bound by a multiplicative factor.

Following Scaman et al. (2018), the idea is to construct an objective function whose coordinates must be activated sequentially, forcing information to alternate between two distant nodes of the network. Fix two nodes $i_0, i_1 \in \mathcal{V}$. Consider the local functions

$$f_j(\boldsymbol{\theta}) = \begin{cases} \frac{\alpha}{2}\|\boldsymbol{\theta}\|^2 + n\left[\gamma \sum_{r=1}^{k} |\theta_{2r+1} - \theta_{2r}| - \beta\,\theta_1\right] & \text{if } j = i_0, \\ \frac{\alpha}{2}\|\boldsymbol{\theta}\|^2 + n\left[\gamma \sum_{r=1}^{k} |\theta_{2r} - \theta_{2r-1}| + \delta\,\theta_{2k+1}\right] & \text{if } j = i_1, \\ \frac{\alpha}{2}\|\boldsymbol{\theta}\|^2 & \text{otherwise.} \end{cases}$$

Here $\alpha > 0$ controls strong convexity, while $\gamma, \beta, \delta > 0$ control the nonsmooth zero-preserving terms. The overall objective $F(\boldsymbol{\theta}) = \frac{1}{n} \sum_j f_j(\boldsymbol{\theta})$ is convex, nonsmooth, and designed so that each gap $|\theta_{2r+1} - \theta_{2r}|$ is controlled by $i_0$, and each gap $|\theta_{2r} - \theta_{2r-1}|$ by $i_1$. Starting from $\boldsymbol{\theta}(0) = \mathbf{0}$, a first-order algorithm can activate at most one new coordinate per communication round between $i_0$ and $i_1$.

In the centralized setting, the nonsmooth lower bound is obtained by considering the family of functions (**?**, see Lemma 3.1):

$$f(\boldsymbol{\theta}) = \frac{\alpha}{2}\|\boldsymbol{\theta}\|^2 + \max_{1 \leq i \leq t} \theta_i \ .$$

This function is convex and 1-Lipschitz on the $\ell_2$ ball of radius $R = \sqrt{t}$, and is $\alpha$-strongly convex. For any deterministic first-order algorithm, after $s$ oracle calls the error satisfies

$$f(\boldsymbol{\theta}(T)) - f^\star \geq c\frac{RL}{\sqrt{T+1}} \ ,$$

for some universal constant $c > 0$. This lower bound relies on the fact that each oracle query can reveal information about at most *one active coordinate* in the max-term, hence the suboptimality decays only as $1/\sqrt{T}$.

In the decentralized setting, each activation requires a communication round-trip between $i_0$ and $i_1$, of length at least $2d_{\mathcal{G}}(i_0, i_1)$, where $d_{\mathcal{G}}(\cdot, \cdot)$ is the graph distance. This dilates the effective time parameter by $2d_{\mathcal{G}}(i_0, i_1)$, and the careful analysis in Scaman et al. (2018) yields, for any $i \in [n]$:

$$F(\boldsymbol{\theta}_i(t)) - \min_{\|\boldsymbol{\theta}\| \leq R} F(\boldsymbol{\theta}) \geq \frac{RL}{36} \sqrt{\frac{1}{(1 + t/(2d_{\mathcal{G}}(i_0, i_1)))^2} + \frac{1}{t+1}} \ ,$$

where $d_{\mathcal{G}}(i_0, i_1)$ is the distance between nodes $i_0$ and $i_1$. Selecting $i_0$ and $i_1$ as to maximize this distance yield a bound where $d_{\mathcal{G}}(i_0, i_1)$ can be replaced by the graph diameter $\Delta$.

In our setting, each local function $f_i$ decomposes as a sum of *pairwise* terms

$$f_i(\boldsymbol{\theta}) = \sum_{j \in [n]} f_{ij}(\boldsymbol{\theta}),$$

where each $f_{ij}$ is stored jointly at the pair $(i, j)$ and depends on both nodes. We can therefore split the nonsmooth chain terms of the previous construction across the network.

Specifically, let $j_{0,1}, \ldots, j_{0,k} \in [n]$ and $j_{1,1}, \ldots, j_{1,k} \in [n]$ be two sets of distinct, intermediary nodes. We then define pairwise functions $f_{uv}$ as follows:

$$f_{uv}(\boldsymbol{\theta}) = \begin{cases} \frac{\alpha}{2}\|\boldsymbol{\theta}\|^2 - n\beta\,\theta_1 & \text{if } u = v = i_0, \\ \frac{\alpha}{2}\|\boldsymbol{\theta}\|^2 + n\delta\,\theta_{2k+1} & \text{if } u = v = i_1, \\ \frac{\alpha}{2}\|\boldsymbol{\theta}\|^2 + n\gamma\,|\theta_{2r+1} - \theta_{2r}| & \text{if } (u,v) = (i_0, j_{0,r}), \\ \frac{\alpha}{2}\|\boldsymbol{\theta}\|^2 + n\gamma\,|\theta_{2r} - \theta_{2r-1}| & \text{if } (u,v) = (i_1, j_{1,r}), \\ \frac{\alpha}{2}\|\boldsymbol{\theta}\|^2 & \text{otherwise.} \end{cases}$$

The global objective $F(\boldsymbol{\theta})$ remains unchanged, but now, computing the gradient of $f_{i_0,j_{0,r}}$ (resp. $f_{i_1,j_{1,r}}$) requires routing information from $j_{0,r}$ to $i_0$ (resp. $j_{1,r}$ to $i_1$). This effectively replaces each occurrence of $d_{\mathcal{G}}(i_0, i_1)$ in the activation time by the length of the path from $i_0$ to $j_{0,r}$ and from $j_{0,r}$ to $i_1$ as well as from $i_1$ to $j_{1,r}$ and from $j_{1,r}$ to $i_0$.

Let $\tilde{d}_{\mathcal{G}}(i_0, i_1)$ be the average effective communication length between $i_0$, $i_1$ and intermediary nodes, that is

$$\tilde{d}_{\mathcal{G}}(i_0, i_1) := \frac{1}{2k} \sum_{r=1}^{k} d_{\mathcal{G}}(i_0, j_{0,r}) + d_{\mathcal{G}}(j_{0,r}, i_1) + d_{\mathcal{G}}(i_1, j_{1,r}) + d_{\mathcal{G}}(j_{1,r}, i_0) \ .$$

This quantity represents the *average communication length per coordinate activation, i.e.,* the effective distance that must be covered by each new gradient piece, on average. Repeating the same zero-preserving argument as in Scaman et al. (2018) and optimizing over $(i_0, i_1)$ then gives the lower bound:

$$F(\boldsymbol{\theta}_i(t)) - \min_{\|\boldsymbol{\theta}\| \leq R} F(\boldsymbol{\theta}) \geq \frac{RL}{36} \sqrt{\frac{1}{(1 + t/\tilde{\Delta})^2} + \frac{1}{t+1}} \ .$$

$\square$

# E    Multiple points per node

*Proof.* Denoting the Kronecker product by $\otimes$, we can write:

$$\mathbf{A}^{G^{\otimes}} = \mathbf{1}_k \mathbf{1}_k^T \otimes \mathbf{A}^G \quad \text{and} \quad \mathbf{D}^{G^{\otimes}} = k\mathbf{I}_k \otimes \mathbf{D}^G.$$

Recall that $\mathbf{L}^G = \mathbf{D}^G - \mathbf{A}^G$ and $\mathbf{L}^{G^{\otimes}} = \mathbf{D}^{G^{\otimes}} - \mathbf{A}^{G^{\otimes}}$. Let $(\boldsymbol{\phi}, \lambda) \in \mathbb{R}^{nk} \times \mathbb{R}$ be an eigenpair of $\mathbf{L}^{G^{\otimes}}$, *i.e.*, $(\mathbf{D}^{G^{\otimes}} - \mathbf{A}^{G^{\otimes}})\boldsymbol{\phi} = \lambda\boldsymbol{\phi}$ and $\boldsymbol{\phi} \neq 0$. Let us write $\boldsymbol{\phi} = [\boldsymbol{\phi}_1^\top \ldots \boldsymbol{\phi}_k^\top]^\top$ where $\boldsymbol{\phi}_1, \ldots, \boldsymbol{\phi}_k \in \mathbb{R}^n$. Exploiting the structure of $\mathbf{A}^{G^{\otimes}}$ and $\mathbf{D}^{G^{\otimes}}$, we have:

$$k\mathbf{D}^G \boldsymbol{\phi}_i - \sum_{j=1}^{k} \mathbf{A}^G \boldsymbol{\phi}_j = \lambda\boldsymbol{\phi}_i, \quad \text{for all } i \in [k]. \tag{49}$$

Summing up (49) over all $i \in [k]$ gives

$$\mathbf{D}^G \sum_{i=1}^{k} \boldsymbol{\phi}_i - \mathbf{A}^G \sum_{i=1}^{k} \boldsymbol{\phi}_i = \frac{\lambda}{k} \sum_{i=1}^{k} \boldsymbol{\phi}_i \ ,$$

which shows that if $(\boldsymbol{\phi}, \lambda)$ is an eigenpair of $\mathbf{L}^{G^{\otimes}}$ with $\sum_{i=1}^{k} \boldsymbol{\phi}_i \neq 0$, then $(\sum_{i=1}^{k} \boldsymbol{\phi}_i, \lambda/k)$ is an eigenpair of $\mathbf{L}^G$. In the case where $\sum_{i=1}^{k} \boldsymbol{\phi}_i = 0$, then there exists an index $j \in [k]$ such that $\boldsymbol{\phi}_j = -\sum_{i \neq j} \boldsymbol{\phi}_j \neq 0$. Hence (49) gives

$$\mathbf{D}^G \boldsymbol{\phi}_j = \frac{\lambda}{k} \boldsymbol{\phi}_j,$$

which shows that $(\boldsymbol{\phi}_j, \lambda/k)$ is an eigenpair of $\mathbf{L}^G$. Observe that $\lambda = kd_i$ for some $i \in [n]$.

We have thus shown that any eigenvalue $\lambda^{G^{\otimes}}$ of $\mathbf{L}^{G^{\otimes}}$ is either of the form $\lambda^{G^{\otimes}} = k\lambda^G$, where $\lambda^G$ is an eigenvalue of $\mathbf{L}^G$, or of the form $\lambda^{G^{\otimes}} = kd_i$ for some $i \in [n]$.

Since $\mathbf{L}^{G^{\otimes}}$ is a Laplacian matrix, its smallest eigenvalue is 0. Let $\lambda_{nk-1}^{G^{\otimes}}$ be the second smallest eigenvalue of $\mathbf{L}^{G^{\otimes}}$. Note that $G^{\otimes}$ is not a complete graph since $G$ is not complete. Therefore, $\lambda_{nk-1}^{G^{\otimes}}$ is bounded above by the vertex connectivity of $G^{\otimes}$ (Fiedler, 1973), which is itself trivially bounded above by the minimum degree $d_{min}^{\otimes} = \min_{i \in [kn]} [\mathbf{D}^{G^{\otimes}}]_{ii}$ of $G^{\otimes}$. This implies that $\lambda_{nk-1}^{G^{\otimes}} = k\lambda_{n-1}^G$, and hence

$$1 - \lambda_2^{G^{\otimes}}(2) = \frac{\lambda_{kn-1}^{G^{\otimes}}}{|E^{\otimes}|} = \frac{k\lambda_{n-1}^G}{k^2|E|} = \frac{1 - \lambda_2^G(2)}{k}.$$

$\square$

