# OpenReview forum: "On Gossip Algorithms for Machine Learning with Pairwise Objectives"
_TMLR — Accepted by TMLR_

### Review · Reviewer_Qri3 · 2025-10-18

**Summary Of Contributions:**

The paper addresses a gap in centralized machine learning, specifically in estimating and optimizing functions that involve pairwise dependencies on the data that is, U-statistics of degree two. The paper makes a strong and clear contribution to this problem by refining and extending the work of Colin et al. on decentralized estimation and by deriving new bounds for decentralized optimization (all focused on degree-2 U-statistics). The authors clearly articulate these contributions in the introduction.

**Audience:**

Yes

**Audience Explanation:**

Yes, definitely! In terms of importance and relevance, the proposed results are highly informative for the community. They provide a theoretical analysis of a practical algorithm that explicitly depends on key quantities such as the data "dispersion" and network topology. Moreover, the analysis of the bias term $C_3(T)$ is technically valuable and, I believe, represents a great contribution of the paper.

**Broader Impact Concerns:**

I don’t think the paper raises any ethical implications or concerns.

**Claims And Evidence:**

Yes

**Claims Explanation:**

Yes, all theoretical claims are supported by theorems and corollaries, each accompanied by proofs. Some empirical statements could be further validated through experiments, which would improve the quality of the paper.

**Requested Changes:**

Overall, I find the paper to be of high quality. My only suggestion would be to include some simple experimental evidence to support the theoretical claims. The paper mentions multiple times that empirical results confirm the theory and that the bias term vanishes quickly in experiments; however, including a few plots to illustrate this behavior would make those claims more convincing.

---

### Review · Reviewer_sSar · 2025-11-06

**Summary Of Contributions:**

This paper studied the decentralized optimization problem where the loss function is not separable across nodes. This problem setting encompasses several applications, including ranking and clustering. First, this paper analyzed the convergence behavior of the existing method, GOSTA-Sync, and provided a novel analysis of it. GOSTA-Sync is the method to estimate the function defined in Eq. (5).
More specifically, the existing analysis established the convergence of $\| \mathbb{E} [z] - \hat{U}_n \mathbf{1}_n \|$, while this paper analyzed $\mathbb{E} \| z - \hat{U}_n \mathbf{1}_n \|$. Then, this paper studied the optimization problem described in Eq. (7), and analyzed the existing optimization method, described in Algorithm 2. The convergence rate is analyzed in Theorem 2 and Corollary 1.

**Audience:**

Yes

**Audience Explanation:**

The problem addressed in this paper appears to be important, and it is good that this paper analyzes the convergence behavior of basic methods, such as GOSTA-Sync and Algorithm 2.

**Broader Impact Concerns:**

There are no ethical concerns.

**Claims And Evidence:**

Yes

**Claims Explanation:**

## Strength
* The problem that this paper addressed seems to have several interesting applications, including clustering and ranking.
* This paper provided the refined analysis for the existing methods, GOSTA-Sync and Algorithm 2.

## Weakness
* The convergence rate presented in Corollary 1 is strange. The left-hand side represents function values, while the right-hand side is, roughly speaking, of order $O(L^2)$. These two sides have different units. More specifically, let the unit of the function be $[f]$ and the unit of the parameter be $[\theta]$. The unit of the Lipschitz coefficient is $[f] / [\theta]$. Thus, the left-hand side has unit $[f]$, whereas the right-hand side has unit $( [f] / [\theta])^2$. It appears inappropriate to compare quantities with different units. I suspect that there might be an error in the proof or that the choice of hyperparameters, e.g., $\gamma (t)$, is suboptimal, or $\| \theta_0 - \theta^\star\|$ is missing.
* Can the author show the explicit equation hidden in $o(1/ \sqrt{T})$ in Corollary 1? The remaining terms are also $O(1 / \sqrt{T})$, and it is unclear whether $o(1 / \sqrt{T})$ diminishes faster than the rest.
* The quantity of $D$ in Theorem 2 and Corollary 1 seems not to be defined.
* The convergence rate shown in Theorem 3 looks a bit strange. The left-hand side and right-hand side depend on $R$, and it is a bit hard to understand this lower bound. This paper claims that "The lower bound for decentralized pairwise optimization is similar to the known lower bounds for separable objective" in page 13, but this paper did not cite the reference. Can the authors cite the appropriate reference here and provide additional discussion on $R$? Furthermore, is it guaranteed that the left-hand side is always positive?
* Is a gossip-based method truly communication-efficient for the problem addressed in this paper? For example, when solving the problem in Eq. (6), each node $k$ can obtain all the observations $\\{ x_j \\}_{j=1}^n$​ in finite time by using some operation. The required communication cost in this case seems to be proportional to the diameter of the communication graph. Compared with such a simple approach, methods like GOSTA-Sync and Algorithm 2 appear to be less communication-efficient. In what kind of scenarios is it necessary to use gossip-based methods?
* This paper did not check the performance of these methods numerically.

**Requested Changes:**

See the weakness section.

---

### Review · Reviewer_q5Em · 2025-12-24

**Summary Of Contributions:**

This paper addresses the problem of analyzing gossip algorithms for pairwise objectives in decentralized settings. Focusing on objectives formulated as U-statistics of degree two, the authors provide a non-asymptotic convergence analysis of the GOSTA algorithm. Additionally, the work establishes a theoretical framework by deriving fundamental lower bounds for learning with pairwise objectives in distributed environments.


Strengths

- The paper provides a clear and mathematical introduction to gossip algorithms and establishes a good notation framework for the problem setting.

-  The paper attempts a rigorous non-asymptotic analysis, addressing a gap in the literature regarding the expected error (variance + bias) of the GOSTA algorithm, rather than just the bias.

Weaknesses

- The introduction fails to compellingly justify the problem setting. The connection between general IoT/sensor network discussions and the specific pairwise objective contributions is unclear.

- Section 2.2 introduces the pairwise objective abruptly without sufficient motivation or simple examples. It is unclear to the reader why decentralized pairwise averaging is a critical problem worth solving compared to standard averaging.

- The paper spends a significant amount of space (Sections 1-3.1) recapping background and existing algorithms (e.g., GOSTA from Colin et al., 2015). The novel contributions do not appear until Section 3.2, making the first half of the paper feel repetitive of prior work.

- Proposition 1 is presented without discussing the technical challenges involved in the derivation. The authors do not clarify how their analysis overcomes the difficulties of U-statistics in a decentralized setting compared to existing work. Is the expectation inside the norm in Proposition 1?

- The paper is purely theoretical and lacks empirical evidence. Experiments demonstrating the convergence behavior and the practical benefits of the proposed analysis on real-world datasets would help to validate the theoretical claims.

**Audience:**

Yes

**Audience Explanation:**

The paper focuses on non-asymptotic analysis of gossip algorithms for pairwise objective, which can be of interest for theoretical research in this area.

**Claims And Evidence:**

No

**Claims Explanation:**

Please refer to the weakness above.

**Requested Changes:**

Please address the weakness mentioned above.

---

> ### Author Response · Authors · 2026-01-16
>
> We thank the reviewer for their careful reading and constructive feedback. We address each point below and outline concrete modifications we will implement.
>
> **Motivation and problem setting.**
> We agree that the motivation can be strengthened. While the introduction discusses IoT and sensor networks as representative decentralized systems, the connection to pairwise objectives may not be sufficiently explicit.
>
> In the revised version, we will clarify that pairwise objectives naturally arise in decentralized settings where data points stored at different nodes must be compared, such as ranking, AUC maximization, anomaly detection, similarity learning, and clustering. In these cases, the objective is fundamentally non-separable and cannot be written as a simple average of local losses, which distinguishes this setting from classical decentralized averaging.
>
> We will explicitly emphasize that the goal of the paper is to understand how such non-separable objectives---formally captured by $U$-statistics of order two---can be estimated and optimized using gossip-based communication under realistic constraints on memory and communication.
>
> **Pairwise objectives and examples (Section 2.2).**
> We respectfully believe this point results from a misunderstanding. Section 2.2 is explicitly devoted to motivating pairwise objectives through several examples, including ranking, AUC maximization and clustering-type criteria.
> That said, to further improve readability, we will add a short running example at the beginning of Section 2.2 and explicitly refer to it throughout the section, so that the reader can more easily follow the different formulations and see how they fit into a unified framework.
>
> **Background material and placement of contributions.**
> We acknowledge that the presentation of the contributions could be made more prominent earlier in the paper. The background sections were intended to make the paper self-contained and accessible, but we agree that this may delay the exposition of the main results.
>
> In the revised version, we will (i) explicitly summarize our contributions at the end of the introduction, (ii) shorten or move part of the background material to the appendix, and (iii) highlight the novel technical elements at the beginning of Section 3 before presenting detailed proofs.
>
> **Proposition 1 and technical challenges of $U$-statistics.**
> We agree that the presentation can be clarified. In the revised version, we will explicitly clarify where expectations are taken in Proposition 1 and add a discussion immediately after the proposition explaining the main technical challenge addressed by our analysis.
>
> In contrast to classical decentralized averaging, where all data points are already present on the network and uniformly sampled from the start, the pairwise setting requires the progressive dissemination of auxiliary data across the graph. As a result, the local estimates are initially formed from a non-uniformly distributed sampling of pairs, which only converges to the uniform distribution after sufficient mixing of the auxiliary variables through gossip interactions.
>
> The core difficulty therefore lies in quantifying how the non-uniformity impacts the estimates over time and in showing that the induced bias decays at a rate controlled by the mixing properties of the communication graph. This perspective motivates the refined non-asymptotic analysis developed in Section 3.2, and is further explored for analyzing the optimization process.
>
>
> **Experimental validation.**
> We agree that the current version lacks experimental results. We are currently implementing numerical experiments that illustrate (i) the convergence behavior predicted by our bounds, (ii) the bias induced by pairwise gossip sampling, and (iii) the impact of network topology and synchronization (synchronous versus asynchronous gossip). These experiments will be included in the revised version.

---

### Author Response · Authors · 2026-01-16

Dear Reviewers,

Thank you for your careful reading of the manuscript and for your constructive comments.

We have prepared an updated version of the paper addressing the points raised in the reviews. All revisions are highlighted in blue in the manuscript, except for the experimental section, which has been entirely rewritten and expanded.
We believe that the revised version better reflects the scope of the contributions and clarifies several aspects of the analysis. In addition, further numerical experiments are currently running and will be added to the supplementary material as soon as they are completed.

We thank you again for your time and valuable feedback.

---

> ### Author Response · Authors · 2026-02-10
>
> Dear editors, dear reviewers,
>
> We are writing to kindly follow up on the status of our submission.
>
> We understand that the reviewers and the editorial board have busy schedules. However, we wanted to ensure that the revision was successfully received and check if there is any further information or clarification needed from our side to facilitate the final decision.
>
> Thank you for your time and for managing the review process for our work. We look forward to hearing from you.

---

### Decision · Action_Editor_2jTC · 2026-02-11

**Recommendation:** Accept as is

**Audience:**

Yes

**Audience Explanation:**

The pairwise objective prevails in machine learning. Although the initial submission has not adequately discussed its motivation, the authors sufficiently spend their effort to clarify the practical scenarios based on the reviewer's feedback.

**Claims And Evidence:**

Yes

**Claims Explanation:**

This paper addresses the problem of pairwise estimation/optimization under the decentralized scenario. This specific topic on the pairwise objective faces an additional challenge over the standard decentralized optimization, and has scarce existing literature. In particular, the vanishing convergence rate for the pairwise optimization has not been obtained so far, where the contribution of this paper lies. By building upon the Gossip-type algorithm, the authors derive 1/sqrt(T) rate for the pairwise optimization.

At the initial phase, the main concerns on this submission were largely divided into: (1) insufficient motivation of the pairwise decentralized optimization; (2) the validity of the derived convergence rate (in terms of the constant dependency); and (3) the lack of numerical simulation. The authors have addressed in all aspects to improve their manuscript, and the reviewers unanimously agree to accept the revised version. Given the high standard of the technical contributions, AE suggests the paper acceptance as is.